

# Relative entropy in scattering and the S-matrix bootstrap

**Anjishnu Bose**[1*], **Parthiv Haldar**[1†], **Aninda Sinha**[1‡],
**Pritish Sinha**[2∘] **and Shaswat S. Tiwari**[1•]

**1** Centre for High Energy Physics, Indian Institute of Science,
C.V. Raman Avenue, Bangalore 560012, India
**2** Chennai Mathematical Institute, H1, SIPCOT IT Park, Siruseri, Kelambakkam 603103, India

⋆ anjishnubose98@gmail.com, † parthivh@iisc.ac.in, ‡ asinha@iisc.ac.in,
∘ pritishsinha06161@gmail.com, • shaswat10041998@gmail.com

## Abstract

We consider entanglement measures in 2-2 scattering in quantum field theories, focusing on relative entropy which distinguishes two different density matrices. Relative entropy is investigated in several cases which include $\phi^4$ theory, chiral perturbation theory ($\chi PT$) describing pion scattering and dilaton scattering in type II superstring theory. We derive a high energy bound on the relative entropy using known bounds on the elastic differential cross-sections in massive QFTs. In $\chi PT$, relative entropy close to threshold has simple expressions in terms of ratios of scattering lengths. Definite sign properties are found for the relative entropy which are over and above the usual positivity of relative entropy in certain cases. We then turn to the recent numerical investigations of the S-matrix bootstrap in the context of pion scattering. By imposing these sign constraints and the $\rho$ resonance, we find restrictions on the allowed S-matrices. By performing hypothesis testing using relative entropy, we isolate two sets of S-matrices living on the boundary which give scattering lengths comparable to experiments but one of which is far from the 1-loop $\chi PT$ Adler zeros. We perform a preliminary analysis to constrain the allowed space further, using ideas involving positivity inside the extended Mandelstam region, and other quantum information theoretic measures based on entanglement in isospin.

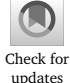

# 1  Introduction and summary

It is often a worthwhile pursuit to tackle old problems using new tools. In this paper, we will consider the very standard 2-2 scattering in quantum field theory using certain tools in quantum information theory. In particular, it should be of considerable interest to figure out how much entanglement is generated in various scattering processes, relevant not only for particle physics but also for condensed matter physics. Since various quantum information measures to address such questions exist [1, 2], it is then natural for us to examine if the properties and behaviour of such measures can shed new light on this age-old problem in physics.

Scattering theory is usually set up using momentum space in quantum field theory. In the simplest scenario of spin-less scattering, the incoming and outgoing states are specified in terms of the momenta of particles. When one computes entanglement entropy, this entails tracing over momentum states [3–7]. This line of questioning in itself is not novel. However, these earlier works also showed that entanglement entropy on its own is ill-defined as the expression is divergent owing to the infinite space-time volume and requires (sometimes ad-hoc) regularizations.

In this paper, we will consider a different measure, quantum relative entropy and more generally Rényi divergences, which will be not only free of regularizations but also have several other uses that we will explore. As far as the question of divergence and therefore regularization is concerned, one can consider variations in entanglement entropy, instead of entanglement entropy itself, which would get rid of the divergences. However, we choose to investigate relative entropy which is a bona fide quantum information quantity having applications in numerous places [1, 2]. Fixing regularizations in an ad hoc manner runs the risk of having a residual constant term, an eventuality we want to avoid dealing with. As explained in [1], relative entropy enables us to perform hypothesis testing. If we describe our observations using theory T1, but the correct theory is, in fact, T2, then how sure can we be after $N$ observations that T1 is wrong? As explained in [1], the confidence that T1 is wrong is controlled by

$$2^{-ND(\rho_{T_2}\|\rho_{T_1})},$$

where $D(\rho_1\|\rho_2)$ is the relative entropy between the two density matrices $\rho_1$ and $\rho_2$ explicitly given by

$$D(\rho_1\|\rho_2) = \text{Tr}\rho_1(\ln\rho_1 - \ln\rho_2). \tag{1.1}$$

Thus the larger $D(\rho_1\|\rho_2)$ is, the fewer observations will be needed to reach a certain confidence limit. We will see how to make use of hypothesis testing in scattering.

We could consider relative entropy in a different way as well. In 2-2 scattering we can take $\rho_1$ and $\rho_2$ to be the reduced density matrices corresponding to one of the outgoing particles reaching detectors placed at certain angles ($\cos\theta = x$) in the centre of mass frame. In such a scenario, where we consider Gaussian detectors of width $\sigma$ and small angular separation $\Delta x$, we will be able to show that

$$D\left(\rho_A^{(1)}\|\rho_A^{(2)}\right) \approx \frac{(\Delta x)^2}{4\sigma} + \frac{(\Delta x)^2}{2}\frac{\partial^2}{\partial x^2}\left(\ln\left(\frac{\sigma'_{el}(x)}{\sigma'_{el}(x_1)}\right)\right)\Bigg|_{x=x_1}, \qquad (1.2)$$

where $\sigma'_{el}(x) = d\sigma_{el}/dx$ is the elastic differential cross-section. The first term in this formula has no angular dependence and can be identified with a "hard-sphere" type scattering. The second term is independent of the width of the detector, $\sigma$, and is in some sense the universal piece. We have dropped sub-leading terms of $O(\sigma, (\Delta x)^3)$. This formula will prove very useful since the known behaviour of the elastic differential cross-section can be used to derive interesting properties for this relative entropy. In the future, one could also exploit this formula to examine experimental data from colliders.

We will also be able to derive novel high energy bounds on relative entropy which arise from general considerations such as analyticity, locality, unitarity similar in spirit to the famous Froissart bound [8,9]. We will show using the results of [10] that eq.(1.2), for $s \to \infty$ leads to

$$D\left(\rho_A^{(1)}\|\rho_A^{(2)}\right) - \frac{(\Delta x)^2}{4\sigma} \leq 2(\Delta x)^2\frac{(1+5x_1^2)}{(1-x_1^2)^2}, \qquad (1.3)$$

for fixed angle and where there are no unknown constants on the RHS. As we will further show, the tree level type II string theory answer for dilaton scattering in the low energy limit in fact respects this bound as well. This is presumably because the distinction between massless and massive scattering disappears in the high energy limit for relative entropy. Note that this is not the situation for other existing high energy bounds like the Froissart bound, whereas of now a rigorous bound, using axiomatic arguments, for massless scattering does not exist, neither can a massless limit be taken. We will also comment on what happens in the Regge limit. Here we will argue that since experiments disfavour a strictly linear Regge trajectory, the curvature of the trajectory has to be positive.

It will also turn out that close to the threshold, our relative entropy expressions following from eq.(1.2) will have simple expressions in terms of scattering lengths. In particular, we will find expressions in terms of ratios of D and S-wave scattering lengths. For instance, for $\pi^0\pi^0 \to \pi^0\pi^0$ we will find

$$D\left(\rho_A^{(1)}\|\rho_A^{(2)}\right) - \frac{(\Delta x)^2}{4\sigma} = 15(\Delta x)^2\left(\frac{2a_2^{(0)} + a_2^{(2)}}{2a_0^{(0)} + a_0^{(2)}}\right)(s-4m^2)^2, \qquad (1.4)$$

where $a_0^{(I)}, a_2^{(I)}$ denote the S and D-wave $I$'th isospin scattering lengths respectively. Now the numerator comprising of the combination of the D-wave scattering lengths can be shown to be positive which arises from the Froissart-Gribov representation for $\ell \geq 2$ scattering lengths. If one makes certain extra assumptions about the S-wave and P-wave scattering lengths (which follow from a Lagrangian formulation), then one finds that the universal piece above is positive! Similar arguments show that for $\pi^+\pi^- \to \pi^+\pi^-$ the above quantity is also positive while for $\pi^+\pi^+ \to \pi^+\pi^+$ and $\pi^+\pi^0 \to \pi^+\pi^0$, it is negative. Relative entropy near the threshold then becomes a potential tool to constrain various putative physical theories for scattering.

We will study relative entropy (as well as Rényi divergences where possible) for a variety of theories including $\lambda\phi^4$ at 1-loop, chiral perturbation theory ($\chi PT$), dilaton scattering in tree-level type II superstring theory as well as the S-matrix bootstrap constraining pion scattering.

Relative entropy expressions in various interesting limits like threshold expansion, high energy limit, are worked out. We then use relative entropy considerations to study S-matrix bootstrap constraining pion scattering.

Before the advent of QCD in the 1970s, the S-matrix bootstrap was advocated as a technique to study hadron interactions. Some profound results, like the Froissart bound [8], were obtained in this pursuit. A similar goal was also pursued a while for conformal field theories. Then with the advent of QCD and the renormalization group, the bootstrap approach was essentially abandoned. In 2008, a fresh numerical approach to study the CFT bootstrap was put forward in [11]. Using this, several new results were obtained for higher dimensional CFTs (see [12] for a recent review); a typical feature in the numerical plots, showing allowed physical theories, was that physically interesting theories like the 3d Ising model lived at a "kink". The numerical techniques make clever use of semi-definite programming algorithm (called Semi-Definite Programming for Bootstrap or SDPB [13]). Recently starting with [14] and [15], SDPB was put to good use to reboot the S-matrix bootstrap. This was further developed in [16]. In [17], this new approach to the S-matrix bootstrap was used to constrain pion scattering. Using phenomenological inputs such as the $\rho$-resonance mass and S-wave scattering lengths, two interesting constrained regions (see fig.(1)) on the plane of the Adler zeros[1] [18] $(s_0, s_2)$ were found.

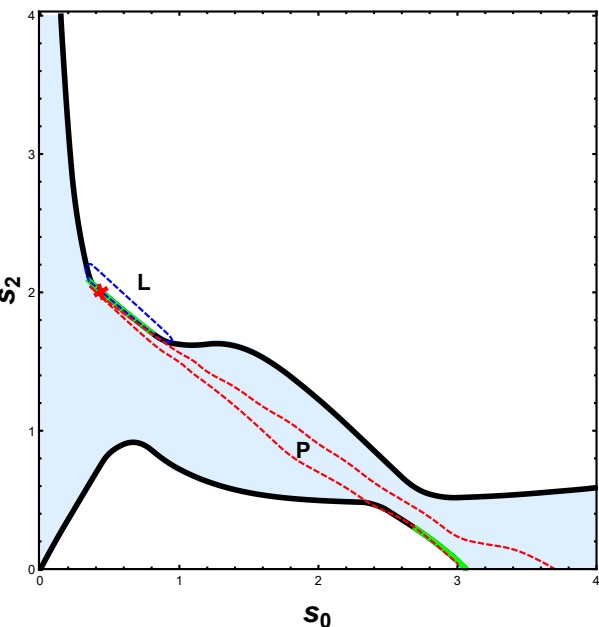

Figure 1: The "River". We get the river by imposing only the $\rho$ resonance and the inequalities mentioned in the main text. The "Lake" and "Peninsula" in [17] are indicated. The green regions are closest, in the sense of hypothesis testing, to the 1-loop $\chi PT$ indicated by the red cross and turn out to have comparable scattering lengths. The white region is excluded. The 1-loop point is close to the "kink" in the boundary.

The first region, dubbed as the "Lake" was found by imposing the $\rho$-resonance at the phenomenological value. This region eliminated possible theories while leaving behind a huge region of potentially allowed models. In fig.(1), the ruled out region using such considerations is indicated by L. The lower boundary of this region was found to allow for S-wave and P-wave scattering lengths compatible with experimental results while the upper boundary had opposite signs. The second region, dubbed as the "Peninsula" was obtained by imposing in addition, the experimentally observed S-wave scattering lengths within errors. This region

---

[1]Adler zeros in 2-2 scattering are $s$ values where the amplitude vanishes when a soft mode is involved.

was substantially smaller, and the standard model was observed to lie on the tip. In fig.(1) this is indicated by P. While being substantially smaller, by construction, this still leaves behind a huge set of potentially interesting S-matrices. This begs the question: Can we distinguish these S-matrices, all of which lead to similar scattering lengths? At leading order in sophistication for instance, which of these S-matrices is closest to 1-loop $\chi PT$–can we use hypothesis testing to answer this? Furthermore, it also raises the question: Can we reduce the set of S-matrices by not imposing the experimental scattering lengths?

In this paper, we will study relative entropy along the lake and use the definite signs in various channels found using the threshold expansion arguments to constrain these allowed regions. In particular, one of our goals will be to ask if a smaller region like the so-called peninsula can be obtained *without* putting in the phenomenological values of the S- and P-wave scattering lengths. Remarkably, the answer will turn out to be yes. The region we find is indicated in fig.(1)–we call this the "River" since the figure is very suggestive of one with two banks! We find this region by imposing the $\rho$-resonance[2] and the inequalities suggested by $\chi PT$ and the D-wave dispersion relations—equivalently the definite sign conditions on relative entropy referred to above. Intriguingly, the 1-loop $\chi PT$ point is close to a kink-type feature in the plot. Furthermore, in fig.(1) we have performed hypothesis testing in the low energy physical region, by comparing the theories living on the new boundary with the 1-loop $\chi PT$ (indicated by a red cross in fig.(1)). The theories closest, in this sense, to the 1-loop $\chi PT$ are indicated in green and live on opposite banks. Somewhat surprisingly, there is a region on the other bank which is far (in the sense of the $(s_0, s_2)$ values) from the perturbative 1-loop region, which gives rise to scattering lengths comparable to experiments. Put differently, this second region admits a set of reduced density matrices which close to the threshold cannot be distinguished from the other green region close to the 1-loop $\chi PT$ point, and hence exhibits comparable entanglement. We make some preliminary studies of this region in our paper but will leave a detailed analysis for future work. A preliminary analysis of elastic unitarity, for instance, seems to disfavour this second region, favouring the theories living on the upper bank instead. We also present our initial findings of a somewhat more constrained "river" using ideas pertaining to positivity [22]. *Our findings suggest that combining quantum information ideas with the bootstrap may be a worthwhile direction to pursue in the future.*

The paper is organized as follows. In Section (2), we set up the formulation of density matrices in 2-2 scattering. In Section (3), we consider measures of entanglement focusing on relative entropy. In Section (4), we turn to consider specific theories such as type II superstring theory, $\lambda \phi^4$ theory, and chiral perturbation theory. In Section (5), we turn to generalities focusing on relative entropy. In particular, we derive high energy bounds as well as give simple expressions for the threshold expansion for pion scattering. In Section (6), we try to use relative entropy to distinguish between amplitudes coming from two different theories (either differing in some underlying parameters or completely different theories) describing the same scattering process. In Section (7), we turn to exploring the S-matrix bootstrap using relative entropy considerations. In Section (8), we explore quantum information measures such as quantum relative entropy and *entanglement power* in connection with entanglement in isospin, which will be manifestly finite. We conclude with future directions in Section (9). Furthermore, the algebraic details skipped in the main text are given in details in several appendices.

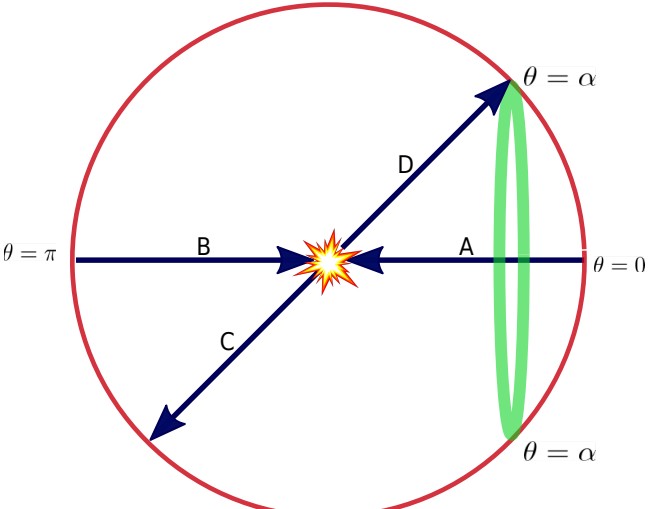

Figure 2: 2-2 scattering configuration in the centre of mass frame, with a Gaussian detector being placed at a point along the ring.

## 2 Density matrices in $2 \rightarrow 2$ scattering

We will consider quantum entanglement that is generated in the 2-2 scattering of relativistic particles $A + B \rightarrow C + D$ as shown in fig. 2. All of our analysis will be in the Center of Mass (CoM) frame. The spatial momenta of the outgoing particles are given by

$$\vec{p} \equiv (p, \theta_C, \varphi_C), \quad \vec{q} \equiv (q, \theta_D, \varphi_D), \tag{2.1}$$

with $p = |\vec{p}|$, $q = |\vec{q}|$ and the angle $\theta$ ranges over $[0, \pi]$ and $\varphi$ ranges over $[0, 2\pi)$. In the CoM frame, we will necessarily have $\vec{p} + \vec{q} = 0$ and thus

$$p = q, \quad \theta_C + \theta_D = \pi. \tag{2.2}$$

This was considered previously in [3–7] but we will differ from this analysis in certain crucial aspects. The main aspect is that unlike [3–7] we will focus on the $B$-particles at a fixed angle– for instance consider a finite resolution detector at a fixed angle $\theta_D$. Say, we have put such a detector at an angle $\theta_D = \alpha$, whose explicit measure we will specify later. Now what we mean is that, the angle of the particle will be measured within a range of $\Delta\alpha$ centred at $\alpha$, i.e., the particle will be detected if its angle lies in the interval $\left[\alpha - \frac{\Delta\alpha}{2}, \alpha + \frac{\Delta\alpha}{2}\right]$; we will generally assume $\Delta\alpha \ll 1$. In the above figure, the green ring corresponds to this angular spread of $\Delta\alpha$.

### 2.1 Density Matrix of the joint system $AB$

We will begin by considering the elastic case $A + B \rightarrow A + B$ and will compute the entanglement between the outgoing particles. For our conventions regarding scattering matrix and momentum states, we refer to Appendix (A).

First, we consider two unentangled non-identical particles, $A$ and $B$, with masses $m_A$ and $m_B$ respectively as the incoming state. Let $\vec{k}$ and $-\vec{k}$ be the respectively incident momenta in the CoM frame. So the initial state is chosen to be

$$|i\rangle := |\vec{k}\rangle\rangle := |\vec{k}_A, -\vec{k}_B\rangle. \tag{2.3}$$

---

[2]To constrain theories in the conformal bootstrap, assumptions of this kind are also made. For instance to get the famous epsilon expansion to work, one assumes the existence of a conserved stress tensor [19–21].

This state is the 2–particle Fock state as defined in eq.(A.14). The initial state is a separable one by construction. Clearly, the entanglement entropy of the initial state vanishes. Now, the state after scattering is given by, $\mathcal{S}|\vec{k}\rangle\!\rangle$, where $\mathcal{S}$ is the S-matrix. Next, we need to introduce a projector $\mathcal{Q}$ which restricts the angle $\theta$. The projector is given by

$$\mathcal{Q}_{AB}^{(F)} := \int d\Pi_{A\vec{p}} d\Pi_{B\vec{q}} \, F(\theta_B) \, |\vec{p},\vec{q}\rangle\,\langle\vec{p},\vec{q}|, \qquad d\Pi_{i\vec{k}} := \frac{d^3\vec{p}}{2E_{\vec{k}_i}}, \tag{2.4}$$

where we have introduced a finite support function $F$ to account for the particle detection as described above. Here, $|\vec{p},\vec{q}\rangle$ is the short form of the 2–particle Fock state $|\vec{p}_A,\vec{q}_B\rangle$ defined in eq.(A.14). We will use this short-hand notation from now on.

If we are considering the particle detector to be at angle $\theta_B = \alpha$ with a finite angular resolution $\Delta\alpha$ with $\Delta\alpha \ll 1$ then, the corresponding $F$ needs to be negligible, ideally vanishing, outside the interval $\left[\alpha - \frac{\Delta\alpha}{2}, \alpha + \frac{\Delta\alpha}{2}\right]$. Then, we have the target final state as

$$|f\rangle = \mathcal{Q}_{AB}^{(F)} \mathcal{S}|\vec{k}\rangle\!\rangle = \int d\Pi_{A\vec{p}} d\Pi_{B\vec{q}} \, F(\theta_B) \, |\vec{p},\vec{q}\rangle\,\langle\vec{p},\vec{q}|\,\mathcal{S}|\vec{k}\rangle\!\rangle. \tag{2.5}$$

Now, this state is not automatically normalized. The norm of the state is given by

$$\langle f\,|f\rangle = \int d\Pi_{A\vec{p}} d\Pi_{B\vec{q}} d\Pi_{A\vec{P}} d\Pi_{B\vec{Q}} \, F(\theta_{B\vec{p}}) F(\theta_{B\vec{P}}) \, \langle \vec{P},\vec{Q}\,|\,\vec{p},\vec{q}\rangle\,\langle\vec{p},\vec{q}|\,\mathcal{S}|\vec{k}\rangle\!\rangle\,\langle\!\langle\vec{k}|\,\mathcal{S}^\dagger\,|\vec{P},\vec{Q}\rangle$$

$$= \int d\Pi_{A\vec{p}} d\Pi_{B\vec{q}} \, F(\theta_{B\vec{p}})^2 \, \left|\langle\vec{p},\vec{q}|\,\mathcal{S}|\vec{k}\rangle\!\rangle\right|^2.$$

Now, in the CoM frame, momentum conservation gives us:

$$\langle\vec{p},\vec{q}|\mathcal{S}|\vec{k}\rangle\!\rangle = \delta^{(3+1)}\left(\sum_i p_i\right)\langle\vec{p},\vec{q}|s|\vec{k}\rangle\!\rangle = \delta\left(E_{A\vec{p}} + E_{B\vec{q}} - E_{A\vec{k}} - E_{B\vec{k}}\right)\delta^{(3)}(\vec{p}+\vec{q})\,\langle\vec{p},\vec{q}|s|\vec{k}\rangle\!\rangle. \tag{2.6}$$

Now, the simplest way to handle the square of a delta function is to introduce a $\delta(0)$ which can be understood from the identity

$$\int d^3\vec{x}\,\left(\delta^{(3)}(\vec{x}-\vec{y})\right)^2 f(\vec{x}) = \delta^{(3)}(0)\,f(\vec{y}). \tag{2.7}$$

Substituting this back into eq.(2.6) and performing one of the integrals leads to

$$\mathcal{N} \equiv \langle f\,|f\rangle = \frac{\delta^{(3+1)}(0)}{4k(E_{A\vec{k}} + E_{B\vec{k}})}\int d^3\vec{p}\,\delta(p-k)\,F(\pi-\theta_A)^2\,|\langle\!\langle\vec{p}|s|\vec{k}\rangle\!\rangle|^2, \tag{2.8}$$

where, we have expressed the delta function as:

$$\delta\left(E_{A\vec{p}} + E_{B\vec{p}} - E_{A\vec{k}} - E_{B\vec{k}}\right) = \frac{E_{A\vec{p}} E_{B\vec{p}}}{k(E_{A\vec{k}} + E_{B\vec{k}})}\delta(p-k).$$

Thus, the density matrix of the joint system in the state $|f\rangle$ is given by,

$$\rho^{(F)} := \frac{1}{\mathcal{N}}|f\rangle\langle f| \tag{2.9}$$

$$= \frac{1}{\mathcal{N}}\int d\Pi_{A\vec{p}} d\Pi_{B\vec{q}} d\Pi_{A\vec{P}} d\Pi_{B\vec{Q}} \, F(\theta_{B\vec{q}}) F(\theta_{B\vec{Q}}) \, \langle\vec{p},\vec{q}|\,\mathcal{S}|\vec{k}\rangle\!\rangle\,\langle\!\langle\vec{k}|\,\mathcal{S}^\dagger\,|\vec{P},\vec{Q}\rangle\,|\vec{p},\vec{q}\rangle\,\langle\vec{P},\vec{Q}|, \tag{2.10}$$

where the generalization to any other space-time dimensions is obvious in the powers of the delta functions and the Lorentz invariant phase space integral.

## 2.2 Reduced Density Matrices

Now, we will work out various reduced density matrices starting from the density matrix $\rho^{(F)}$ in 3+1 dimensions. First, we construct the reduced density matrices by tracing out subsystems. For most of our purpose, we will consider the reduced density matrix by tracing out $B$ particles, $\rho_A^{(F)} = \text{tr}_B \rho^{(F)}$.

$$\rho_A^{(F)} = \frac{1}{\delta^{(3)}(0)} \frac{\int d\Pi_{A\vec{p}} \, \delta(p-k) \, F(\pi - \theta_A)^2 \, |\langle\!\langle \vec{p}|\mathbf{s}|\vec{k}\rangle\!\rangle|^2 \, |\vec{p}\rangle \langle\vec{p}|}{\int d^3\vec{p} \, \delta(p-k) \, F(\pi - \theta_A)^2 \, |\langle\!\langle \vec{p}|\mathbf{s}|\vec{k}\rangle\!\rangle|^2} \, . \tag{2.11}$$

Here in taking the partial trace, we have exploited tensor product structure of the 2−particle Fock states as explained in Appendix (A.2). Next we will consider $\left(\rho_A^{(F)}\right)^n$ for $n \in \mathbb{Z}^+$. Trace of this quantity will play the central role in our analysis of various measures of entanglement. First we note that,

$$\left(\rho_A^{(F)}\right)^n = \left[\frac{1}{\delta^{(3)}(0)}\right]^n \frac{\int d\Pi_{A\vec{p}} \left[\delta(p-k) \, F(\pi - \theta_A)^2 \, |\langle\!\langle \vec{p}|\mathbf{s}|\vec{k}\rangle\!\rangle|^2\right]^n \, |\vec{p}\rangle \langle\vec{p}|}{\left[\int d^3\vec{p} \, \delta(p-k) \, F(\pi - \theta_A)^2 \, |\langle\!\langle \vec{p}|\mathbf{s}|\vec{k}\rangle\!\rangle|^2\right]^n} \, . \tag{2.12}$$

From this it readily follows that,

$$\text{tr}_A \left(\rho_A^{(F)}\right)^n \equiv \text{tr}_A \left(\rho_A^{(g)}\right)^n = \left[\frac{\delta(0)}{2\pi k^2 \delta^{(3)}(0)}\right]^{n-1} \int_{-1}^{1} dx \left[\mathcal{P}_{(g)}(x)\right]^n \, , \tag{2.13}$$

with,

$$\mathcal{P}_g(x) = \frac{g(x) \, |\langle\!\langle \vec{p}|\mathbf{s}|\vec{k}\rangle\!\rangle|^2}{\int_{-1}^{1} dx \, g(x) \, |\langle\!\langle \vec{p}|\mathbf{s}|\vec{k}\rangle\!\rangle|^2} \, , \tag{2.14}$$

where, we have assumed that $F(\alpha) \equiv F(\cos\alpha)$, defined $x := \cos\theta_A$ and defined $g(x) := F(-x)$[23]. Note that, $\mathcal{P}_g$ satisfies

$$\int_{-1}^{1} dx \, \mathcal{P}_g(x) = 1 \, . \tag{2.15}$$

## 2.3 Generalizations

So far we have considered the scattering event $A + B \rightarrow A + B$ with $A$ and $B$ non-identical particles. This analysis has an obvious generalization to identical particles as well as more general scattering event $A + B \rightarrow C + D$ where, now $A, B$ and $C, D$ can be identical particles. Furthermore, one can consider scattering of particles with global symmetry indices like isospin in the case of pions. The generalization is quite straightforward, and therefore we won't repeat the analysis in details. We will just spell out the main steps, reserving the full details for Appendix (C). We will be focusing on $3 + 1$ dimensions only, keeping in mind that it can be generalized to other dimensions trivially.

**Density Matrices**

We will start with the generalization where the final state of two scalar particles can be identical as well as different from incoming particles. Schematically, we can consider the generalized scattering event

$$A + B \rightarrow C + D \, , \tag{2.16}$$

---

[3]Note that, the functions $F(x)$ and $g(x)$ are *not* same mathematically. We want to clarify this in order to avoid any potential confusion. However, one may consider them to be *equivalent* in reference to the physical effect they are used to describe!

where now it can be that $A$ and $B$ are identical particles and so can $C$ and $D$[4]. To account for this case, we will consider the generic two-particle state $|\vec{p}, a; \vec{q}; b\rangle$ where, the labels/group-indices $a, b$ now encapsulates all the possibilities charted above such that each particle gets a label/group index corresponding to it ($A$ has label/group index $a$ and so on). This is again a 2–particle bosonic Fock state i.e., $|\vec{p}, a; \vec{q}; b\rangle = \mathrm{a}_a^\dagger(\vec{p})\mathrm{a}_b^\dagger(\vec{q})|\Omega\rangle$, $|\Omega\rangle$ being the bosonic Fock vacuum and $\mathrm{a}_i$ being the annihilation operator of the particle corresponding to the $i^{th}$ label. If we consider the scattering in eq.(2.16) in the CoM frame, the density matrix corresponding to the final state configuration is given by

$$\rho_{CD}^{(F)} \equiv \frac{1}{\mathcal{N}_{CD}} |f_{CD}\rangle \langle f_{CD}|, \tag{2.17}$$

with

$$|f_{CD}\rangle = \int d\Pi_{\vec{q}_1}^c \, d\Pi_{\vec{q}_2}^d \, F(\theta_{1\,\vec{q}_1}) \, |\vec{q}_1, c; \vec{q}_2, d\rangle \langle \vec{q}_1, c; \vec{q}_2, d \,|\, \mathcal{S} \,|\, \vec{p}_1, a; \vec{p}_2, b\rangle, \tag{2.18}$$

where, now we have considered the initial state to be $|\vec{p}_1, a; \vec{p}_2, b\rangle$ and $\mathcal{N}_{CD} := \langle f_{CD} | f_{CD}\rangle$. Furthermore, we can trace out the $D$ particle states to obtain the reduced density matrix, $\rho_C = \mathrm{tr}_D\left(\rho_{CD}^{(F)}\right)$ as

$$\rho_C = \frac{\int d\Pi_{\vec{p}} \left[\delta_{cd}(F(\theta_p)^2 + 2F(\theta_p)F(\pi - \theta_p)) + F(\pi - \theta_p)^2\right] |\vec{p}, c\rangle\langle\vec{p}, c| \; \delta(p - k) \left|\langle\!\langle \vec{p}; c, d|\mathbf{s}|\vec{k}; a, b\rangle\!\rangle\right|^2}{\delta^{(3)}(0) \int d^3\vec{p} \; \delta(p - k) \left[\delta_{cd}(F(\theta_{1p})^2 + 2F(\theta_{1p})F(\pi - \theta_p)) + F(\pi - \theta_p)^2\right] \left|\langle\!\langle \vec{p}; c, d|\mathbf{s}|\vec{k}; a, b\rangle\!\rangle\right|^2}. \tag{2.19}$$

Here, we have introduced the notation $|\vec{k}; a, b\rangle\!\rangle := |\vec{k}, a; -\vec{k}, b\rangle$. Now we consider separately the cases:

1. Particles are identical ($\delta_{cd} = 1$) and $F(\theta) = F(\pi - \theta)$. The reduced density matrix $\rho_C$ becomes in this case

$$\rho_C = \frac{1}{\delta^{(3)}(0)} \frac{\int d\Pi_{\vec{p}} \, F(\pi - \theta_p)^2 \, |\vec{p}, c\rangle\langle\vec{p}, c| \; \delta(p - k) \left|\langle\!\langle \vec{p}; d, d|\mathbf{s}|\vec{k}; a, b\rangle\!\rangle\right|^2}{\int d^3\vec{p} \; \delta(p - k) \, F(\pi - \theta_p)^2 \left|\langle\!\langle \vec{p}; c, c|\mathbf{s}|\vec{k}; a, b\rangle\!\rangle\right|^2}, \tag{2.20}$$

which gives

$$\mathrm{tr}_C\left((\rho_C)^n\right) = \left[\frac{\delta(0)}{2\pi k^2 \delta^{(3)}(0)}\right]^{n-1} \int_{-1}^{1} dx \, [\mathcal{P}_g(x)]^n, \tag{2.21}$$

with

$$\mathcal{P}_g(x) := \frac{g(x) \, |\langle\!\langle \vec{p}; c, c|\mathbf{s}|\vec{k}; a, b\rangle\!\rangle|^2}{\int_{-1}^{1} dx \, g(x) \, |\langle\!\langle \vec{p}; c, c|\mathbf{s}|\vec{k}; a, b\rangle\!\rangle|^2}, \tag{2.22}$$

where, we have again defined $F(-x)^2 := g(x)$.

2. Now we consider the situation where the outgoing particles are identical and $F(\varphi_1)$ is not centered anyway about $\theta = \pi/2$. In this case, from eq.(2.19) we will encounter cross terms like $F(\theta_p)F(\pi - \theta_p)$. However, since the supports of $F(\theta_p)$ and $F(\pi - \theta_p)$ do not overlap significantly in this scenario (especially and quite accurately true for $2x = 2\cos(\theta_p) \gg N\sigma$ where $Erf(N\sigma)$ gives the desired accuracy). Hence, we can safely ignore these terms. Furthermore, using the same logic will also get rid of the cross terms coming from the binomial expansion present in the numerator. Then, we

---

[4]To simplify life, we will consider all masses to be equal, but this can be easily relaxed. See eq.(C.26) for a generalization to the case of all unequal masses.

see that the numerator and the denominator are both comprised of two integrals each, one with $F(\theta_p)$ and the other being exactly the same except with $F(\pi - \theta_p)$ respectively. Upon using polar co-ordinates and carrying out the azimuthal and radial integral using the delta function, we are only left with the $x = \cos(\theta_p)$ integral. Then, if we make the substitution $x \to -x$, the integral of the first part of the integrand (in both the numerator and the denominator) becomes the same as the second integral, since the amplitude must be symmetric in $t$ and $u$ in the identical case. Therefore, finally we get

$$tr_C\left((\rho_C)^n\right) = \left[\frac{\delta(0)}{4\pi k^2 \delta^{(3)}(0)}\right]^{n-1} \int_{-1}^{1} [\mathcal{P}_g(x)]^n, \tag{2.23}$$

where,

$$\mathcal{P}_g(x) := \frac{g(x)\,|\,\langle\!\langle \vec{p}; c, d|\mathbf{s}|\vec{k}; a, b\rangle\!\rangle|^2}{\int_{-1}^{1} dx\, g(x)\,|\,\langle\!\langle \vec{p}; c, d|\mathbf{s}|\vec{k}; a, b\rangle\!\rangle|^2}\,. \tag{2.24}$$

Note that the factor of 2 is different in this case than when $x = 0$ which will only be important when we consider the relative entropy between these 2 cases. Otherwise it makes no difference and will cancel as in previous calculations.

3. When the particles are non-identical $\delta_{cd} = 0$. In this case, eq.(2.13) holds good with the suitably generalized amplitude.

# 3 Measures of entanglement

In this section, we discuss various measures of entanglement such as entanglement entropy, quantum relative entropy [23] and quantum Rényi divergence[5] as given in [25] and quantum information variance as discussed in [26].

## 3.1 Entanglement Entropy

For a bipartite system comprised of two subsystems $A$ and $B$, the entanglement entropy is defined by the *von Neumann entropy of either of its reduced states*. That is, for a pure state of the joint system $A \cup B$ given by the density matrix $\rho_{AB} = |\psi\rangle\langle\psi|_{AB}$, it is given by:

$$S_{EE} := S_{\mathrm{vN}}(\rho_A) = -\mathrm{tr}\left[\rho_A \log \rho_A\right] = -\mathrm{tr}\left[\rho_B \log \rho_B\right] = S_{\mathrm{vN}}(\rho_B)\,, \tag{3.1}$$

where, $\rho_{A(B)}$ stands for the reduced density matrix obtained via partial tracing

$$\rho_{A(B)} = \mathrm{tr}_{B(A)}\rho_{AB}\,. \tag{3.2}$$

Here we will calculate entanglement entropy in some specific cases. For this we will consider a particular form for $g(x)$. For practical purposes, $S_{EE}$ can be calculated by the replica trick which is given by in our cases by

$$S_{EE}^{(g)} = -\lim_{n \to 1} \partial_n \mathrm{tr}_A\left(\rho_A^{(g)}\right)^n\,. \tag{3.3}$$

Then using eq.(2.13), we have

$$S_{EE}^{(g)} = -\ln\left(\frac{2\,\pi T}{k^2 V}\right) - \int_{-1}^{1} dx\, \mathcal{P}_g(x) \ln \mathcal{P}_g(x)\,, \tag{3.4}$$

---

[5]See [24] for a recent application in holography.

where $\mathcal{P}_g(x)$ is given in eq.(2.14) and we have written $2\pi\delta(0) = T$ and $(2\pi)^3\delta^{(3)}(0) = V$. We will consider a Gaussian profile for $g(x)$,

$$g(x) \equiv \delta_\sigma(x-y) = \frac{1}{2\sqrt{2}\sigma} e^{-\frac{(x-y)^2}{4\sigma}} . \tag{3.5}$$

Here we are considering $\sigma > 0$ but $\sigma \ll 1$ and $y \neq \pm 1$. In the sense of distributional limit, $\sigma \to 0$ corresponds to $g(x) \to \delta(x-y)$. However, note that we are emphasizing here the importance of taking Gaussian profile, *not* the Dirac delta function (note that, we have to work with $g(x)^n$). Now we exploit the fact that we have chosen $y \in (-1, 1)$. Then, in this situation we can use

$$\langle\!\langle \vec{p}|\mathbf{s}|\vec{k}\rangle\!\rangle \equiv \mathcal{M}(s,x) , \tag{3.6}$$

where $\mathcal{S} = 1 + i\mathcal{M}$. Then, we have for $\mathcal{P}_g(x)$,

$$\mathcal{P}_g(x) = \frac{g(x)|\mathcal{M}(s,x)|^2}{\int_{-1}^{1} dx\, g(x)|\mathcal{M}(s,x)|^2} . \tag{3.7}$$

Now, doing the integral over $\mathcal{P}_g(x)$ in eq.(3.4), one obtains

$$-\mathcal{I}_E = \ln(2\sqrt{\pi}\sigma) + \frac{1}{2} + \frac{1}{\sum_{i=0}^{\infty} \frac{\sigma^i}{i!} \frac{\partial^{2i}}{\partial x^{2i}}(|\mathcal{M}(s,x)|^2)\Big|_y} \left( \sum_{i=0}^{\infty} i \frac{\sigma^i}{i!} \frac{\partial^{2i}}{\partial x^{2i}}(|\mathcal{M}(s,x)|^2)\Big|_y \right) . \tag{3.8}$$

We provide the details of this calculation in Appendix (B.1). In the limit $\sigma \to 0$, this evaluates to

$$S_{EE} = \ln\left( \frac{\sqrt{\sigma}k^2 V}{\sqrt{\pi}T} \right) + \frac{1}{2} . \tag{3.9}$$

This implies that, for perfectly precise detectors ($\sigma = 0$), the absolute angular entanglement is basically a constant independent of the amplitude. The sub-leading terms in $\sigma$ will of course depend on the amplitude. Note that we didn't put any restriction on $\phi$, thus our detector uniformly detects along $\phi$ (which also explains the ring structure of the Gaussian detector). Now, the scattering amplitude itself doesn't depend on $\phi$ and hence in our target final state $|f\rangle$, we will have uniform contribution of 2-particle states along $\phi$, for a fixed $\theta$. Thus, our 2-particle states will be maximally entangled in $\phi$. Hence, in the above expression, even if we assumed an ideal detector, it is not a pure state as the expression contains the maximal entropy contribution from $\phi$.

## 3.2 Quantum Relative Entropy

If $\rho_1$ and $\rho_2$ be two density matrices then the entropy of $\rho_1$ relative to $\rho_2$ is given by

$$D(\rho_1\|\rho_2) = \mathrm{tr}\rho_1(\ln\rho_1 - \ln\rho_2) . \tag{3.10}$$

Quantum relative entropy acts as a measure of distinguishability of two states. Now, we will do a relative entropy calculation with the configuration given in fig.(3), where there are two detectors placed at two different angles. Thus, we need to consider two support functions centred on two different angles. One can take the angular spreads for the two functions to be same or different. We will consider the simpler case of the two detectors having the same Gaussian width $\sigma$. We have two density matrices $\rho_A^{(1)}$ and $\rho_A^{(2)}$ corresponding to two detectors at angles $\theta_{B1}$ and $\theta_{B2}$ respectively. Furthermore, we will assume that $\theta_{B1}$ and $\theta_{B2}$ differ only slightly to the extent that their difference is more than the angular sensitivity of the detectors

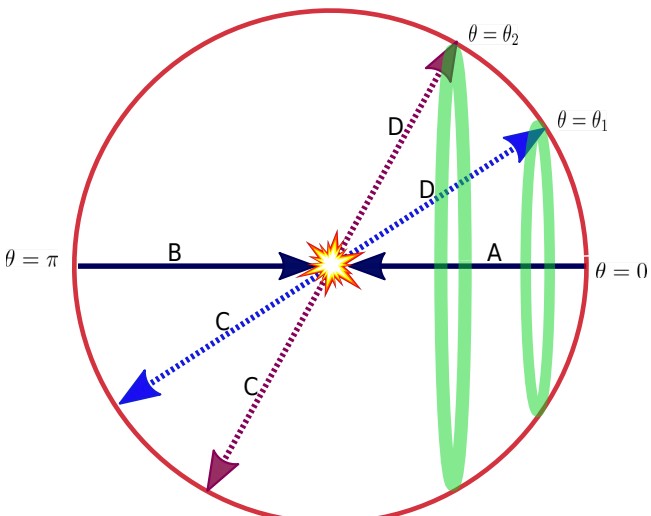

Figure 3: Two different configurations of Gaussian Detectors for 2 to 2 scattering.

but small compared to the magnitude of the angles. Thus, we have the reduced density matrix as

$$\rho_A^{(i)} = \frac{(2\pi)^3}{V} \frac{\int d\Pi_{A\vec{p}}\, \delta(p-k)\, F_i(\pi-\theta_A)^2\, |\langle\!\langle \vec{p}|\mathbf{s}|\vec{k}\rangle\!\rangle|^2\, |\vec{p}\rangle\langle\vec{p}|}{\int d^3\vec{p}\, \delta(p-k)\, F_i(\pi-\theta_A)^2\, |\langle\!\langle \vec{p}|\mathbf{s}|\vec{k}\rangle\!\rangle|^2}, \tag{3.11}$$

for $i = 1, 2$. Now, we employ the replica trick to calculate the relative entropy such that

$$D(\rho_1\|\rho_2) = \lim_{n\to 1} \frac{\partial}{\partial n}\left[\operatorname{tr}\rho_1^n - \operatorname{tr}\rho_1\rho_2^{n-1}\right]. \tag{3.12}$$

This eventually gives

$$D\left(\rho_A^{(1)}\|\rho_A^{(2)}\right) = \int_{-1}^{1} dx\, \mathcal{P}_{g_1}(x) \ln\left(\frac{\mathcal{P}_{g_1}(x)}{\mathcal{P}_{g_2}(x)}\right). \tag{3.13}$$

Again if we consider that $\theta_{B1}$ and $\theta_{B2}$ both are way off from the forward direction, we can use eq.(3.6).

Now relative entropy is known to be positive, whose usual proofs in quantum mechanics follow using properties of probability distributions. $\mathcal{P}_g(x)$ *above is not a probability distribution since it is not less than unity; rather it is a density such that* $\int_{-1}^{1} dx\, \mathcal{P}_g(x) = 1$. Nonetheless, using some simple tricks as shown below we can still demonstrate positivity. Consider the steps:

$$\begin{aligned}
D\left(\rho_A^{(1)}\|\rho_A^{(2)}\right) &= \int_{-1}^{1} dx\, \mathcal{P}_{g_1}(x) \ln\left(\frac{\mathcal{P}_{g_1}(x)}{\mathcal{P}_{g_2}(x)}\right) = -\int_{-1}^{1} dx\, \mathcal{P}_{g_1}(x) \ln\left(\frac{\mathcal{P}_{g_2}(x)}{\mathcal{P}_{g_1}(x)}\right) \\
&\geq \int_{-1}^{1} dx\, \mathcal{P}_{g_1}(x)\left(1 - \frac{\mathcal{P}_{g_2}(x)}{\mathcal{P}_{g_1}(x)}\right) = 0,
\end{aligned} \tag{3.14}$$

where we have used $\mathcal{P}_{g_i}(x) \geq 0$, eq.(2.15) and in the second step, we have used the identity $\ln(x) \leq (x-1)\,\forall x > 0 \implies -\ln(x) \geq (1-x)\,\forall x > 0$. Let us now move to the Gaussian detector case. We have now that

$$g_i(x) = \delta_\sigma(x - x_i), \tag{3.15}$$

where $\delta_\sigma(x - y)$ is given in eq.(3.5). This leads to

$$D\left(\rho_A^{(1)} \| \rho_A^{(2)}\right) \approx \ln\left(\frac{\mathcal{I}_g(s, x_2)}{\mathcal{I}_g(s, x_1)}\right) + (\Delta x)\frac{\partial}{\partial x}\left(\ln\left(\frac{\mathcal{I}_g(s, x)}{\mathcal{I}_g(s, x_1)}\right)\right)\bigg|_{x_1} + \frac{(\Delta x)^2}{4\sigma}, \tag{3.16}$$

where

$$\begin{aligned}
\mathcal{I}_g(s, x_0) &= \int_{-1}^1 dx \left(\frac{1}{2\sqrt{\pi\sigma}}e^{-\frac{(x-x_0)^2}{4\sigma}}\right)|\mathcal{M}(s, x)|^2 \\
&= \sum_{i=0}^{\infty} \frac{\partial^{2i}}{\partial x^{2i}}(|\mathcal{M}(s, x)|^2)\bigg|_{x_0} \frac{\sigma^i}{i!} = |\mathcal{M}(s, x_0)|^2 + O(\sigma).
\end{aligned} \tag{3.17}$$

In the limit of small $\Delta x := x_1 - x_2$ and $\sigma \to 0$, one obtains[6]

$$\begin{aligned}
D\left(\rho_A^{(1)} \| \rho_A^{(2)}\right) &\approx \frac{(\Delta x)^2}{4\sigma} + \frac{(\Delta x)^2}{2}\left(\frac{\partial^2}{\partial x^2}\left(\frac{|\mathcal{M}(s, x)|^2}{|\mathcal{M}(s, x_1)|^2}\right)\bigg|_{x_1} - \left(\frac{\partial}{\partial x}\left(\frac{|\mathcal{M}(s, x)|^2}{|\mathcal{M}(s, x_1)|^2}\right)\bigg|_{x_1}\right)^2\right) \\
&\quad + O((\Delta x)^2\sigma).
\end{aligned} \tag{3.18}$$

Let us understand what the above approximation exactly is. In the expression above, the terms which have been thrown away are at least of the order $(\Delta x)^2\sigma$. However, physically it only makes sense to accurately measure the entanglement between states which are separated more than the resolution of the detector which is $\sigma$. Hence, physically we must have $\sigma \ll (\Delta x)$. Therefore, we can safely say that $O((\Delta x)^2\sigma) \ll O((\Delta x)^3) \implies O((\Delta x)^2\sigma) \sim O((\Delta x)^3, \sigma)$ i.e., the terms are either higher order than $(\Delta x)^2$ or higher order than $\sigma^0$.

Furthermore, the first term given by $(\Delta x)^2/(4\sigma)$ can be identified with just the hard sphere scattering result where there is no angular dependence in the scattering. Henceforth, we will separate out this piece and define

$$D_Q\left(\rho_A^{(1)} \| \rho_A^{(2)}\right) = \frac{(\Delta x)^2}{2}\left[\frac{\partial^2}{\partial x^2}(\ln \mathcal{M}_1(s, x))\right]_{x=x_1}, \tag{3.19}$$

where, we have defined for later convenience,

$$\mathcal{M}_1(s, x) := \frac{|\mathcal{M}(s, x)|^2}{|\mathcal{M}(s, x_1)|^2}. \tag{3.20}$$

Note that, $D_Q$ is the $\sigma^0$ term in the $\sigma \to 0$ expansion of the relative entropy $D$, eq.(3.18). Thus, $D_Q$ is universal and does *not* depend upon the human chosen parameter $\sigma$.

**Validity of the expansion**

The above expansion is quite generally valid even in the neighbourhood of s-channel poles. First note that we are in the s-channel physical region so that $t < 0$ and hence we do not have to worry about $t$-channel poles. Furthermore, if we consider an $s$-channel pole of the form $\rho(t)/(s - s_0)$ and plug this form into eq.(3.16), it can be easily verified that no infinity is encountered. These observations are verified by the behaviour found in the string theory example considered below.

---

[6]We present the details of the analysis in Appendix (B.2). Note that, the term linear in $\Delta x$ gets canceled after expanding the log term in eq.(3.16)

**Regarding Positivity of $D_Q\left(\rho_A^{(1)}\|\rho_A^{(2)}\right)$**

We would like to point out that $D_Q$ is NOT positive automatically even though relative entropy is, which has been established in eq.(3.14). This is because of the fact that, $D_Q$ differs from the relative entropy by $(\Delta x)^2/4\sigma$ and $O(\sigma)$ terms. In the limit $\sigma \to 0$, former is the dominant term while $O(\sigma)$ terms vanish. Thus, irrespective of sign of $D_Q$, the relative entropy is bound to be positive.

## 3.3 Réyni divergences

Both the relative entropy and the entanglement entropy are actually a specific limit of a general concept known as the Réyni Divergence (see [24,25]). Réyni Divergence of order $n$ of a density matrix $\rho_1$ from another density matrix $\rho_2$ is defined by

$$D_n(\rho_1\|\rho_2) = \frac{1}{n-1} \ln\left[\mathrm{tr}\rho_1^n \rho_2^{1-n}\right],\tag{3.21}$$

for normalized density matrices $\rho_1$ and $\rho_2$. The Réyni divergence is defined for $0 < n < \infty$ and $n \neq 1$. We can define the Rényi divergence for the special values $n = 0, 1, \infty$ by taking a limit, and in particular the limit $n \to 1$ gives the quantum relative entropy of $\rho_1$ relative to $\rho_2$. We can reach the relative entropy from the Rényi divergence using the quantity

$$T_n(\rho_1\|\rho_2) = \mathrm{tr}\rho_1^n \rho_2^{1-n}.\tag{3.22}$$

Following steps similar to those used to reach the relative entropy from the Rényi divergence, it is straightforward to see that,

$$D(\rho_1\|\rho_2) = \lim_{n\to 1} \frac{\partial}{\partial n} T_n(\rho_1\|\rho_2).\tag{3.23}$$

In fact, this can be easily generalized to higher derivatives with respect to $n$ to get that

$$\lim_{n\to 1} \frac{\partial^i}{\partial n^i}(T_n(\rho_1\|\rho_2)) = \mathrm{tr}\rho_1(\ln(\rho_1) - \ln(\rho_2))^i.$$

Though working both with $D_n(\rho_1\|\rho_2)$ and $T_n(\rho_1\|\rho_2)$ are equivalent, we would prefer working with $T_n(\rho_1\|\rho_2)$ because taking its derivative is much easier than taking the limit of $D_n(\rho_1\|\rho_2)$ due to the pesky log present in the Réyni Divergence. Eq.(3.23) is precisely the replica trick formula for calculating the quantum relative entropy of a density matrix $\rho_1$ relative to $\rho_2$. Following a similar procedure as in before, it is also straightforward to show

$$T_n(\rho_1\|\rho_2) = \int_{-1}^{1} dx \, (\mathcal{P}_{g_1}(x))^n (\mathcal{P}_{g_2}(x))^{1-n}.\tag{3.24}$$

Specializing for the Gaussian Detectors has $T_n(\rho_1\|\rho_2)$ take the form

$$\begin{aligned}
T_n\left(\rho_A^{(1)}\|\rho_A^{(2)}\right) &= e^{-\frac{(\Delta x)^2}{4\sigma}n(1-n)}\left(\frac{\mathcal{I}_g(s,x_2)}{\mathcal{I}_g(s,x_1)}\right)^{n-1}\left(\frac{\mathcal{I}_g(s,x_n^{12})}{\mathcal{I}_g(s,x_1)}\right),\\
&\xrightarrow{\sigma\to 0} e^{-\frac{(\Delta x)^2}{4\sigma}n(1-n)}\left(\frac{|\mathcal{M}(s,x_2)|^2}{|\mathcal{M}(s,x_1)|^2}\right)^{n-1}\left(\frac{|\mathcal{M}(s,x_n^{12})|^2}{|\mathcal{M}(s,x_1)|^2}\right),
\end{aligned}\tag{3.25}$$

with $x_n^{12} = n\,x_1 + (1-n)x_2$. The derivation is similar to that of the entropy calculation and is done in its full glory in Appendix (B.3). We also have the equivalent expression for the Réyni Divergence as

$$\begin{aligned}
D_n\left(\rho_A^{(1)}\|\rho_A^{(2)}\right) &= n\frac{(\Delta x)^2}{4\sigma} + \ln\left(\frac{\mathcal{I}_g(s,x_2)}{\mathcal{I}_g(s,x_1)}\right) + \frac{1}{n-1}\ln\left(\frac{\mathcal{I}_g(s,x_n^{12})}{\mathcal{I}_g(s,x_1)}\right),\\
&\xrightarrow{\sigma\to 0} n\frac{(\Delta x)^2}{4\sigma} + \ln\left(\frac{|\mathcal{M}(s,x_2)|^2}{|\mathcal{M}(s,x_1)|^2}\right) + \frac{1}{n-1}\ln\left(\frac{|\mathcal{M}(s,x_n^{12})|^2}{|\mathcal{M}(s,x_1)|^2}\right).
\end{aligned}\tag{3.26}$$

Using this expression it is easy to see that this satisfies all the properties given in [24], at least in the limit of $\sigma \to 0$. Firstly, eq.(3.26) is continuous since composition of continuous functions (here being $x_n^{12}, \mathcal{I}_g, \mathcal{M}$ and log) is still continuous. Secondly, $\partial_n D_n(\rho_1 \| \rho_2) > 0$ since it is dominated by $(\Delta x)^2/4\sigma > 0$. By the same logic, positivity is also followed. Note that $\partial_n^2 D_n$ would get rid of the hard-sphere term. Lastly, $(1-n)D_n(\rho_1 \| \rho_2)$ is concave, which is again trivially seen in leading order. However, the case $n = 0$ is a quantity considered in the quantum information literature and is defined via a continuation in $n$. Taking the limit $n \to 0$ of eq.(3.24) gives us that

$$T_0(\rho_1 \| \rho_2) = \int_{-1}^{1} dx \, \mathcal{P}_{g_1}(x) = 1 \,, \tag{3.27}$$

leading us to

$$D_0(\rho_1 \| \rho_2) = \frac{1}{0-1} \ln\left(T_0(\rho_1 \| \rho_2)\right) = 0 \,. \tag{3.28}$$

This can also be seen from eq.(3.25) and eq.(3.26) when we remind ourselves that $x_n^{12} = x_2$ for $n = 0$.

## 3.4 Quantum Information Variance

We follow the definition of the variance in quantum information as given in [26],

$$
\begin{aligned}
V\left(\rho_A^{(1)} \| \rho_A^{(2)}\right) &= \operatorname{tr}\left(\rho_A^{(1)}\left(\ln \rho_A^{(1)} - \ln \rho_A^{(1)}\right)^2\right) - \left(D\left(\rho_A^{(1)} \| \rho_A^{(2)}\right)\right)^2 \\
&= \lim_{n \to 1}\left[\frac{\partial^2}{\partial n^2}\left(T_n\left(\rho_A^{(1)} \| \rho_A^{(2)}\right)\right) - \left(\frac{\partial}{\partial n}\left(T_n\left(\rho_A^{(1)} \| \rho_A^{(2)}\right)\right)\right)^2\right].
\end{aligned} \tag{3.29}
$$

We have derived the expression for it in Appendix (B.4) which we just quote here,

$$
\begin{aligned}
V\left(\rho_A^{(1)} \| \rho_A^{(2)}\right) &= \frac{(\Delta x)^2}{2\sigma} + (\Delta x)^2 \frac{\partial^2}{\partial x^2}\left(\ln\left(\frac{\mathcal{I}_g(s,x)}{\mathcal{I}_g(s,x_1)}\right)\right)\Bigg|_{x_1} \\
&\xrightarrow{\sigma \to 0} \frac{(\Delta x)^2}{2\sigma} + (\Delta x)^2 \frac{\partial^2}{\partial x^2}\left(\ln\left(\frac{|\mathcal{M}(s,x)|^2}{|\mathcal{M}(s,x_1)|^2}\right)\right)\Bigg|_{x_1}.
\end{aligned} \tag{3.30}
$$

We observe a very interesting fact in eq.(3.30). In the approximation that $(\Delta x)$ is small (can also take $\sigma \to 0$ or not), we have that

$$D\left(\rho_A^{(1)} \| \rho_A^{(2)}\right) \xrightarrow{\text{leading order in } (\Delta x)} \frac{1}{2} V\left(\rho_A^{(1)} \| \rho_A^{(2)}\right), \tag{3.31}$$

where the LHS of the equation is in leading order *w.r.t* $\Delta x$ while the RHS is exact.

## 3.5 Generalized case

### 3.5.1 Entanglement Entropy

Now, we turn to calculating the entanglement entropy for scattering in the setup as in Section (2.3). The replica trick generalizes quite trivially and we are left with the following expression for the entanglement entropy in the general case,

$$S_{EE} = -\ln\left[\frac{4\pi T}{Ak^2V}\right] - \int_{-1}^{1} dx \, \mathcal{P}_g(x) \ln \mathcal{P}_g(x), \tag{3.32}$$

where $A = 2$ for the non-identical and the zero mean identical case and $A = 4$ for non-zero mean identical case with $\mathcal{P}_g(x)$ defined in eq.(2.22). Again, as before, we will consider Gaussian detectors and therefore will be using eq.(3.5). Then, we repeat the steps and obtain

$$S_{EE} = \ln\left(\frac{A\sqrt{\sigma}k^2 V}{2\sqrt{\pi}T}\right) + \frac{1}{2} + \frac{1}{\left(\sum_{i=0}^{\infty} \frac{\sigma^i}{i!}\frac{\partial^{2i}}{\partial x^{2i}}(|\mathcal{M}_{a,b}^{c,d}(s,x)|^2)\Big|_y\right)}\left(\sum_{i=0}^{\infty} i\frac{\sigma^i}{i!}\frac{\partial^{2i}}{\partial x^{2i}}(|\mathcal{M}_{a,b}^{c,d}(s,x)|^2)\Big|_y\right). \tag{3.33}$$

The $\sigma \to 0$ conclusion holds in this case as well, i.e., the entanglement entropy goes to infinity. [3–7] considered certain regularizations to obtain the finite part. As we will see below, the dependence on such regularizations does not arise when we consider relative entropy.

### 3.5.2 Relative Entropy

Next we proceed to define relative entropy in this setting. We define $F_1(\theta)$ and $F_2(\theta)$ to be centered around $\theta^{B1}$ and $\theta^{B2}$. We shall construct the density matrix $\rho_C$ using $F_1(\theta)$ and $\tilde{\rho}_C$ using $F_2(\theta)$ such that $\rho_C$ is as in eq.(2.19) and $\tilde{\rho}_C$ is the same with $F_1(-x) \to F_2(-x)$. We shall again use the replica trick in order to evaluate the relative entropy. Now we will consider the identical and non-identical cases separately.There will be total four cases as follows:

1. First we consider when the outgoing particles are identical and either $x_1$ or $x_2$ is 0. As mentioned previously, there is a factor difference between the $x = 0$ and $x \neq 0$ cases when outgoing particles are identical. Going through the algebra, which can get messy, we find that the $x_1 = 0$ case is not salvageable at all since the divergence do not cancel. However, the $x_2 = 0$ case is somehow saved except for a constant shift of $-\ln(2)$, which can be safely ignored when compared to the divergence $(\Delta x)^2/4\sigma$ in our previously found expression.

2. When both particles are identical and $x_1, x_2 \neq 0$: Similarly as before, using the expressions for the density matrices and carrying out the polar and azimuthal integrals, we obtain that

$$tr\left(\rho_C(\tilde{\rho}_C)^{n-1}\right) =$$
$$\frac{1}{2}\left[\frac{\pi T}{k^2 V}\right]^{n-1}\frac{\int_{-1}^{1}dx(F_1(x)^2 + F_1(-x)^2)(F_2(x)^2 + F_2(-x)^2)^{(n-1)}|\langle\!\langle \vec{p};c,d|\mathbf{s}|\vec{k};a,b\rangle\!\rangle|^{2n}}{(\int_{-1}^{1}dxF_1(-x)^2|\langle\!\langle \vec{p};c,d|\mathbf{s}|\vec{k};a,b\rangle\!\rangle|^2)(\int_{-1}^{1}dxF_2(-x)^2|\langle\!\langle \vec{p};c,d|\mathbf{s}|\vec{k};a,b\rangle\!\rangle|^2)^{n-1}}. \tag{3.34}$$

Then, carrying out the binomial expansion of the term $(F_2(x)^2 + F_2(-x)^2)^{(n-1)}$, we can again do away with the cross terms. Hence, only the terms, $F_1(-x)^2 F_2(-x)^{2n-2} + F_1(-x)^2 F_2(x)^{2n-2} + F_1(x)^2 F_2(-x)^{2n-2} + F_1(x)^2 F_2(x)^{2n-2}$ will be left. Now we have two cases. When $x_1$ and $x_2$ have the same sign, the terms $F_1(-x)^2 F_2(-x)^{2n} + F_1(x)^2 F_2(x)^{2n-2}$ dominates, and when $x_1$ and $x_2$ have the opposite sign, the other two terms dominate. This is because $F_1(x)$ and $F_2(x)$ approximate delta functions of the form $\delta(x - x_1)$ and $\delta(x - x_2)$. Hence, when the signs of $x_1$ and $x_2$ are the same, the Gaussians approximating the deltas, $\delta(x - x_1)$ and $\delta(x - x_2)$ are closer and when the signs are different, the deltas, $\delta(x - x_1)$ and $\delta(x + x_2)$ are closer. Furthermore, the individual integrals involving the two remaining terms in both cases can be shown to be the same by substituting $x \to -x$. Hence, we can express this behaviour as

$$tr(\rho_C(\tilde{\rho}_C)^{n-1}) = \left[\frac{\pi T}{k^2 V}\right]^{n-1}\int_{-1}^{1}dx\mathcal{P}_{g_1}(x)(\mathcal{P}_{g_2}(sgn(x_1 x_2)x))^{(n-1)}, \tag{3.35}$$

where $sgn(z) = 1, z > 0$ and $sgn(z) = -1, z < 0$. Using the expression of $tr((\rho_C)^n)$, the partial derivative after taking the limit $n \to 1$ gives us the expression for the relative entropy as

$$D(\rho_C \| \tilde{\rho}_C) = \int_{-1}^{1} dx \mathcal{P}_{g_1}(x) \ln \left( \frac{\mathcal{P}_{g_1}(x)}{\mathcal{P}_{g_2}(sgn(x_1 x_2) x)} \right). \tag{3.36}$$

3. The case of non-identical particle is the one already treated in details previously.

As mentioned previously, all the cases *w.r.t* signs of $x_1$ and $x_2$ for the identical particles in the final state (eq.(3.36)), along with the case of $x_2 = 0$ can be combined into the following single expression (with slight modification to existing definitions and remembering that $sgn(0) = 1$) and we get the equivalent of eq.(3.16) as

$$D(\rho_C(x_1) \| \rho_C(x_2)) \approx \ln \left( \frac{\mathcal{I}_g(s, |x_2|)}{\mathcal{I}_g(s, |x_1|)} \right) + (\Delta x) \frac{\partial}{\partial x} \left( \ln \left( \frac{\mathcal{I}_g(s, x)}{\mathcal{I}_g(s, |x_1|)} \right) \right) \Bigg|_{|x_1|} + \frac{(\Delta x)^2}{4\sigma}, \tag{3.37}$$

with

$$\Delta x := |x_1| - |x_2|. \tag{3.38}$$

Keep in mind that $\mathcal{I}_g(s, x)$ is an even function of $x$ when the amplitude has symmetry in $t \leftrightarrow u$ which it should in the identical outgoing particle case. This is so as $\mathcal{I}_g(s, x)$ consists of the even derivatives of the amplitude and even derivatives of an even function are still even. The derivative in the second term in eq.(3.37) is an odd function, however its sign is absorbed by the modified $\Delta x$ as explained in Appendix (C.1).

Furthermore, it is obvious than in the approximation as in eq.(3.16), the only change would be $x_1 \to |x_1|$ and $\Delta x \to |x_1| - |x_2|$.

# 4 Known theories

In this section, we will consider the behaviour of relative entropy and entanglement entropy in certain known theories which include $\phi^4$ theory, chiral perturbation theory ($\chi PT$) and type II string theory.

## 4.1 $\phi^4$ theory

We begin by considering the $\lambda \phi^4$ amplitude up-to 1-loop with the amplitude given by

$$\mathcal{M}(s, t, u) = \lambda - \frac{\lambda^2}{32\pi^2} \left( 2 + G(s) + G(t) + G(u) \right)$$

$$\text{where,} \quad G(x) = -2 + \sqrt{\left( 1 - \frac{4}{x} \right)} \ln \left( \frac{\sqrt{\left( 1 - \frac{4}{x} \right)} + 1}{\sqrt{\left( 1 - \frac{4}{x} \right)} - 1} \right), \tag{4.1}$$

where $\lambda$ is the renormalized coupling and we have fixed the renormalized mass to be $m^2 = 1$. Here Dimensional Regularization and on-shell renormalization are used to calculate loop corrections, as given in Section 10.2 in [27].

### 4.1.1 Threshold Expansion

Near the threshold, which is now $s = 4$, we find using eq.(4.1) that

$$\frac{1}{2}\left(\frac{\partial^2}{\partial x^2}(\mathcal{M}_1(s,x))\bigg|_{x_1} - \left(\frac{\partial}{\partial x}(\mathcal{M}_1(s,x))\bigg|_{x_1}\right)^2\right) \xrightarrow[\text{leading order in }\lambda]{s\to 4} \tag{4.2}$$

$$\frac{\lambda}{1920\pi^2}(s-4)^2\left(1 - \frac{3}{14}(s-4) + \cdots\right), \tag{4.3}$$

where, we Hence, we can conclude that the Relative Entropy is monotonically increasing near the threshold for all $x_1 \in (-1,1)$, only for $\lambda > 0$ (up to order $\lambda$, which is till where we can trust the expression at 1-loop). However, $\lambda \geq 0$ is the physical regime of the $\phi^4$-perturbation coefficient, $\lambda$. Consequently, we have that

$$D_Q\left(\rho_A^{(1)}\|\rho_A^{(2)}\right) = (\Delta x)^2\left[\frac{\lambda}{1920\,\pi^2}(s-4)^2\left(1 - \frac{3}{14}(s-4)\right) + O((s-4)^4)\right] \tag{4.4}$$

$$+ O((s-4)^{\frac{5}{2}},(\Delta x)^3,\lambda^2). \tag{4.5}$$

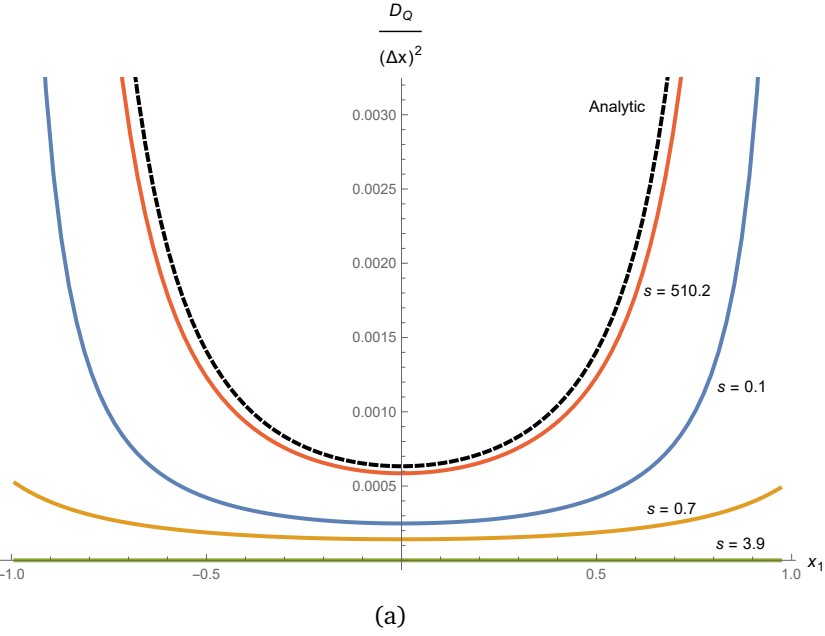

(a)

Figure 4: Behaviour of $D_Q$ corresponding to the $\phi^4$ amplitude in eq.(4.1) and its comparison to the high energy bounds found in eq.(4.7), given by the black dashed line.

### 4.1.2 High Energy Expansion

Now we switch to the high energy limit of the $\phi^4$-amplitude in eq.(4.1). In this regime, the amplitude can be expanded as

$$\frac{\mathcal{M}(s,x)}{\mathcal{M}(s,x_1)} = \frac{1 + \lambda\left(4 - i\pi + \ln\left(\frac{4}{s^3(1-x^2)}\right)\right)}{1 + \lambda\left(4 - i\pi + \ln\left(\frac{4}{s^3(1-x_1^2)}\right)\right)} + O\left(\frac{1}{s}\right) = 1 + \frac{\lambda}{32\pi^2}\ln\left(\frac{1-x_1^2}{1-x^2}\right) + O\left(\lambda^2,\frac{1}{s}\right). \tag{4.6}$$

Using this we find that

$$D_Q\left(\rho_A^{(1)}\|\rho_A^{(2)}\right) \approx (\Delta x)^2\left(\frac{\lambda}{16\pi^2}\right)\left(\frac{(1+x_1^2)}{(1-x_1^2)^2}\right) + O\left(\lambda^2,\frac{1}{s^2},(\Delta x)^3\right). \tag{4.7}$$

Hence, again we have the same condition that the Relative Entropy is monotonically increasing *w.r.t* $\Delta x$ at large $s$, for all $x_1 \in (-1, 1)$ in leading order if and only if $\lambda > 0$.

In fig.(4) we have plotted $D_Q$ for the $\phi^4$ theory for $\lambda = 0.1$. As fig.(4) shows, $D_Q$ first decreases to zero at $s = 4m^2$ before monotonically rising. In Section (5.2), we will compare the $s \gg 1$ formula in eq.(4.7) to a more general high energy bound.

## 4.2 Chiral perturbation theory

We now turn to $\chi PT$ which is a famous effective field theory to understand the low energy phenomenology of QCD. $\chi PT$[7] was invented as an effective field theory [33, 34] to explain the low energy dynamics of QCD while being approximately consistent with the underlying symmetry (which exactly match in the chiral limit *i.e.,* quark mass going to 0). Chiral perturbation theory gives good agreement to phenomenology for energies up to approximately 500 MeV. In order to incorporate 1-loop effects, one writes down a four-derivative counter-term Lagrangian [32, 35] in which there are several low energy constants (LEC's) to fix. These are fixed by comparing with experiments. Roughly speaking, scattering and resonance data enables one to fix these LEC's. This procedure can be iterated to higher orders as well with the LEC's proliferating. The state of the art is two-loops [36], although in this section we will focus on the older 1-loop results[8].

In this section, we will first start by considering the scattering amplitude as given in [32, 37] only till 1-loop (where everything is in units of the pion mass, $m_\pi$),

$$\mathcal{M}_{ab \to cd}(s, t(x), u(x)) = A(s, t, u)\delta_{ab}\delta^{cd} + A(t, u, s)\delta_a^c\delta_b^d + A(u, s, t)\delta_a^d\delta_b^c, \tag{4.8}$$

where,

$$\begin{aligned}
A(s, t, u) = & \frac{1}{f^2}(s - 1) + \frac{1}{f^4}\left(b_1 + b_2 s + b_3 s^2 + b_4(t - u)^2\right) \\
& + \frac{1}{f^4}\left(F^{(1)}(s) + G^{(1)}(s, t) + G^{(1)}(s, u)\right),
\end{aligned} \tag{4.9}$$

$$F^{(1)}(s) = \frac{1}{2}(s^2 - 1)J(s), \text{ and } G^{(1)}(s, t) = \frac{1}{6}(14 - 4s - 10t + st + 2t^2). \tag{4.10}$$

Here $f$ is the pion decay width, $f \approx 95$ MeV and

$$J(z) = \frac{1}{16\pi^2}\left(-2\sqrt{\left(1 - \frac{4}{z}\right)}\ln\left(\frac{1}{2}(\sqrt{z - 4} + \sqrt{z})\right) + i\pi\sqrt{\left(1 - \frac{4}{z}\right)} + 2\right), \tag{4.11}$$

where $J(z)$ is analytically continued to $z < 4$ in such a way that $J(0) = 0$. Furthermore, all the $b_i, i = \{1, 2, 3, 4\}$ are re-normalized, scale dependent constants.

First we will look at the high energy limit of the relative entropy. We first expand the amplitude as

$$\frac{\mathcal{M}(s, x)}{\mathcal{M}(s, x_1)} = \left(\frac{3 + x^2}{3 + x_1^2}\right) + O\left(\frac{1}{\ln(s)}\right), \tag{4.12}$$

which gives us that:

$$D_Q\left(\rho_A^{(1)} \| \rho_A^{(2)}\right) \approx 2\frac{(3 - x_1^2)}{(3 + x_1^2)^2} + O\left(\frac{1}{\ln(s)}, (\Delta x)^3\right). \tag{4.13}$$

---

[7]For some important reviews and uses of $\chi PT$, please refer to [28], [29], [30], [31] and [32].

[8]We are thankful to B. Ananthanarayan for educating us on $\chi PT$!

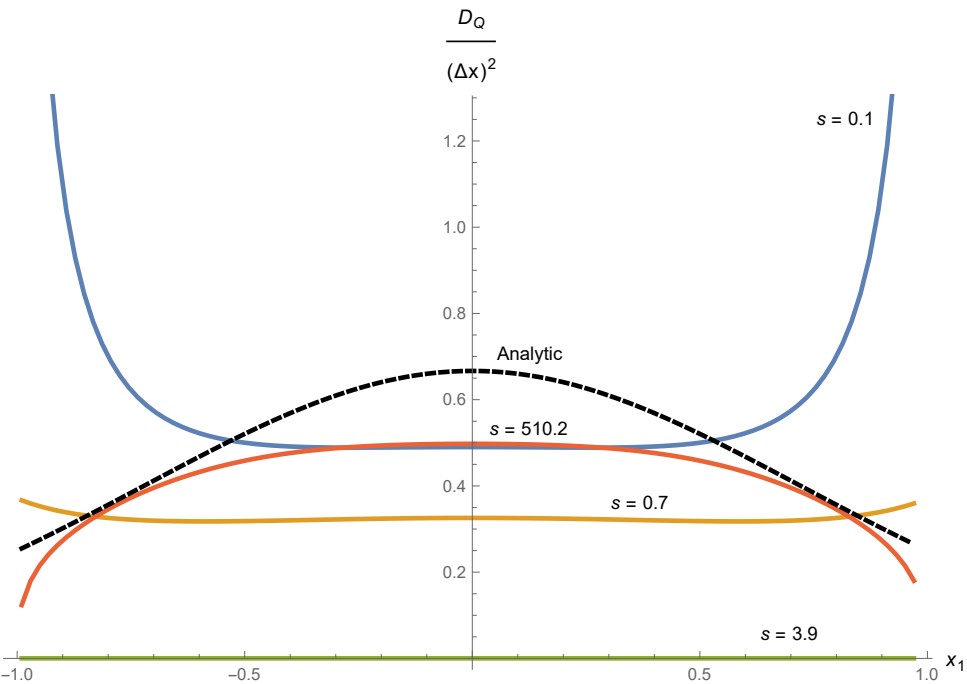

Figure 5: Behaviour of $D_Q$ for the $\chi PT$ (similar to fig.(4) for the $\phi^4$ amplitude) amplitude as a function of $x_1$ for different values of $s$ (with Data-fitted values of the parameters taken from [37] and [38]). The analytic bound is the one found in eq.(4.13) which is different than the one found in Section (5.2).

Note that this form is going to be different from the general high energy limits to be considered in later sections–this is because $\chi PT$ is an effective field theory which will not satisfy the polynomial boundedness conditions assumed non-perturbatively.

Next, we will again be focusing on checking the behaviour of the Relative entropy *w.r.t* $\Delta x$ near the threshold, $s = 4$ and to leading order in perturbation theory i.e., in order of $\frac{1}{f^2}$. In doing so, we will find constraints on the $b_i$-coefficients (effectively only on $b_3$ and $b_4$) which we will compare with the best fit values found in [38] (Note that all the $b_i$ values are compared at the scale of the pion mass). We will be finding these constraints separately for the following three reactions, which we have chosen to be the three independent reactions to which the rest are related by crossing symmetry. These are:

$$
\begin{aligned}
\pi^0\,\pi^0 &\longrightarrow \pi^0\,\pi^0\,, \\
\pi^+\,\pi^- &\longrightarrow \pi^0\,\pi^0\,, \\
\pi^{+(-)}\,\pi^{+(-)} &\longrightarrow \pi^{+(-)}\,\pi^{+(-)}\,.
\end{aligned}
\tag{4.14}
$$

### 4.2.1 $\pi^0\,\pi^0 \longrightarrow \pi^0\,\pi^0$

The amplitude for this reaction is as found in table (6). So using eq.(4.10) for the amplitude and expanding around $s = 4$ to leading order in perturbation, we get (keeping in mind that in leading order the amplitude is real and hence can simply be whole squared when substituted

into eq.(3.18))

$$D_Q\left(\rho_{\pi^0}^{(1)}\|\rho_{\pi^0}^{(2)}\right) \xrightarrow[\text{leading order in } \frac{1}{f^2}]{s\to 4} \frac{(\Delta x)^2}{f^2}\left(b_3 + 3\,b_4 - \frac{37}{1920\,\pi^2}\right)(s-4)^2$$
$$+ O\left((s-4)^{\frac{5}{2}}, \frac{1}{f^3}, (\Delta x)^3\right). \tag{4.15}$$

Therefore, whether the relative entropy is monotonically increasing or decreasing as a function of $(s-4)^2$, depends upon the sign of $D_Q$ governed by the combination,

$$D_Q\left(\rho_{\pi^0}^{(1)}\|\rho_{\pi^0}^{(2)}\right) \propto \left(b_3 + 3\,b_4 - \frac{37}{1920\,\pi^2}\right). \tag{4.16}$$

### 4.2.2 $\pi^+\pi^- \longrightarrow \pi^0\pi^0$

The amplitude goes as mentioned in table (6), and from eq.(4.10), following similar steps as in the previous reaction, we get that

$$D_Q\left(\rho_{\pi^0}^{(1)}\|\rho_{\pi^0}^{(2)}\right) \xrightarrow[\text{leading order in } \frac{1}{f^2}]{s\to 4} \frac{2}{3}\frac{(\Delta x)^2}{f^2}\left(b_4 - \frac{31}{5760\,\pi^2}\right)(s-4)^2 + O\left((s-4)^{\frac{5}{2}}, \frac{1}{f^3}, (\Delta x)^3\right). \tag{4.17}$$

Hence the sign of $D_Q$ will depend on

$$D_Q\left(\rho_{\pi^0}^{(1)}\|\rho_{\pi^0}^{(2)}\right) \propto \left(b_4 - \frac{31}{5760\,\pi^2}\right). \tag{4.18}$$

### 4.2.3 $\pi^{+(-)}\pi^{+(-)} \longrightarrow \pi^{+(-)}\pi^{+(-)}$

Now, the amplitude can again be found in table (6), so we get that

$$D_Q\left(\rho_{\pi^{+(-)}}^{(1)}\|\rho_{\pi^{+(-)}}^{(2)}\right) \xrightarrow[\text{leading order in } \frac{1}{f^2}]{s\to 4} \frac{1}{2}\frac{(\Delta x)^2}{f^2}\left(-b_3 - b_4 + \frac{49}{5760\,\pi^2}\right)(s-4)^2$$
$$+ O\left((s-4)^{\frac{5}{2}}, \frac{1}{f^3}, (\Delta x)^3\right). \tag{4.19}$$

Therefore, like before, we have that the sign depends on

$$D_Q\left(\rho_{\pi^{+(-)}}^{(1)}\|\rho_{\pi^{+(-)}}^{(2)}\right) \propto -\left(b_3 + b_4 - \frac{49}{5760\,\pi^2}\right). \tag{4.20}$$

We will now represent the combinations of $b_i$'s in terms of the scattering lengths. We use the definition of the scattering lengths (eg. [22]),

$$a_\ell^{(I)} = \frac{4^\ell \ell!}{(2\ell+1)}\frac{\partial^\ell}{\partial t^\ell}\left(\mathcal{M}^{(I)}(s,t)\right)\Big|_{s=4,t=0} = \frac{4^\ell \ell!}{(2\ell+1)}\sum_J\left(C_{st}^{IJ}\frac{\partial^\ell}{\partial s^\ell}\left(\mathcal{M}^{(J)}(s,t)\right)\Big|_{s=0,t=4}\right), \tag{4.21}$$

where $\ell = 0,1,2,\dots$ is the spin of the partial wave and $I = 0,1,2$ are the three channels of the amplitude with $O(3)$ symmetry, namely the singlet, anti-symmetric and the traceless symmetric channels defined as

$$\mathcal{M}^{(0)}(s,t) = 3\,A(s,t,4-s-t) + A(t,4-s-t,s) + A(4-s-t,s,t),$$
$$\mathcal{M}^{(1)}(s,t) = A(t,4-s-t,s) - A(4-s-t,s,t), \tag{4.22}$$
$$\mathcal{M}^{(2)}(s,t) = A(t,4-s-t,s) + A(4-s-t,s,t),$$

with $A(s, t, u)$ as given in eq.(4.10). $C_{st}^{IJ}$ is the involutory (*i.e.* is its own inverse) crossing matrix relating the $s$ and $t$ channels such that

$$\mathcal{M}^{(I)}(s, t, u) = \sum_J \left( C_{st}^{IJ} \mathcal{M}^{(J)}(t, s, u) \right), \tag{4.23}$$

with

$$C_{st} = \frac{1}{6} \begin{pmatrix} 2 & 6 & 10 \\ 2 & 3 & -5 \\ 2 & -3 & 1 \end{pmatrix}. \tag{4.24}$$

Please note that the expression in eq.(4.21) differs from the standard definition of the scattering lengths in eq.(5.2) by just some constant factors which can be explicitly worked out to be of the form $32\pi \frac{(2\ell)!}{4^\ell}$.

Now, since the derivative is *w.r.t* $t$, we can just expand $M^{(I)}(4, t)$ in powers of $t$ for each channel up-to some order to get the respective scattering lengths. Note that by its definition, since the $I = 0, 2$ channels are symmetric in $t \leftrightarrow u = -t$ (at $s = 4$), we must have that $I = 0, 2$ channels only have even powers of $t$ in their expansion while the $I = 1$ channel only has odd powers of $t$. Upon doing this exercise, we get the expressions for the S-wave scattering lengths as

$$
\begin{aligned}
a_0^{(0)} &= \frac{7}{f^2} + \frac{1}{f^4} \left( 5 b_1 + 12 b_2 + 48 b_3 + 32 b_4 + \frac{49}{16\pi^2} \right), \\
a_0^{(2)} &= -\frac{2}{f^2} + \frac{2}{f^4} \left( b_1 + 16 b_4 + \frac{1}{8\pi^2} \right),
\end{aligned} \tag{4.25}
$$

and for the P-wave scattering length, we have

$$a_1^{(1)} = \frac{8}{3 f^2} + \frac{8}{3 f^4} \left( b_2 + 8 b_4 - \frac{17}{576\pi^2} \right), \tag{4.26}$$

while the D-wave scattering lengths are

$$
\begin{aligned}
a_2^{(0)} + 2 a_2^{(2)} &= \frac{1152}{5 f^4} \left( b_3 + 3 b_4 - \frac{37}{1920\pi^2} \right), \\
a_2^{(0)} - a_2^{(2)} &= \frac{768}{5 f^4} \left( b_4 - \frac{31}{5760\pi^2} \right), \\
a_2^{(2)} &= \frac{128}{5 f^4} \left( b_3 + b_4 - \frac{49}{5760\pi^2} \right),
\end{aligned} \tag{4.27}
$$

which are exactly the combinations obtained above for the three independent reactions. Now as we will review in the next section, precisely the first two D-wave scattering length combinations are positive! This condition follows from the Froissart-Gribov representation of the $\ell \geq 2$ scattering lengths and is quite general. Namely, we have that

$$a_2^{(0)} + 2 a_2^{(2)} \geq 0, \quad a_2^{(0)} - a_2^{(2)} \geq 0, \quad 2 a_2^{(0)} + a_2^{(2)} \geq 0. \tag{4.28}$$

Now for the $a_2^{(2)}$ scattering length, there appears to be a choice. By choosing this to be positive, we will find that the phenomenological values lie within the admitted region in fig.(6). The other sign is in fact strongly disfavoured as it would rule out the best fit and experimental values. We do not know of a fundamental reason for this and will assume that $a_2^{(2)} \geq 0$ continues to hold for physically interesting theories. Furthermore, note that in $\chi PT$ from eq.(4.25) and eq.(4.26) we have $a_0^{(0)}, a_1^{(1)} \geq 0, a_0^{(2)} \leq 0$ but more usefully

$$a_0^{(0)} + 2 a_0^{(2)} \geq 0, \quad 2 a_0^{(0)} + a_0^{(2)} \geq 0, \quad a_0^{(0)} - a_0^{(2)} \geq 0, \quad a_0^{(2)} \leq 0, \tag{4.29}$$

at leading order. In fact, these inequalities in eq.(4.28) and eq.(4.29) are also supported by experimental data taken from [38]. Now, if we combine all the three constraints eq.(4.28) and plot them along with the best fit values from experimental data of the partial waves found in [38], we find the allowed region in fig.(6). As can be seen, the best fit value is quite close to the boundary of the combined allowed region.

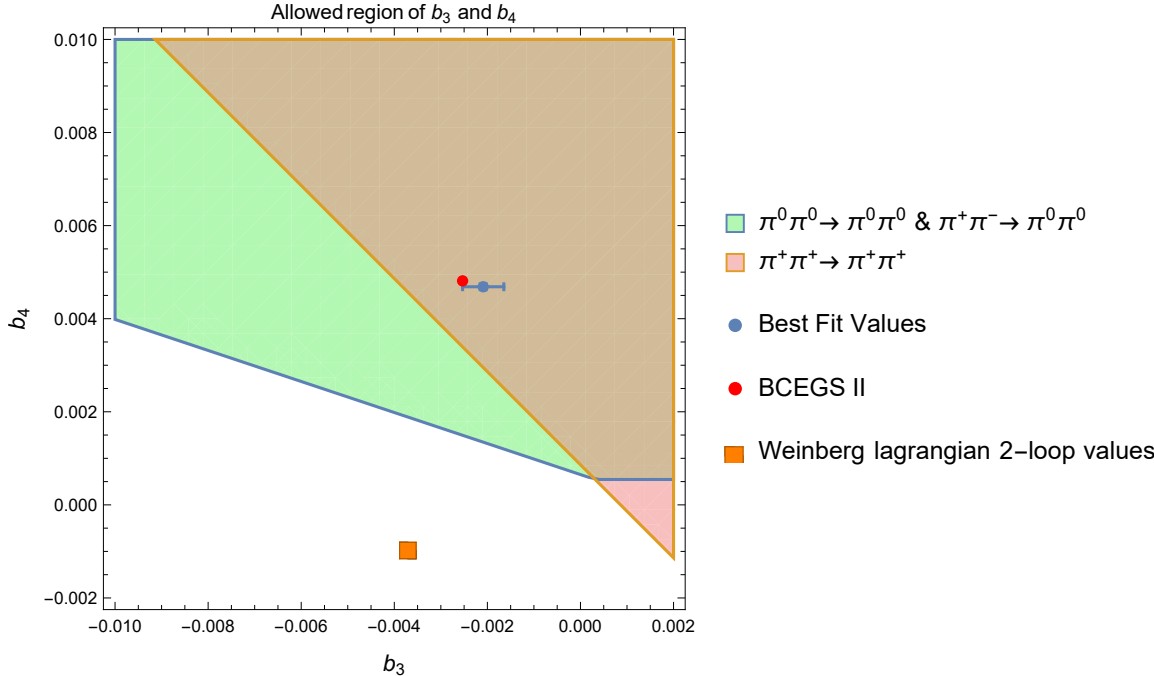

Figure 6: Allowed region of $b_3$ and $b_4$ for monotonically changing Relative Entropy with signs fixed as in eq.(4.29) and eq.(4.28) and $a_2^{(2)} \geq 0$, compared to the Data-fitted values taken from [37] and [38].

## 4.3 Type II superstring theory

After considering $\phi^4$ and the effective field theory $\chi PT$ we turn to the scattering amplitude for four dilatons in tree level type II string Theory (in the units of the length of string squared, $\alpha'$, which has been set as $\alpha' = 4$) [39],

$$\mathcal{M}(s,t,u) = \frac{\Gamma(-s)\Gamma(-t)\Gamma(-u)}{\Gamma(1+s)\Gamma(1+t)\Gamma(1+u)} \,. \tag{4.30}$$

Noting that the Gamma function, $\Gamma(z)$ does not have any zeros and has poles at $z \in \mathbb{Z}_-$, we highlight the following simple properties of this amplitude which will be relevant later on:

- **Zeroes of the Amplitude :** The zeroes of the amplitude will occur when the Gamma functions in the denominator has a pole. This will happen when either $(1+s), (1+t), (1+u) \in \mathbb{Z}_-$. However, in the physical region we have that $s \geq 0, t \leq 0, u \leq 0$. Therefore, the only zeros will occur when

$$s \in \left\{ 2\frac{(1+n)}{1+x}, 2\frac{(1+n)}{1-x}; \quad n \in \mathbb{N} \right\}, \text{ for fixed } x \,. \tag{4.31}$$

- **Poles of the Amplitude :** The poles of the amplitude will occur when the Gamma functions in the numerator encounters a pole. This will happen when either $(-s), (-t),$

$(-u) \in \mathbb{Z}_-$. However, again, since in the physical region we have that $s \geq 0$, $t \leq 0$, $u \leq 0$, hence, the poles will occur only when

$$s \in \mathbb{N}. \tag{4.32}$$

These effect of the aforementioned properties on the Entanglement and Relative Entropy will be clear when we look at eq.(3.7) and also use expressions derived in Appendix (B.2). Noting that

$$\mathcal{P}_g(x) = \frac{g(x)\,|\mathcal{M}(s,x)|^2}{|\mathcal{M}(s,y)|^2 + \sigma \frac{\partial^2}{\partial x^2}\left(|\mathcal{M}(s,x)|^2\right)\big|_y + O(\sigma^2)}\,, \tag{4.33}$$

we see that $\mathcal{P}_g(x)$ has a peak when $(s,y)$ is a zero of the amplitude and conversely, is 0 when $(s,y)$ is a pole of the amplitude. Therefore, we expect the Entanglement and the Relative Entropy to have such behaviour also. In the following section, we mark the zeros with red dotted lines and the poles with green dotted lines.

### 4.3.1 Fixed $x_1$ and $|\Delta x|$ with $\sigma = 10^{-4}$

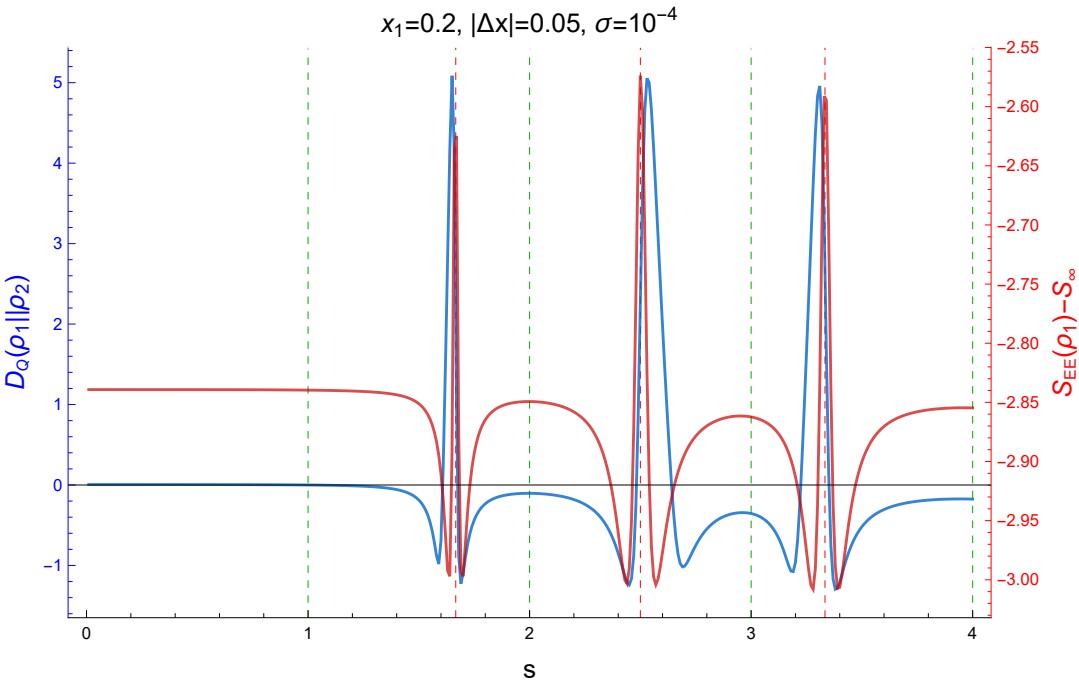

Figure 7: Plot of Relative Entropy between states at $(s, x_1)$ and $(s, x_1 + |\Delta x|)$ (in Blue) and the Entanglement Entropy for states at $(s, x_1)$ (in Red) vs $s$

The main observation is that the Relative Entropy and the Entanglement Entropy are positively correlated and this correlation is higher for smaller $\Delta x$ as can be seen in fig.(7).

### 4.3.2 Fixed $s$ and $|\Delta x|$

Before we plot, we remind ourselves of the simple observation that the relative entropy is expected to be the same for $x_1, x_1 - |\Delta x|$; $x_1 > 0$ and $x_1, x_1 + |\Delta x|$; $x_1 < 0$ and vice versa by physical symmetry. So to make this symmetry explicit in our plots of $S_R(s, x_1, x_1 - \Delta x)$, we should choose a convention where the sign of $\Delta x$ is fixed to be opposite for $x_1 > 0$ and $x_1 < 0$. In fig.(8), we have chosen it such that $\Delta x > 0$; $x_1 > 0$ and vice versa (Note that this does not change the nature of the plots at all, just shifts it a bit).

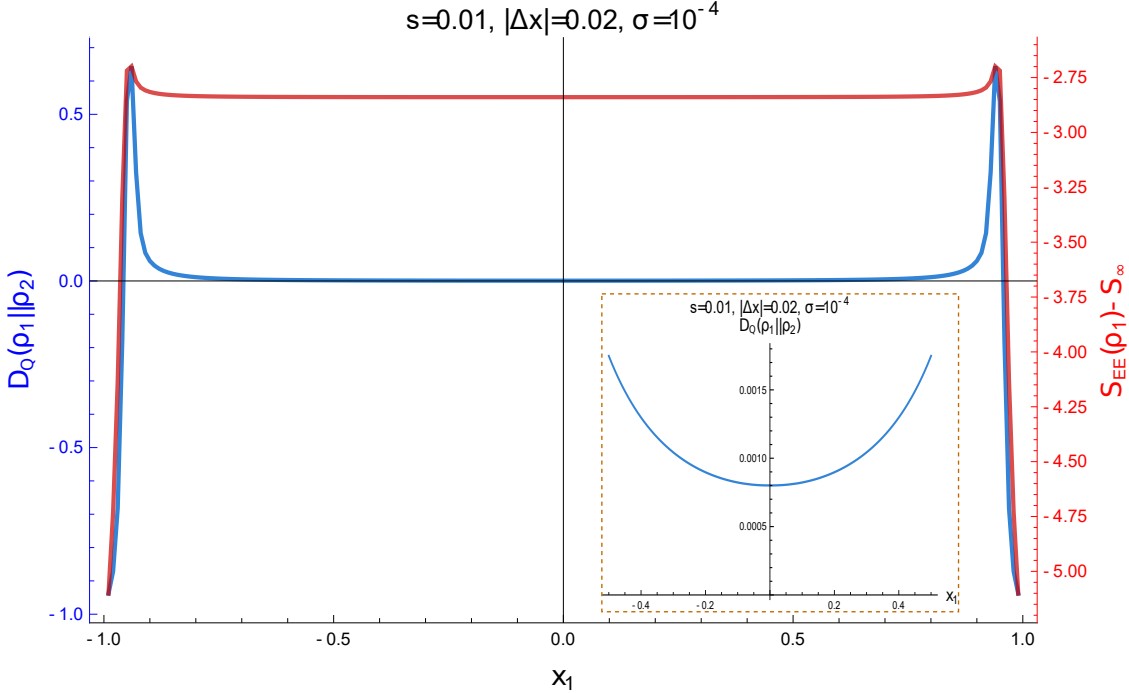

Figure 8: Zoomed in plots to show behaviour near $x_1 = 0$.

### 4.3.3 Monotonicity in $\Delta x$

Now we perform a simple analytic exercise to check monotonicity of the Relative Entropy *w.r.t* $\Delta x$ for small enough values of $\Delta x$ similar to eq.(B.38). Furthermore, we check the sign of the coefficient of $(\Delta x)^2$ in eq.(B.38) close to the threshold $s = 0$. Using eq.(4.30) and expanding around the threshold gives us

$$\frac{|\mathcal{M}(s,x)|^2}{|\mathcal{M}(s,x_1)|^2} = \left(\frac{\mathcal{M}(s,x)}{\mathcal{M}(s,x_1)}\right)^2 \xrightarrow{s\to 0} \frac{(1-x_1^2)^2}{(1-x^2)^2} + O(s^3), \tag{4.34}$$

using which we find

$$\frac{1}{2}\frac{\partial^2}{\partial x^2}\left(\ln\left(\frac{|\mathcal{M}(s,x)|^2}{|\mathcal{M}(s,x_1)|^2}\right)\right)\Bigg|_{x_1} \xrightarrow{s\to 0} 2\frac{(1+x_1^2)}{(1-x_1^2)^2} + O(s^3). \tag{4.35}$$

Therefore, in leading order, $D_Q$ in eq.(3.19), which is the relative entropy above the hard sphere value of $(\Delta x)^2/4\sigma$ near the threshold will be monotonically increasing for all $x_1 \in (-1,1)$ as

$$D_Q\left(\rho_A^{(1)}\|\rho_A^{(2)}\right) = 2(\Delta x)^2\left(\frac{(1+x_1^2)}{(1-x_1^2)^2} - \zeta(3)s^3 + O(s^4)\right) + O((\Delta x)^3). \tag{4.36}$$

Here we have explicitly shown the $O(s^3)$ term which is a constant in $x_1$ in this case.

## 5 Relative entropy: general considerations

In this section, we will give a unifying framework to explain some of the observations above. In particular, we will consider the behaviour of relative entropy close to the threshold $s = 4m^2$ as well as in the high energy limit $s \to \infty$ to extract certain general conclusions.

## 5.1 Relative entropy in terms of scattering lengths

If we are close to the threshold $s = 4m^2$, we can derive a general formula in terms of the scattering lengths for the relative entropy considered above. This is generally valid in absence of massless exchange poles. We start with the partial wave expansion,

$$\mathcal{M}^{(I)}(s,t) = \sum_{\ell=0}^{\infty} (2\ell + 1) f_\ell^{(I)}(s) P_\ell(x). \tag{5.1}$$

Then we write an expansion for $f_\ell^{(I)}(s)$ valid near the threshold $s = 4m^2$ as follows,

$$\text{Re}[f_\ell^{(I)}(s)] = \left(\frac{s - 4m^2}{4}\right)^\ell \left(a_\ell^{(I)}(s) + b_\ell^{(I)}\left(\frac{s - 4m^2}{4}\right) + \cdots\right), \tag{5.2}$$

where $a_\ell^{(I)}$ are the scattering lengths for the $I$'th isospin and the $b_\ell^{(I)}$'s are the effective ranges. We can also show that $f_\ell^{(I)}$'s are real at leading order by using the expansion of $1/f_\ell^{(I)}$ given in Section 5 of [54]. Hence we would get scattering lengths at the leading order even when we consider the expansion of $f_\ell^{(I)}$. This is equivalent to the observation that we had noted previously in Section (4) where the amplitudes were real in leading order. With this in mind, it is now straightforward to verify that when only even spins are exchanged, the quantum relative entropy $D_Q$ is given by

$$D_Q(\rho_1 \| \rho_2) = \frac{15}{4} (\Delta x)^2 \frac{\sum_I c_I a_2^{(I)}}{\sum_I c_I a_0^{(I)}} (s - 4m^2)^2. \tag{5.3}$$

Here $\sum_I c_I a_\ell^{(I)}$ is a linear combination of the scattering lengths depending on the process being considered. We tabulate the coefficients $c_I$ below for the processes where only even spins are exchanged:

$$\pi^0 \pi^0 \to \pi^0 \pi^0 \quad : \quad c_0 = 1, c_2 = 2, \tag{5.4}$$

$$\pi^+ \pi^+ \to \pi^+ \pi^+ \quad : \quad c_0 = 0, c_2 = 1, \tag{5.5}$$

$$\pi^+ \pi^- \to \pi^0 \pi^0 \quad : \quad c_0 = 1, c_2 = -1. \tag{5.6}$$

Now writing down the Froissart-Gribov representation for the $\ell = 2$ scattering lengths, one can derive inequalities for the combination appearing in the numerator [22, 40]. The logic is to observe that assuming the Froissart bound to hold and writing down a twice-subtracted dispersion relation, the $\ell \geq 2$ scattering lengths admit a Froissart-Gribov representation whose integrands can be shown to be positive, being related to the scattering cross sections. This is reviewed in [22] and we refer the reader to that reference. We start with the Froissart-Gribov representation for the derivative of the amplitude (as given in [22]),

$$\frac{\partial^2}{\partial s^2} \left(\mathcal{M}^{(I)}(s,t)\right)\Big|_{s=0,t=4} = \frac{2}{\pi} \sum_J \left(\int_{4m^2}^{\infty} \frac{ds'}{(s')^3} \left(\delta^{IJ} + C_{su}^{IJ}\right) \text{Im}[\mathcal{M}^{(J)}(s',4)]\right). \tag{5.7}$$

Here, the first term is the contribution from the cut $s' \geq 4$ and the second term is from the cut $u' \geq 4 \leftrightarrow s' \leq 4$ after a simple variable change. Here $C_{su}^{IJ}$ is the equivalent of eq.(4.23) for the crossing matrix relating the $s$ and the $u$ channels such that

$$C_{su} = \frac{1}{6} \begin{pmatrix} 2 & -6 & 10 \\ -2 & 3 & 5 \\ 2 & 3 & 1 \end{pmatrix}. \tag{5.8}$$

Then using the optical theorem we simply observe that for the physical processes of the form $A+B \to A+B$, the integrand in eq.(5.7) is positive. This is so as the Optical Theorem guarantees the positivity of the absorptive part at $s \geq 4$, $t = 0$ which can trivially be extended to $s \geq 4$, $t \geq 0$ since all the Legendre Polynomials are positive for $x \geq 1$. This argument implies that the first part of the integrand i.e., the $s$-cut contribution is positive. However, for the process $A + B \to A + B$, the crossing symmetric process in the $u-$ channel is $A + \bar{B} \to A + \bar{B}$ which is also a valid process for applying the Optical Theorem and consequently, positivity. Hence, for such processes, or equivalently, such linear combinations of the LHS in eq.(5.7) is positive. Thereafter, using eq.(4.21), we find certain linear combinations of the scattering length to be positive so that

$$\sum_I c_I \, a_2^{(I)} \geq 0 \, , c_I = \sum_J d_J C_{st}^{JI} \, , \tag{5.9}$$

where $d_I$ are the coefficients corresponding to the reactions $\pi^0 \, \pi^0 \to \pi^0 \, \pi^0$, $\pi^+ \, \pi^+ \to \pi^+ \, \pi^+$, and $\pi^0 \, \pi^+ \to \pi^0 \, \pi^+$ (as given in table (6)) which lead to the the the $c_I$'s displayed in eq.(4.28), namely

$$a_2^{(0)} + 2a_2^{(2)} \geq 0 \, , \quad a_2^{(0)} - a_2^{(2)} \geq 0 \, , \quad 2a_2^{(0)} + a_2^{(2)} \geq 0 \, . \tag{5.10}$$

These also imply that $a_2^{(0)} \geq 0$. The last inequality in eq.(5.10) follows from the first two and need not be considered independently. Now, as explained before, the $\chi PT$ calculations also imply $a_2^{(2)} \geq 0$. However, imposing this last inequality does not have any significant effect in the S-matrix bootstrap numerics.

Now unfortunately, inequalities for the $\ell = 0$ scattering lengths do not follow from similar arguments since the an analogous integral representation does not exist. However, using the $\chi PT$ Lagrangian, it is easy to show that the following inequalities hold (see [38] and the explicit formulas eq.(4.25), eq.(4.26) in the previous section):

$$a_0^{(0)} + 2a_0^{(2)} \geq 0 \, , \quad 2a_0^{(0)} + a_0^{(2)} \geq 0 \, , \quad a_0^{(0)} - a_0^{(2)} \geq 0 \, , \quad a_0^{(2)} \leq 0 \, . \tag{5.11}$$

These are similar to the D-wave scattering inequalities. These S-wave inequalities have the strongest effect on the S-matrix bootstrap numerics. If we insist that these continue to hold for a physical theory, then we find the following signs for $D_Q$ for small $\Delta x$ and $s \sim 4m^2$:

$$\pi^0\pi^0 \to \pi^0\pi^0 \quad : \quad D_Q \geq 0 \tag{5.12}$$

$$\pi^+\pi^+ \to \pi^+\pi^+ \quad : \quad D_Q \leq 0 \tag{5.13}$$

$$\pi^+\pi^- \to \pi^0\pi^0 \quad : \quad D_Q \geq 0 \, . \tag{5.14}$$

The bottom-line of the analysis in this section is that the sign of $D_Q$ is correlated with dispersion relation bounds. The other two processes, namely $\pi^+\pi^0 \to \pi^+\pi^0$ and $\pi^+\pi^- \to \pi^+\pi^-$ also involve odd spin partial waves and lead to more complicated expressions like

$$\pi^+\pi^0 \to \pi^+\pi^0 \quad : \quad D_Q = -3(\Delta x)^2(s - 4m^2)^2 \frac{\frac{1}{4}(a_1^{(1)})^2 - \frac{5}{4}a_0^{(2)}a_2^{(2)}}{(a_0^{(2)})^2} \, , \tag{5.15}$$

$$\pi^+\pi^- \to \pi^+\pi^- \quad : \quad D_Q = \frac{3}{4}(\Delta x)^2(s - 4m^2)^2 \left[ 5\frac{a_2^{(0)} + 2a_2^{(2)}}{a_0^{(0)} + 2a_0^{(2)}} - \frac{9}{4}\frac{(a_1^{(1)})^2}{(a_0^{(0)} + \frac{1}{2}a_0^{(2)})^2} \right] . \tag{5.16}$$

Using eq.(5.11) and assuming $a_2^{(2)} \geq 0$ we in fact find $D_Q \leq 0$ for $\pi^+\pi^0 \to \pi^+\pi^0$. The other reaction does not appear to have a definite sign. Note that we did not need to assume anything about $a_1^{(1)}$.

The point of view that we will adopt in what follows is that imposing the above signs for $D_Q$ will enable us to consider the positivity constraints on the D-wave scattering lengths which when supplemented by the S-wave scattering length constraints, we will get consistency conditions that will enable us to rule out various regions.

**Comment on Rényi divergence**

In the limit when $\Delta x \ll 1, \sigma \ll 1$, using eq.(B.45) one can easily verify that the only change that happens in the Rényi divergence expression is that the relative entropy expressions get multiplied by a factor of $n$. This is a pleasingly simple result. Beyond the limit $\Delta x \ll 1$, the results will be dependent on $n$ in a more interesting manner, but this is something that we will not pursue in this paper.

## 5.2 High energy bounds

In this section we will consider the high energy behaviour of relative entropy. For definiteness, we have in mind the $\pi^0 \pi^0 \to \pi^0 \pi^0$ (or more generally $AA \to AA$ in massive QFTs) scattering. Our starting point will be eq.(3.19) which we reproduce below for convenience:

$$D_Q\left(\rho_A^{(1)}\|\rho_A^{(2)}\right) = \frac{1}{2}(\Delta x)^2\left(\frac{\partial^2}{\partial x^2}(\mathcal{M}_1(s,x))\Big|_{x_1} - \left(\frac{\partial}{\partial x}(\mathcal{M}_1(s,x))\Big|_{x_1}\right)^2\right) + O((\Delta x)). \quad (5.17)$$

Using the fact that

$$|\mathcal{M}(s,x)|^2 = \frac{s}{8\pi}\frac{d\sigma_{el}}{dx}, \quad (5.18)$$

where $\frac{1}{2\pi}\frac{d\sigma_{el}}{dx}$ is the elastic differential cross-section, we can easily show that

$$D_Q\left(\rho_A^{(1)}\|\rho_A^{(2)}\right) = \frac{1}{2}(\Delta x)^2\left(\frac{\sigma_{el}'''}{\sigma_{el}'} - \left(\frac{\sigma_{el}''}{\sigma_{el}'}\right)^2\right)\Big|_{x=x_1}, \quad (5.19)$$

where we have used the shorthand notation $\sigma_{el}' = d\sigma_{el}/dx$ and have dropped the $O((\Delta x)\sigma)$ terms. We expect that this form will be useful for phenomenological studies in the future. This immediately leads to the inequality

$$D_Q\left(\rho_A^{(1)}\|\rho_A^{(2)}\right) \le \frac{1}{2}(\Delta x)^2 \frac{\sigma_{el}'''}{\sigma_{el}'}\Big|_{x=x_1}. \quad (5.20)$$

This is quite generally true irrespective of the regime of $s$. Now high energy bounds on differential elastic cross sections have been worked out previously by Martin and collaborators in [10] and by Singh and Roy in [41] building on the work by [42]. The actual bound is not going to be relevant for us. It suffices to note that the bound on the differential cross section for $s \gg 4m^2$ and $-1 < x < 1$ is of the form [10, 41]

$$\sigma_{el}' \le \frac{f(s)}{(1-x^2)^p}. \quad (5.21)$$

The power $p = 1/2$ in [41] while it is $p = 2$ in [10]–these papers use different convergence criteria. The derivation of such "Froissart-like" bounds for massive QFTs uses analyticity, unitarity and polynomial boundedness inside the so-called Lehmann-Martin ellipse [41] or a larger ellipse [10] and $f(s)$ works out to be a known function having dependence on $\sqrt{s}\ln s$ as well as some unfixed parameters[9]. In [41], a more complicated form of the bound is also given, dropping the polynomial boundedness condition, making phenomenological studies where the energy is never so high as to be sensitive to the $\ln s$ behaviour more plausible. In our case, the $f(s)$ dependence will eventually drop out and we will focus first on using the form in eq.(5.21). In order to use the above inequality in a useful manner we write

$$\sigma_{el}' = \frac{f(s)}{(1-x^2)^p}\left(1 + \frac{g(x)}{s^\alpha}\right), \quad (5.22)$$

---

[9]Thankfully in the relative entropy bound there are no unfixed parameters.

where $\alpha > 0$ and $g(x)$ is a continuous positive function with bounded derivatives, i.e., there exists some $g_{max}$ such that $g''(x) < g_{max}$ with $g_{max} > 0$ for $-1 < x < 1$. Using this it is easy to show that

$$D_Q\left(\rho_A^{(1)} \| \rho_A^{(2)}\right) \leq p\,(\Delta x)^2 \left(\frac{1 + (2p+1)x_1^2}{(1-x_1^2)^2} + \frac{g_{max}}{s^\alpha}\right). \tag{5.23}$$

Therefore, in the $s \to \infty$ regime we find

$$D_Q\left(\rho_A^{(1)} \| \rho_A^{(2)}\right) \leq p\,(\Delta x)^2 \frac{1 + (2p+1)x_1^2}{(1-x_1^2)^2}. \tag{5.24}$$

The R.H.S is positive everywhere in $-1 < x_1 < 1$ and monotonically goes towards infinity in $0 \leq x_1 < 1$. We expect that this bound will hold in situations where the assumptions of unitarity, analyticity and polynomial boundedness hold. For instance the form in eq.(4.7) for the $\lambda\phi^4$ theory is very similar to the above form. Using the high energy bound derived above we land up with the constraint:

$$(\Delta x)^2 \left(\frac{\lambda}{16\pi^2}\right)\left(\frac{(1+x_1^2)}{(1-x_1^2)^2}\right) \leq p\,(\Delta x)^2 \frac{1 + (2p+1)x_1^2}{(1-x_1^2)^2}, \tag{5.25}$$

which leads to a bound on the coupling $\lambda \leq 8\pi^2$ for $p = 1/2$ and $\lambda \leq 32\pi^2$ for $p = 2$. Of course this should not be taken too seriously since we have used only the 1-loop perturbation theory to reach this conclusion. Nevertheless, we do expect the general philosophy behind such an argument to be true–namely high energy considerations should restrict the coupling. Numerical S-matrix bootstrap arguments[10] also lead to bounds on the coupling but the considerations there are quite different [16].

The string theory answer for $s \ll 4/\alpha'$, which is essentially the supergravity limit, is very similar to this form except that we do not expect perturbative string theory to respect polynomial boundedness [43]. So in situations where there are massless exchanges or where the stringy modes become relevant, the form of the above bound is expected to change. It is easy to check that in the string theory answer when $s \gg 1/\alpha'$, the behaviour of the relative entropy develops vigorous oscillations. Presumably, this is an indication of the extended length of the string coming into play and affecting distinguishability. In fact the low energy limit of the string answer given in eq.(4.36) violates the $p = 1/2$ result of [41] but respects the $p = 2$ result of [10]. This is still quite surprising, and perhaps the reason for this is that in the massive QFT context when we take $s \to \infty$ we can essentially ignore the masses of the scattering particles.

**Comment on the Rényi divergence bound**

Using eq.(B.45) it is easy to repeat the above exercise using eq.(5.21) leading to

$$D_n(\rho_1 \| \rho_2) \leq n\frac{(\Delta x)^2}{4\sigma} + n\,p\,(\Delta x)^2 \left(\frac{1 + (2p+1)x_1^2}{(1-x_1^2)^2}\right). \tag{5.26}$$

## 5.3 Regge behaviour of $D_Q$

Let us turn to the $s-$channel Regge behaviour of $D_Q$ for $AA \to AA$ scattering. In this limit, we will consider taking $|s| \to \infty$ keeping $t$ fixed. The amplitude behaves as [44]

$$|\mathcal{M}(s,t)| \approx \beta(t)\left(\frac{s}{s_0}\right)^{\ell(t)}. \tag{5.27}$$

---

[10]Translating the results of [16] using the 1-loop $\phi^4$ gives $\lambda \lesssim 267.4$ which we quote for fun!

The function $\ell(t)$ is called the Regge trajectory and we are considering the leading trajectory. The study of the functional form of $\ell(t)$ has been actively pursued on both theoretical grounds (in the 1960's which eventually led to string theory) as well as on experimental grounds. In fact, realistic Regge trajectories are drawn from experimental data. These are found to have the generic form [45] for small $t$

$$\ell(t) = \alpha_0 + \alpha_1 t + \frac{\alpha_2}{2} t^2 + O(t^3), \tag{5.28}$$

with $\alpha_2 > 0$ (supported by experiments; we do not know of any other existing reason!). This non-linearity is *crucial*. From [44], one can also write $\beta(t) \sim \beta_0 + \beta_1 t + \frac{\beta_2}{2} t^2 + \cdots$ if there are no massless resonances. A strictly linear Regge trajectory has been shown to violate the Cerulus-Martin fixed angle bound as well as the Froissart bound–see [45]. To get some mileage out of the relative entropy considerations, let us rewrite eq.(3.19) as

$$D_Q\left(\rho_A^{(1)}\|\rho_A^{(2)}\right) = \frac{(\Delta t)^2}{2} \partial_t^2 \left[\ln\left(\frac{|\mathcal{M}(s,t)|^2}{|\mathcal{M}(s,t_1)|^2}\right)\right]_{t=t_1}, \tag{5.29}$$

where, we have used $x_i = 1 + 2t_i/(s - 4m^2)$. Now, using the Regge amplitude eq.(5.27), one obtains

$$D_Q\left(\rho_A^{(1)}\|\rho_A^{(2)}\right) \approx (\Delta t)^2 \left[\beta_2 + \alpha_2 \ln\left(\frac{s}{s_0}\right)\right] \xrightarrow{s \gg s_0} (\Delta t)^2 \alpha_2 \ln\left(\frac{s}{s_0}\right) > 0, \tag{5.30}$$

where the last inequality arises if $\alpha_2 > 0$. Note that, this expression is valid near $t_1 = 0$ i.e., near $x_1 = 1$. In this sense, we have a stronger expression in the Regge limit than from eq.(5.24). Consider the situation where we have Regge behaviour and the answer can be as large as what is allowed from eq.(5.24) when $x_1$ is away from 1. It is expected that as $x_1 \to 1$ we should get the behaviour in eq.(5.30). So we want the two behaviours to be "stitched" continuously in the transition to the limit $t \to 0 \iff x \to 1$. Now, suppose $\alpha_2 = 0$ then $D_Q \propto \beta_2 (\Delta t)^2 \approx \beta_2 s^2 (\Delta x)^2$ which would contradict the trend suggested by eq.(5.24) unless[11] $\beta_2 > 0$. Subsequently, if $\alpha_2 \neq 0$, we have from eq.(5.30) that $\alpha_2 > 0$ to avoid a discontinuous transition from the behaviour in eq.(5.24). This is so as if $\alpha_2 < 0$ then we can always choose a large enough $s$ such that $D_Q$ in eq.(5.30) will become negative (for $s \geq s_0 \exp[\beta_2/(-\alpha_2)]$). Thus, our analysis of relative entropy further bolsters the experimental observation of non-linear Regge trajectories by providing an explanation why $\alpha_2 > 0$ must be respected.

## 5.4 A new type of positivity

Here we will discuss a new type of positivity for $AA \to AA$ scattering in massive QFTs which appears to be valid at least in the high energy limit. Up to $O(\sigma^0)$ we have

$$D_Q\left(\rho_A^{(1)}(x_1)\|\rho_A^{(2)}(x_2)\right) = \sum_{n=2}^{\infty} \frac{(|x_1| - |x_2|)^n}{n!} \partial_x^n \ln|\mathcal{M}(s,x)|^2 \bigg|_{x=x_1}, \tag{5.31}$$

where we have used eq.(B.35) and expanded. If we write[12]

$$\ln|\mathcal{M}(s,x)|^2 = \sum_{\ell=0}^{\infty} (2\ell + 1)\mu_\ell(s) P_\ell(x), \tag{5.32}$$

---

[11]In the narrow resonance approximation, following [44], it can be shown that $\beta_2 > 0$ holds.

[12]To be rigorous, we should consider $\ln|\mathcal{M}(s,x)|^2/|\mathcal{M}(s,x_{min})|^2$ where $x_{min}$ is where $|\mathcal{M}(s,x_{min})|^2$ is minimized (we assume this is non-zero) so that $\mu_\ell(s)$ is implicitly dependent on $x_{min}$.

with $\ell$'s running over even integers (as LHS is an even function for identical particles) and *if* $\mu_\ell(s) \geq 0$ for $\ell > 0$, then using

$$\left. \partial_x^n P_\ell(x) \right|_{x=x_1} \leq \left. \partial_x^n P_\ell(x) \right|_{x=1}, \tag{5.33}$$

with $n \geq 2$ and where $x_1 \in (-1, 1)$, we have

$$\left. \partial_x^n \ln|\mathcal{M}(s, x)|^2 \right|_{x=x_1} \leq \left. \partial_x^n \ln|\mathcal{M}(s, x)|^2 \right|_{x=1}, \tag{5.34}$$

which leads to an interesting bounding behaviour, namely

$$\left. D_Q\left(\rho_A^{(1)} \| \rho_A^{(2)}\right) \right|_{x=x_1} \leq \left. D_Q\left(\rho_A^{(1)} \| \rho_A^{(2)}\right) \right|_{x=1}, \tag{5.35}$$

a feature verified by many of our plots. However, $\mu_\ell(s) > 0$ does not appear to follow from any known properties of the amplitude, does not appear to have a mention in the literature and is distinct from Martin positivity [46] (see Section 7.6).

In the high energy limit discussed above, we can use $|\mathcal{M}(s, x)|^2 \sim f(s)/(1 - x^2)^p$ with $p \geq 1/2$. Using this it is easy to verify[13] that $\mu_\ell(s) > 0$ for $\ell \geq 2$. This kind of positivity emerging in the high energy limit[14] is reminiscent of what happens in the conformal bootstrap [51, 52] where in the large conformal dimension limit, there is an underlying cyclic polytope picture for the CFT. Furthermore, we numerically checked the sign of $\mu_\ell(s), \ell \geq 2$ in the low energy regime and it turns out to be positive as well. This should have been anticipated keeping in mind our observations in fig.(4), where the maxima is clearly at $x = 1$ for fixed $s$ in every regime.

Lastly, we also checked the behaviour of $\mu_\ell(s)$ for the string amplitude and observed that positivity is guaranteed to be satisfied before we encounter any of the infinite poles that the amplitude has at integer values of $s$. However, it was also interestingly noted that the higher poles affected (changed the sign of) of $\mu_\ell(s)$ after a certain spin *i.e.* only for $\ell \geq \ell(s_n = n)$ only! In fig.(9), we have plotted the even spin $\mu_\ell(s)$ for the partial wave expansion,

$$\ln\left(\frac{|\mathcal{M}(s, x)|^2}{|\mathcal{M}(s, x_{min}(s))|^2}\right) = \sum_{\ell=0}^{\infty} (2\ell + 1)\mu_\ell(s, x_{min}(s))P_\ell(x). \tag{5.36}$$

However, the denominator inside the log is a constant *w.r.t* $x$. Hence, when we split the log as a difference (dimensionally taking care of each term inside the log by dividing with a constant $s_0 = 1 m^2$), it will only contribute to the $0^{th}$ partial wave $\mu_0(s, x_{min}(s))$. All the higher spin partial waves will therefore be independent of $x_{min}(s)$. Furthermore, in our positivity claim regarding $\mu_\ell(s)$, we are only concerned with $\ell \geq 2$ because of the derivatives present in eq.(5.31) and hence the claim is independent of $x_{min}(s)$. Therefore, for ease of calculation, we can effectively fix $x_{min}(s) = x_0$ to be anything we want for convenience.

We also noted that for small $s$, the partial waves are just a constant, $\mu_\ell(s) \approx 4/\ell(\ell + 1), \ell \geq 2$ and hence we will divide out this factor for ease of plotting all the partial waves on the same scale.

# 6  Hypothesis testing using relative entropy

So far, we have been considering two density matrices at two different angles, corresponding to the same theory with all other parameters the same. However, we can also consider two

---

[13] $\mu_0$ sign will not affect since it multiplies $P_0(x) = 1$.

[14] $\chi PT$ will not respect this positivity since it is an effective field theory and does not obey the high energy bound.

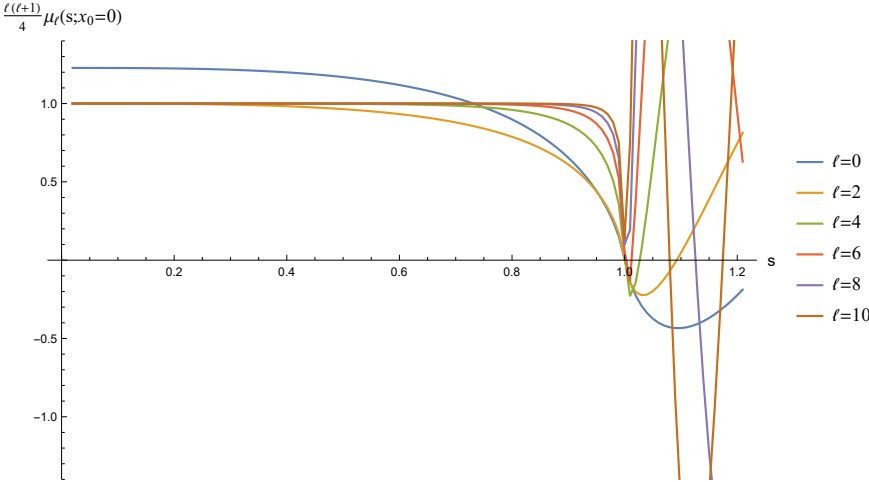

Figure 9: Behaviour of normalized $\mu_\ell(s)$ for even spins for the String amplitude with $x_0 = 0$. It can be seen that the $\ell = 2, 4$ violates positivity after encountering the first pole at $s = 1$ while the higher spins violate at higher $s$.

different density matrices, $\rho_1$ and $\rho_2$ where $\rho_2$ has been obtained from $\rho_1$ by varying the underlying parameters in the theory (which could be some couplings, mass parameters or even $s$ itself) by an infinitesimal amount, keeping the angle fixed. Let us examine what happens in this case.

We have in this situation

$$
\begin{aligned}
D(\rho_1\|\rho_2) &= \int_{-1}^1 dx \, P(c_i, x) \ln \frac{P(c_i, x)}{P(c_i + \Delta c_i, x)}, \\
&= -\int_{-1}^1 dx \, P_1 \left( \Delta c_i \frac{\partial_i P_1}{P_1} + \frac{(\Delta c_i)^2}{2} \left( \frac{\partial_i^2 P_1}{P_1} - \left( \frac{\partial_i P_1}{P_1} \right)^2 \right) + \cdots \right),
\end{aligned} \tag{6.1}
$$

where $\partial_i \equiv \partial_{c_i}$ and $P_1 \equiv P(c_i^1, x)$ for shorthand and $c_i$'s are parameters like coupling, mass etc or $s$. It is easy to see that terms like $\int_{-1}^1 dx \, \Delta c_{i_1} \cdots \Delta c_{i_k} \partial_{i_1} \cdots \partial_{i_k} P_1$ will vanish since we can pull out $\Delta c_{i_1} \cdots \Delta c_{i_k} \partial_{i_1} \cdots \partial_{i_k}$ out and the integral is just unity from normalization. Hence to leading order we have

$$
D(\rho_1\|\rho_2) = \frac{1}{2} \int_{-1}^1 dx \frac{(\partial_i P_1 \Delta c_i)^2}{P_1} = \frac{1}{2} \int_{-1}^1 dx P_1 \left( \Delta c_i \partial_i \left( \ln \left( \frac{P}{P_1} \right) \right) \Big|_1 \right)^2. \tag{6.2}
$$

Next, using eq.(3.7) and eq.(3.17) we expand $P$ occurring inside the ln up-to order $\sigma$. Subsequently, we expand in powers of $(x - x_1)$. In leading order, i.e. the $(x - x_1)^2$ term integrates to give $2\sigma$ (since $(x - x_1)$ integrated with the Gaussian in $P$ will just vanish). Hence we get something like

$$
D(\rho_1\|\rho_2) \approx \sigma \left( \Delta c_i \partial_i \partial_x \ln \left( \frac{|\mathcal{M}(c_i, x)|^2}{|\mathcal{M}(c_i^1, x_1)|^2} \right) \right)^2 \Bigg|_{x_1, c_i^1} + O(\sigma^2, \Delta c_i^3). \tag{6.3}
$$

Thus the distinguishability of two density matrices with slightly different parameters is governed by the above quantity.

**Example 1: $\phi^4$**

In the $\phi^4$ theory, let us consider the parameter to be $\lambda$. It is straightforward to check that in this case to leading order in the coupling, and near the threshold we have

$$
D(\rho_1\|\rho_2) \approx \frac{1}{921600 \, \pi^4} \sigma (\Delta\lambda)^2 x_1^2 (s - 4)^4, \tag{6.4}
$$

which is always positive as $\sigma > 0$.

**Example 2: String theory**

For the string case, let us consider the parameter to be $s$ and expand in small $s$. We find to leading order

$$D(\rho_1\|\rho_2) \approx 36\,\zeta(3)^2\sigma(\Delta s)^2 x_1^2 s^4. \tag{6.5}$$

Note that the leading answer is sensitive to the massive stringy modes. In the pure supergravity regime, the answer at this order vanishes.

## 6.1 Hypothesis testing using different theories

Now, as mentioned in the introduction, $\rho_1$ and $\rho_2$ could also be density matrices for different theories. For instance, imagine that the scattering was happening in massless $\lambda\phi^4$ theory, but we wanted to describe it using string theory. What is the relative entropy in this case? Here we will content ourselves with some numerical exploration. As can be seen in fig.(10), the relative entropy is comparatively low until the string amplitude encounters a zero since naively speaking that is where the string and the $\phi^4$ amplitude differ drastically. However, it would be wise to caution ourselves at this point since the relative entropy does not distinguish at the level of the amplitude; instead, it does so at the level of the probability density, $\mathcal{P}_g(x)$. This can be seen in the plot since even though the string amplitude differs from the $\phi^4$ amplitude by orders of magnitude, the relative entropy is really small for most of the range of $s$ values. However, near a zero or pole of the amplitude, the behaviour is carried over into the density function as well, and hence the relative entropy shows a sharp peak there.

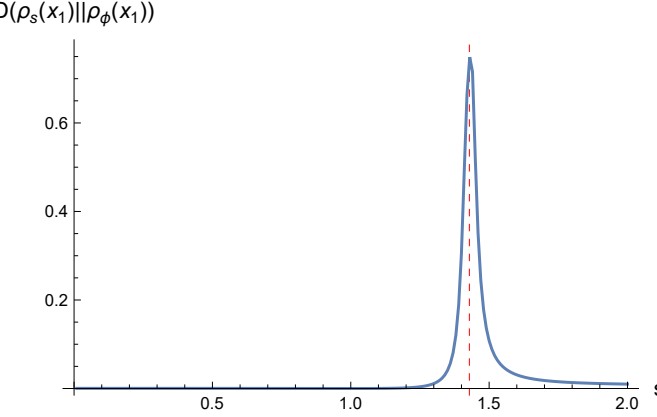

Figure 10: Plot of Relative entropy between the massless $\phi^4$ amplitude and the string theory amplitude at the common angle $x_1 = 0.4$ vs s. The red dotted line marks the zero of the string amplitude.

We will use hypothesis testing in a significant way to isolate interesting S-matrices in the context of the S-matrix bootstrap for pion scattering in the next section.

## 7 Constraining S-matrix bootstrap

In this section, we will consider pion scattering using the S-matrix bootstrap techniques discussed in [17]. We consider the $2 \to 2$ scattering of particles with $O(3)$ symmetry using the

ansatz

$$A(s|t,u) = \sum_{n \leq m}^{\infty} a_{nm} \left( \eta_t^m \eta_u^n + \eta_t^n \eta_u^m \right) + \sum_{n,m}^{\infty} b_{nm} \left( \eta_t^m + \eta_u^m \right) \eta_s^n \,, \tag{7.1}$$

where

$$\eta_s = \frac{\left( \sqrt{4 - \frac{4}{3}} - \sqrt{4 - s} \right)}{\left( \sqrt{4 - \frac{4}{3}} + \sqrt{4 - s} \right)} \,,$$

and the amplitude, $\mathcal{M}$ is defined similar to eq.(D.2). In this case, crossing symmetry becomes $A(s|t,u) = A(s|u,t)$, which the ansatz satisfies trivially. The partial wave unitarity condition, $\left| S_\ell^{(I)}(s) \right|^2 \leq 1$ is imposed using SDPB [13] for a grid of $s$ values similar to [16]. Here $I$ denotes the isospin channel such that partial waves and the amplitude[15] are related by the expression

$$\mathcal{M}(s,t,u) = 16 \, i \, \pi \frac{\sqrt{s}}{\sqrt{s-4}} \sum_{I=0,1,2} \mathbb{P}_I \sum_{\ell=0} (2\ell+1) \left( 1 - S_\ell^{(I)}(s) \right) P_\ell \left( x = \frac{u-t}{u+t} \right). \tag{7.2}$$

Subsequently, SDPB extremizes a linear combination of parameters and gives us the corresponding maximal S-matrix. Since this a numerical venture, we need a cutoff for the infinities occurring in the summation in eq.(7.1). Hence we restrict our $n$ and $m$ in our ansatz to have cutoff $N_{max}$ and only consider partial waves upto a finite spin, $L_{max}$. To specialize further for pions, constraints of resonance and Adler zeros were used. $\rho$–resonance was imposed as a zero of the $\ell = 0$ partial wave of the anti-symmetric channel as

$$S_1^{(1)}(m_\rho^2) = 0 \,, \tag{7.3}$$

where $m_\rho = 5.5 + 0.5 \, i$.

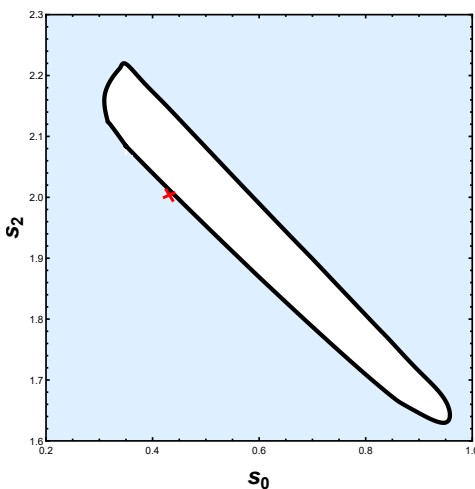

Figure 11: Disallowed region in the space of $s_0$ and $s_2$ where both $s_0$ and $s_2$ cannot be imposed. The cross marked separately is the 1–loop position of the Adler zeroes from best fit $\chi$PT amplitude.

Leading order chiral perturbation theory predicts the presence of Adler zeros. This can be easily seen (at least at tree level) in eq.(4.10). Adler zeros are actually the zeros of the amplitude when the 4-momenta of an external massless Goldstone-boson goes to 0 under a critical assumption that there are no poles due to other parts of the diagram [53] at zero 4-momenta of the Goldstone bosons. Pions are approximate Goldstones and not exactly massless. They

---

[15]See Appendix (D) for the details

also do not interact through a stable particle, thus for 2-2 scattering in CoM frame, they do not have poles below the threshold. Thus, pion amplitudes have Adler zeros. However, it requires one of the external pions to go off-shell (4-momenta to be 0) despite being an external particle, hence making $s + t + u = 4m^2$ no longer true (for details, see [18]). In that case, the zero is found exactly at $s = t = u = m^2$. However, it is non-trivial to locate the Adler zeros non-perturbatively in the $s$-plane when $s + t + u = 4m^2$ holds everywhere. Thus, bootstrap methods become really handy to find the allowed regions of Adler zeros for the partial waves. The $T_\ell^{(I)}(s) = \sqrt{\frac{s}{s-4}} \frac{S_\ell^{(I)}(s)-1}{2i}$ is defined such that the identical case unitarity condition $\text{Im}\left(T_\ell^{(I)}(s)\right) \geq 2\sqrt{\frac{s-4}{s}} \left|T_\ell^{(I)}(s)\right|^2$ is satisfied. Plotting $T_0^{(0)}(s)$ and $T_0^{(2)}(s)$ as a function of $s$ in the unphysical region $0 < s < 4$ gives us the location of Adler zeros $s_0$ and $s_2$. At tree level they are simply $s_0^{(0)} = 0.5, s_0^{(2)} = 2$ and at 1-loop they become $s_0^{(0)} = 0.437, s_0^{(2)} = 2.003$. In general, they can be written down as

$$T_0^{(0)}\left(s_0^{(0)}\right) = 0 \text{ and } T_0^{(2)}\left(s_0^{(2)}\right) = 0. \tag{7.4}$$

The next step is to find all pairs of $(s_0, s_2)$ (for ease of notation we will just refer the the zeroes as $s_0, s_2$ keeping in mind that they are zeroes of the $0^{th}$ partial wave) that can be imposed in the ansatz. This can be done by imposing the Adler zero $s_0$ and extremizing the value of $T_0^{(2)}(s)$ for values of $s$ in $(0, 4)$. If a particular $T_0^{(2)}(s)$ has positive maximum and negative minimum then we can impose Adler zero $s_2 = s$, else the pair $(s_0, s_2)$ is not allowed. Upon repeating the steps for values of $s_0$, it is discovered that the allowed region is a closed area, which is known as the lake as shown in fig.(11).

In order to determine which values of these extremal matrices are closer to the physical region, scattering lengths are required which are found through

$$\text{Re}\left[T_\ell^{(I)}(k)\right] = k^{2\ell}\left[a_\ell^{(I)} + b_\ell^{(I)}k^2 + \mathcal{O}(k^2)\right], \tag{7.5}$$

where $k = \sqrt{\frac{s}{4} - 1}$. Here $a_\ell^{(I)}$ is the $I$'th isospin, spin-$\ell$ scattering length. $\ell = 0, 1, 2$ are the S,P,D-wave scattering lengths respectively. $b_\ell^{(I)}$'s are called the effective ranges. Note that this definition of the scattering length differs with the one given in eq.(4.21) by just an inverse factor of $32\pi (2\ell)!$ and in this section, we will be referring to this definition only. The main scattering lengths used to distinguish are $a_0^{(0)}, a_0^{(2)}$ and $a_1^{(1)}$ which have the experimental values $0.2196 \pm 0.0034, -0.0444 \pm 0.0012$ and $0.038 \pm 0.002$ respectively. Upon plotting the lake boundary in the $a_0^{(0)}, a_0^{(2)}, a_1^{(1)}$ space it is found that the lower boundary, more notably the left side of the lower boundary is closer to the physical region. More details can be found in [17].

We now look at eq.(3.19). For small $\Delta x$, the sign of $C(s, x_1) = \frac{1}{2}(\mathcal{M}''|_{x_1} - (\mathcal{M}'|_{x_1})^2)$ determines whether the Quantum relative entropy $D_Q$ is monotonically increasing or not. We wish to check the sign of $C$ as a function of $x_1$ as we move around the lake. We will avoid the point $x_1 = 0$ for the $\pi_0 + \pi_0 \to \pi_0 + \pi_0$ since this causes unnecessary complications in the form of the $D_Q$. All the remaining cases have been shown to be the same.

We consider the same three reactions as in eq.(4.14). We use the $\mathcal{M}$ defined in Appendix (D) in order to plot $C(s, x_1)$ around the lake. Armed with the observations of Section (5), we now know which reactions are monotonically increasing $D_Q$ and which are not. We will use[16] $N_{max} = 10$ and $L_{max} = 11$ for the plots but we have checked that none of the features we find change significantly when $N_{max}$ is increased to 14 and $L_{max} = 17$.

---

[16]We apologise but computing resources available to us during the time of Covid-19 was not ideal.

## 7.1 Lake plots

The monotonicity described in the Section (5) is near threshold. Hence we start our checks with values very close to 4, say around $s \approx 4.0001$ and then increase. We observed that until $s = 4.1$ the nature of the plots remains unchanged, only the values shift. Hence without loss of generality, we choose to see the behaviour of the lake at $s = 4.01$. As expected, different reactions have different effects on the lake. The sample $x_1$ behaviour for two points of $\pi^0 + \pi^0 \to \pi^0 + \pi^0$ reaction is given by fig.(12). For $x_1 > 0$, some of the points are monoton-

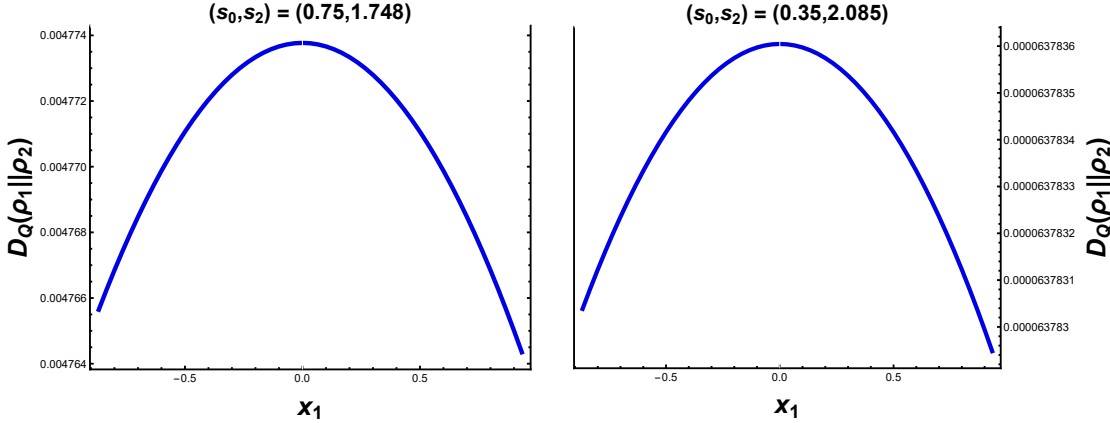

Figure 12: For $\pi^0 + \pi^0 \to \pi^0 + \pi^0$. S-matrices around the lake show a similar behaviour for all reactions

ically increasing with $x_1$ and some are monotonically decreasing, as seen in fig.(12). It will be interesting to explore this behaviour. Now, compiling for all points around the lake, we get the results of fig.(13). It is important to note that $\pi^+ + \pi^+ \to \pi^+ + \pi^+$ has $D_Q \leq 0$ as given in eq.(5.4).

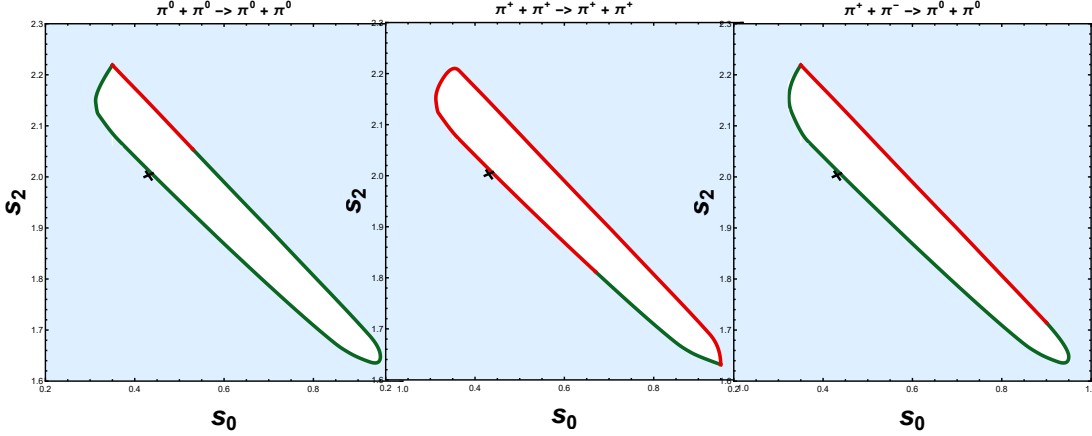

Figure 13: For s=4.01. Green points respect $\chi$PT and red do not

Now combining the disallowed regions of the three reactions, we can rule out a significantly large portion of the lake boundary as given in fig.(14). This suggests that the lake boundary can be theoretically increased by constraining the ansatz to respect eq.(5.10) (with $a_2^{(2)} \geq 0$) and eq.(5.11).

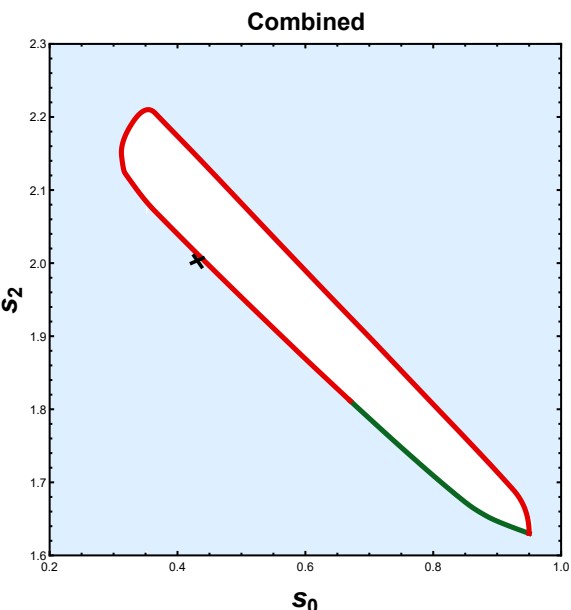

Figure 14: s=4.01 green points respect $\chi PT$ and red do not

## 7.2 New constraints: The River

As discussed in the previous section, we intend to increase the lake boundary to make the points satisfy $\chi$PT constraints. To that effect, we are lucky, since the scattering length constraints of Section(5) are linear in our ansatz parameters and hence can be easily imposed. We summarize the additional constraints that we imposed along with unitarity to find the new allowed region:

$$a_2^{(0)} + 2\,a_2^{(2)} \geq 0\,,\; a_2^{(0)} - a_2^{(2)} \geq 0\,,\; a_2^{(2)} \geq 0\,,$$
$$a_0^{(0)} + 2\,a_0^{(2)} \geq 0\,,\; 2a_0^{(0)} + a_0^{(2)} \geq 0\,,\; a_0^{(2)} \leq 0\,,\; a_0^{(0)} - a_0^{(2)} \geq 0\,. \tag{7.6}$$

We do not assume anything about the P-wave scattering length. The new allowed region is given by the fig.(15). As mentioned in the introduction, we shall call this figure, "The River". The fact that we could rule out a large portion of previously allowed regions without any phenomenological input (except resonance) is remarkable!

Of note is the fact that all the constraints were theoretically motivated. The sign of the spin-2 scattering lengths being fixed by dispersion relations and the spin-0 ones from the $\chi PT$ Lagrangian perturbatively. Note that the spin-0 constraints are more powerful then spin-2 constraints. Imposing the D-wave constraints alone results in just a larger version of the lake with the upper boundary shifted upwards in comparison to the lake. The Adler zeroes corresponding to 0–loop $\chi PT$ $(0.5, 2)$ lie outside the river while the 1–loop $(0.437, 2.003)$ lies approximately on the upper bank. The 2-loop point $(0.4195, 2.008)$ lies inside the river.

## 7.3 Hypothesis testing in S-matrix bootstrap

We aim to find theories which are close to $\chi PT$ along the river banks. By close, we mean that at least the S- and P-wave scattering lengths must be comparable for such theories. Relative entropy provides a measure of distance in theory space. Using the formalism of Section(6) we calculate $D(\rho_1 \| \rho_2)$. Since bootstrap S-matrices are non-perturbative they shall be considered as the physical theory and $\chi PT$ S-matrix will be considered as an approximation. Hence $\rho_2$ comes from $\chi PT$, while $\rho_1$ is calculated using the boundary theories of the "River". Note that the distance will be calculated separately for each reaction.

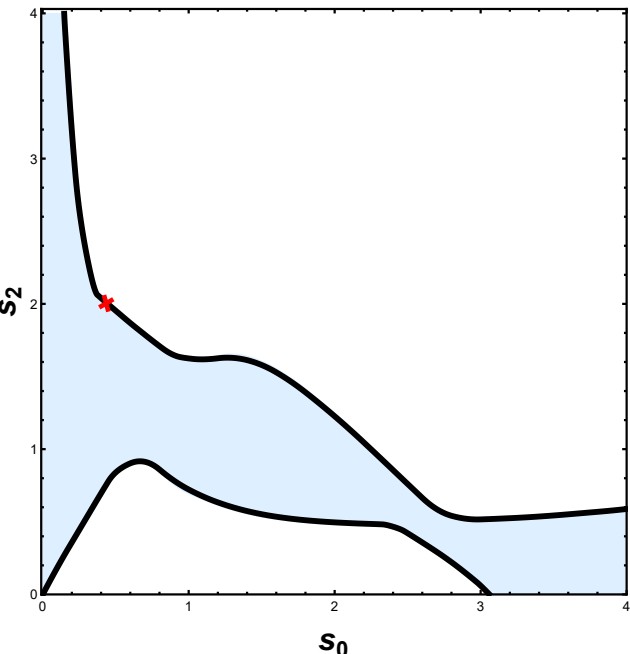

Figure 15: New allowed region (shaded in blue) after imposing the constraints. Note that the 1-loop $\chi PT$ lives close to the "kink" in the upper bank.

This gives us a set of values. Now we must set up a rule to consider some theories and discard others. We want to allow as many theories as possible discarding only those who are manifestly distant. The following set of rules seem reasonable to us:

1. We shall consider sequence of theories with violations of up to 5 orders more than the minimum violation.

2. While sequentially looking at theories with increasing violations of up to order 5, if one finds that there is no theory with a violation at an intermediate order, then all theories with greater violations will be discarded.

Using these rules we get the allowed regions for various reactions in fig.(16). We use $s = 4.01$ and $x_1 = 1/3$ for this analysis. However, this behaviour will remain unchanged for all $x_1 \in (-1,1)$ and $s \in (4, 4.15]$. By demand, we are close to $s = 4$ since we want to be most sensitive to the S- and P-wave scattering lengths and not the effective ranges. The validity of $\chi PT$ does not depend on the initial and final states therefore only those points who are "close" for all reactions can serve as candidates of a theory close to $\chi PT$. Hence, taking the intersection of these allowed regions we get the green regions in fig.(1).

As shown in fig.(1), compared to [17], the green regions lie near the top and bottom portions of the so-called peninsula. The distances of some sample points on the physical regions are given in table (1). Except for $\pi^+ + \pi^0 \to \pi^+ + \pi^0$ the lower physical region does remarkably well for other reactions. This should mean that this region is close to $\chi$PT. Looking at fig.(1) we can see that the peninsula boundary is very close to the lower bank physical region. Hence, it can indeed be considered "close" to $\chi PT$ in terms of scattering lengths $a_0^{(0)}, a_0^{(2)}$ and also $a_2^{(2)}$. This implies that hypothesis testing indeed works and can be considered a reliable measure for comparing theories. This is remarkable because the theories being compared can have very different amplitudes. In our case, one is perturbative from an effective action while the other is in a crossing symmetric basis following analyticity! Nonetheless, when we compare the two using relative entropy and minimise their "distance", we somehow end up with the

Table 1: Distances of some boundary theories from $\chi$PT. Here ijkl means $\pi^i + \pi^j \rightarrow \pi^k + \pi^l$. The order of the values are broadly regulated by the choice of $\sigma\,(10^{-6})$, $s - 4\,(10^{-2})$ and $X_1(1/3)$

| Distance using relative entropy | | | | | |
|---|---|---|---|---|---|
| $(s_0,s_2)$ | 0000 | +-+- | +0+0 | ++++ | +-00 |
| ( 0.46 , 1.987 ) (U) | $7.02 \times 10^{-16}$ | $1.753 \times 10^{-12}$ | $8.98 \times 10^{-12}$ | $7.62 \times 10^{-20}$ | $1.37 \times 10^{-16}$ |
| ( 0.57 , 1.893 ) (U) | $5.82 \times 10^{-16}$ | $1.58 \times 10^{-12}$ | $1.27 \times 10^{-12}$ | $5.16 \times 10^{-20}$ | $8.87 \times 10^{-17}$ |
| ( 0.67 , 1.893 ) (U) | $4.68 \times 10^{-16}$ | $1.23 \times 10^{-12}$ | $1.85 \times 10^{-13}$ | $1.87 \times 10^{-19}$ | $6.02 \times 10^{-17}$ |
| ( 0.70 , 1.787 ) (U) | $4.57 \times 10^{-16}$ | $1.06 \times 10^{-12}$ | $2.87 \times 10^{-11}$ | $3.36 \times 10^{-18}$ | $4.52 \times 10^{-17}$ |
| ( 0.73 , 1.763 ) (U) | $4.67 \times 10^{-16}$ | $8.68 \times 10^{-13}$ | $8.70 \times 10^{-16}$ | $4.97 \times 10^{-20}$ | $4.31 \times 10^{-17}$ |
| ( 2.70 , 0.293 ) (L) | $7.27 \times 10^{-18}$ | $1.58 \times 10^{-10}$ | $9.72 \times 10^{-12}$ | $2.50 \times 10^{-20}$ | $5.62 \times 10^{-17}$ |
| ( 2.80 , 0.222 ) (L) | $3.12 \times 10^{-19}$ | $8.65 \times 10^{-11}$ | $1.56 \times 10^{-11}$ | $1.64 \times 10^{-19}$ | $2.20 \times 10^{-17}$ |
| ( 2.85 , 0.184 ) (L) | $1.90 \times 10^{-18}$ | $5.45 \times 10^{-11}$ | $1.95 \times 10^{-11}$ | $2.78 \times 10^{-19}$ | $2.35 \times 10^{-17}$ |
| ( 2.90 , 0.145 ) (L) | $2.08 \times 10^{-17}$ | $2.21 \times 10^{-11}$ | $2.60 \times 10^{-11}$ | $5.25 \times 10^{-19}$ | $3.27 \times 10^{-18}$ |
| ( 2.95 , 0.104 ) (L) | $1.33 \times 10^{-17}$ | $1.464 \times 10^{-12}$ | $3.07 \times 10^{-11}$ | $9.75 \times 10^{-19}$ | $6.66 \times 10^{-19}$ |

similar values of physical observables like the scattering lengths. To emphasise, we were *not* imposing the values of these scattering lengths, instead only the correct signs were imposed which had theoretical motivations behind them.

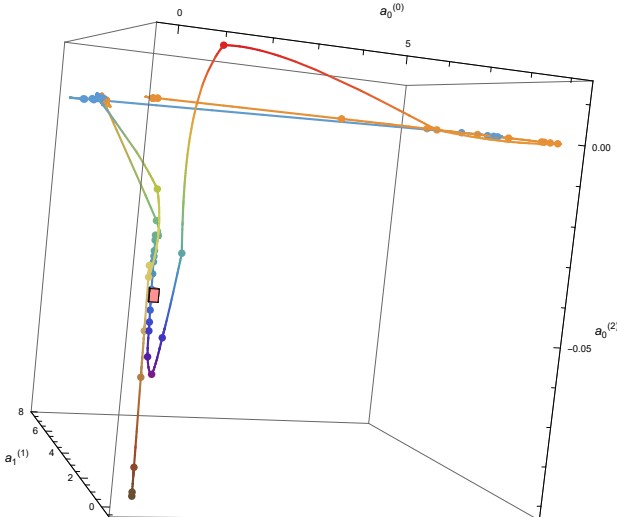

Figure 17: Bluish-purple points denote the upper bank and the yellowish-brown points describe the lower boundary. The experimental value are given by the pink box. Points close to the box are less distant from $\chi PT$ in terms of hypothesis testing.

## 7.4 Comparison of scattering lengths and effective ranges

Here we will check the values of scattering lengths for some points of the physical regions and compare them to their experimental value given in the table (2). Using table (3), we see that apart from $b_0^{(2)}$, the matrices take on a similar range of values in both the physical regions. We chose a value of $s \approx 4$ so that we were sensitive to the differences in the S- and P-wave scattering lengths, which turn out to be in the same range in the upper and lower banks in table (3). This is still intriguing since the Adler zeroes corresponding to $\chi PT$ are far away

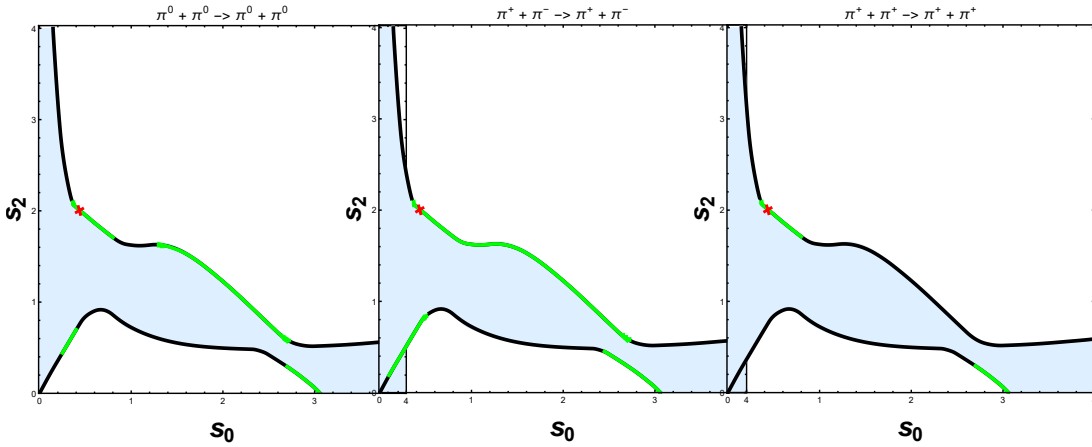

Figure 16: Allowed region for different reactions. Remaining reactions have the same profile.

from the lower boundary. We can also conclude from fig.(17) that these are the only regions with $\ell = 0$ and $\ell = 1$ scattering lengths close to the experimental values. This confirms our hypothesis testing observations in the previous section. While the S- and P- wavelengths are roughly comparable with experiments, the $a_2^{(2)}, a_2^{(0)}$ values are around an order of magnitude bigger. Furthermore, no single point agrees with all the experimental values–this is indicative of the fact that the actual phenomenological point from [38] is inside the allowed region and not on the boundary where the comparison is being made.

## 7.5 Elastic Unitarity–Preliminary findings

The unitarity was imposed by the condition $\left|S_\ell^{(I)}(s)\right|^2 \leq 1$. What interests us in this section is the elastic unitarity condition, $\left|S_\ell^{(I)}(s)\right|^2 = 1$ which has to hold between $4 < s < 16$ since there is no particle production in this energy. Ideally, we would like to impose this as a constraint. However, the framework of SDPB does not allow (as far as we have checked) imposition of elastic unitarity since the constraints are quadratic in the free parameters of the ansatz. Hence, we restrict ourselves to numerical checks of the available S-matrices for now. To check elastic unitarity of a S-matrix, we find the deviation from unitarity $1 - \left|S_\ell^{(I)}(s)\right|^2$ for $\ell \in 0, 1$ and $I \in 0, 1, 2$. If the violations for all channels and the first two spins are below a set tolerance, then the point shall be included or else, it shall be rejected. We shall choose a liberal tolerance of 12% i.e., the absolute values needs to be greater than 0.88. Upon doing this for all points along the two river banks, we find the fig.(18). This seems to discard a portion of the lower boundary. However, $N_{max} = 16$ considerations reduce the violation s.t the 12% criteria now allows the lower bank physical region. Hence one perhaps needs to consider the rate of change of maximal violations with $N_{max}$ before eliminating regions.

Elastic unitarity is still a work in progress and we shall report on it later–very recently there appeared [54] which provides some promising ways to incorporate elastic unitarity in the numerics. Our preliminary findings would disfavour the points on the lower bank in fig.(1) and it will be gratifying to confirm this using the more rigorous proposals in [54].

Table 2: Experimental values of scattering lengths and effective ranges taken from [38].

| Experimental values | | | |
|---|---|---|---|
| **Observable** | **Value** | **Error** | **Units** |
| $a_0^{(0)}$ | 0.220 | 0.005 | 1 |
| $a_0^{(2)}$ | -0.0444 | 0.0010 | 1 |
| $a_1^{(1)}$ | 0.0379 | 0.0005 | $m_\pi^{-2}$ |
| $a_2^{(0)}$ | 0.00175 | 0.00003 | $m_\pi^{-4}$ |
| $a_2^{(2)}$ | 0.000170 | 0.000013 | $m_\pi^{-4}$ |
| $b_0^{(0)}$ | 0.276 | 0.006 | $m_\pi^{-2}$ |
| $b_0^{(2)}$ | -0.0803 | 0.0012 | $m_\pi^{-2}$ |

Table 3: Values of scattering lengths and effective ranges on the upper(U) bank and lower(L) bank for $N_{max}$ = 12 and $L_{max}$ = 21. The values for the S- and P-wave scattering lengths differ from $N_{max}$ = 10 and $L_{max}$ = 11 at most in the second significant figure.

| Scattering lengths and Effective Ranges | | | | | | | |
|---|---|---|---|---|---|---|---|
| $(s_0,s_2)$ | $a_0^{(0)}$ | $a_0^{(2)}$ | $a_2^{(0)}$ | $a_2^{(2)}$ | $a_1^{(1)}$ | $b_0^{(0)}$ | $b_0^{(2)}$ |
| ( 0.46 , 1.989 ) (U) | 0.135 | -0.031 | 0.047 | $\approx 0$ | 0.032 | 0.131 | -0.055 |
| ( 0.57 , 1.895 ) (U) | 0.155 | -0.038 | 0.041 | $\approx 0$ | 0.035 | 0.155 | -0.065 |
| ( 0.67 , 1.813 ) (U) | 0.169 | -0.044 | 0.041 | $\approx 0$ | 0.038 | 0.168 | -0.071 |
| ( 0.80 , 1.710 ) (U) | 0.219 | -0.056 | 0.097 | 0.029 | 0.045 | 0.196 | -0.087 |
| ( 2.70 , 0.290 ) (L) | 0.070 | -0.035 | 0.031 | 0.0014 | 0.040 | 0.185 | -7.36 |
| ( 2.80 , 0.219 ) (L) | 0.098 | -0.049 | 0.025 | 0.0004 | 0.0457 | 0.371 | -8.40 |
| ( 2.85 , 0.181 ) (L) | 0.117 | -0.058 | 0.025 | 0.0007 | 0.0480 | 0.482 | -8.58 |
| ( 2.95 , 0.101 ) (L) | 0.153 | -0.076 | 0.024 | 0.0019 | 0.0491 | 0.746 | -8.39 |

## 7.6 Positivity–Preliminary findings

Using positivity of the amplitude in the so-called Mandelstam triangle (reviewed below) to constrain theories is an old idea–see, e.g. [55–57]. In this section, we will consider using positivity[17] in the extended Mandelstam region, following the discussion in [22]. Starting with the generalization of eq.(5.7)

$$\frac{\partial^{2n}}{\partial s^{2n}}\left(\mathcal{M}^{(I)}(s,t)\right) = \frac{(2n)!}{\pi}\sum_J\left(\int_{4m^2}^\infty ds'\left(\frac{\delta^{IJ}}{(s'-s)^{2n+1}} + \frac{C_{su}^{IJ}}{(s'-u)^{2n+1}}\right)\text{Im}[\mathcal{M}^{(J)}(s'+i\epsilon,t)]\right).$$

$$(7.7)$$

---

[17]It is difficult to conclude anything definitive about the positivity in the sense discussed in Section (5.2) since for high values of $s$ we will have more partial waves contributing. A preliminary study reveals that for very large $s$, indeed $\mu_\ell > 0$ but we will not attribute any significance to this finding with the relatively low number of partial wave spins we have incorporated in this present work.

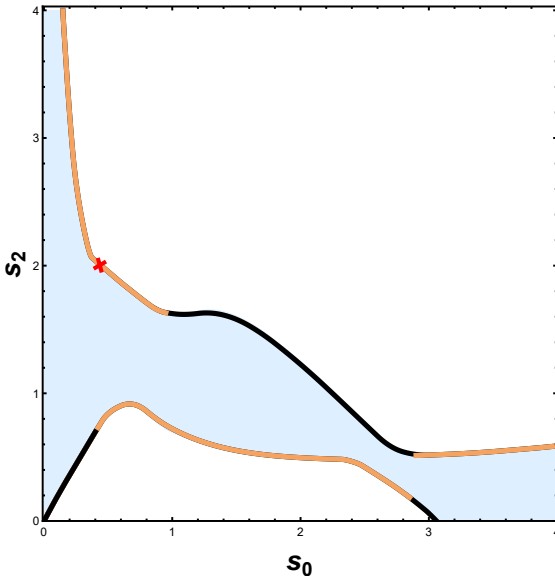

Figure 18: The brown regions are the theories satisfying the elastic unitarity constraint up to the tolerance indicated in the main text.

Now in the region $s < 4m^2, s + t > 0$ the denominators $(s' - s)^{2n+1}, (s' - u)^{2n+1}$ are positive. As reviewed in [22], crossing implies that the amplitude is analytic inside $s, t, u < 4m^2$, so that inside $s, t < 4m^2, s + t > 0$, the amplitude is real. Furthermore, the Legendre polynomials in the partial wave expansion, $P_\ell \left(1 + \frac{2t}{s'-4m^2}\right) > 0$ for $s' > 4m^2$ and $t > 0$. So together we define the extended Mandelstam region [22]: $s, t < 4m^2, t > 0, s + t > 0$ shown in the fig.(19) below. Considering linear combinations in the LHS, $\sum_I \alpha_I \mathcal{M}^{(I)}$ such that for the integrand in the RHS, we have $\sum_{IJ} \alpha_I C_{su}^{IJ} \text{Im}[\mathcal{M}^{(J)}(s' + i\epsilon, t)] = \sum_J \beta_J \text{Im}[\mathcal{M}^{(J)}(s' + i\epsilon, t)]$ with $\beta_J > 0$, arguments based on the optical theorem lead to the positivity conditions

$$\frac{\partial^{2n}}{\partial s^{2n}} \left(\mathcal{M}^{\pi^0 \pi^0 \to \pi^0 \pi^0}(s, t)\right) \geq 0, \tag{7.8}$$

$$\frac{\partial^{2n}}{\partial s^{2n}} \left(\mathcal{M}^{\pi^+ \pi^0 \to \pi^+ \pi^0}(s, t)\right) \geq 0, \tag{7.9}$$

$$\frac{\partial^{2n}}{\partial s^{2n}} \left(\mathcal{M}^{\pi^+ \pi^+ \to \pi^+ \pi^+}(s, t)\right) \geq 0, \tag{7.10}$$

where $n \geq 1$ and $(s, t)$ are inside the blue and red regions shown in fig.(19).

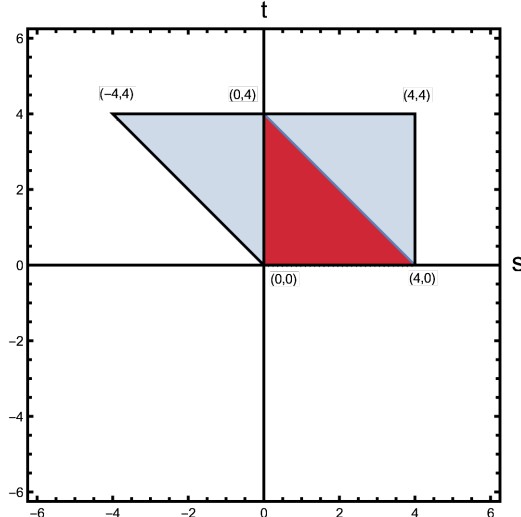

Figure 19: The extended Mandelstam region. The red region is the Mandelstam triangle $0 \leq s, t, u \leq 4m^2$. The blue region is the extended region satisfying $s, t < 4m^2, t > 0, s + t > 0$.

These constraints can be used to further constrain the river. We choose to impose the positivity constraints close to points $(-4, 4), (4, 0), (0, 0)$ and $(4, 4)$. No violations were observed along the river boundary for the point $(0, 4)$. Hence, we imposed constraints eq.(7.8), eq.(7.10), eq.(7.9) for these 4 edges upto $n = 4$ and re-evaluated the river. The "new river" is given by fig.(20). These constraints were found to be satisfied for $n = 5$ along the new river banks, thus implying convergence with $n$. It is quite intriguing that the shape of the river changes even though we have imposed unitarity. This happens since we are imposing unitarity only for a grid of s-values and upto a maximum spin $L_{max}$. Interestingly the maximal violations result from the points in the extended region outside the mandelstam triangle. This leads us to wonder which subset of all the conditions we have considered so far will lead to the fastest numerics–we will leave this for future work.

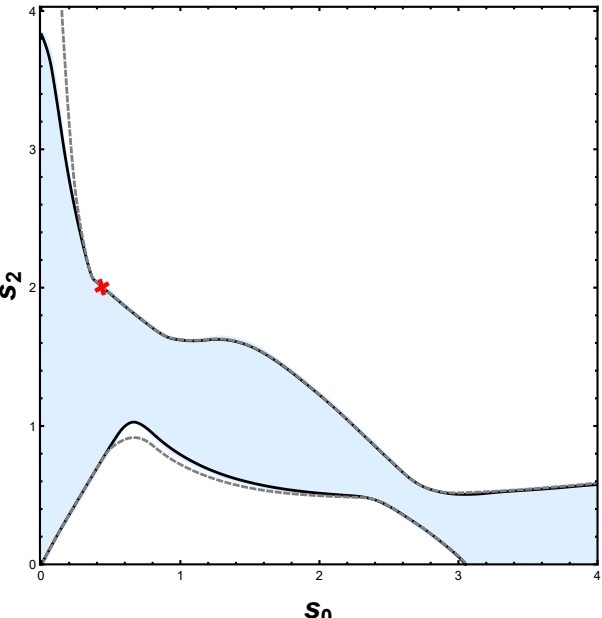

Figure 20: New river upon imposing positivity for $N_{max} = 10$, $L_{max} = 11$. Dashed gray line denotes the old river boundary and joined black line gives the new river boundary.

# 8 More Entanglement Measures

We have mainly explored entanglement due to S-Matrix evolution in momentum space. However, since the state space for momentum degree of freedom is infinite, we had to deal with various kinds of infinities. It would have been nice if we could explore the entanglement due to S-Matrix evolution for finite Hilbert space. For pion scattering, we have such an opportunity indeed. We can consider entanglement in isospin caused by S-Matrix evolution. The isospin state space for pion is three dimensional. Thus, the entanglement measures will not be plagued with the infinities of the likes of those encountered in momentum space. We will consider two such entanglement measures for analysing entanglement in isospin: the usual quantum relative entropy and *entanglement power* of S-matrix.

## 8.1 Quantum Relative Entropy

To define an isospin analogue we shall fix the the momentum of final states and allow sum over the $O(N)$ indices. If we consider the initial state,

$$\lvert p\hat{z}, a_1; -p\hat{z}, a_2 \rangle \,, \tag{8.1}$$

then the analogue of 2.4 for the isospin case will be,

$$\lvert f \rangle = (2\pi)^4 \delta^{(4)}(0) \sum_{b_1, b_2} \lvert q\hat{n}, b_1; -q\hat{n}, b_2 \rangle \, \mathcal{M}^{b_1 b_2}_{a_1 a_2}(s, \cos\theta) \,, \tag{8.2}$$

where $\hat{n}$ is a fixed direction $(\sin\theta, 0, \cos\theta)$. Using this one can calculate the final state density matrix $\rho_f$. Tracing over isopin states we get the reduced density matrix,

$$\rho_1 = \sum_{b_1} \sum_{c_1} \frac{\sum_\chi \left( \mathcal{M}^{b_1 \chi}_{a_1 a_2}(s, \cos\theta) \mathcal{M}^{a_1 a_2}_{c_1 \chi}(s, \cos\theta)^* \right)}{\sum_{x,y} \mathcal{M}^{xy}_{a_1 a_2}(s, \cos\theta) \left[ \mathcal{M}^{a_1 a_2}_{xy}(s, \cos\theta) \right]^*} \, \lvert q\hat{n}, b_1 \rangle \langle q\hat{n}, c_1 \rvert \,. \tag{8.3}$$

If we redefine everything in terms of the following matrix in the isospin basis

$$M_{ij}(s, x; a_1, a_2) = \mathcal{M}^{ij}_{a_1 a_2}(s, x) \,, \tag{8.4}$$

then we see that the reduced density matrix takes on a simple form of (removing the dependence on the initial $a_1, a_2$ for brevity)

$$\rho_1 = \frac{1}{\text{Tr}(M.M^\dagger)} M(s, x).M^\dagger(s, x) \,. \tag{8.5}$$

Now, to calculate the trace of $\rho_1^n$, needed for the replica trick, we need to diagonalize $\rho_1$. To do that, we see that we just need to diagonalize the matrix $M(s, x)$. In the invariant decomposition, we have

$$M_{ij}(s, x) = A(s, t(x), u(x))\delta_{a_1, a_2}\delta_{i,j} + A(t(x), u(x), s)\delta_{i,a_1}\delta_{j,a_2} + A(u(x), s, t(x))\delta_{i,a_2}\delta_{j,a_1} \,. \tag{8.6}$$

We see that for $a_1 = a_2$, $M$ is already in the diagonal form with eigenvalues

$$\lambda_M(s, x) \in \{A(s, t(x), u(x)), A(s, t(x), u(x)), A(s, t(x), u(x)) + A(t(x), u(x), s) + A(u(x), s, t(x))\} \,, \tag{8.7}$$

which gives $M.M^\dagger$ automatically a diagonal form with eigenvalues $\lvert\lambda_M\rvert^2$. Whereas, for $a_1 \neq a_2$, we have that

$$
\begin{aligned}
M_{ik}M^\dagger_{kj} &= (A(t(x), u(x), s)\delta_{i,a_1}\delta_{k,a_2} + A(u(x), s, t(x))\delta_{i,a_2}\delta_{k,a_1}). \\
&\quad (A^*(t(x), u(x), s)\delta_{j,a_1}\delta_{k,a_2} + A^*(u(x), s, t(x))\delta_{j,a_2}\delta_{k,a_1}) \\
&= \lvert A(t(x), u(x), s)\rvert^2 \delta_{i,a_1}\delta_{j,a_1} + \lvert A(u(x), s, u(x))\rvert^2 \delta_{i,a_2}\delta_{j,a_2} \,,
\end{aligned}
\tag{8.8}
$$

which is diagonalized! Hence, we have in general that (using obvious shorthand notation)

$$\text{Tr}((M.M^\dagger)^n) = \begin{cases} 2(|A(s)|^2)^n + (|A(s) + A(t(x)) + A(u(x))|^2)^n & \text{if } a_1 = a_2, \\ (|A(t(x))|^2)^n + (|A(u(x))|^2)^n & \text{if } a_1 \neq a_2 \end{cases} \tag{8.9}$$

and $\text{Tr}(\rho_1)^n = \frac{1}{(\text{Tr}(M.M^\dagger))^n} \text{Tr}((M.M^\dagger)^n)$ can be directly found from eq.(8.9). Differentiating this *w.r.t n* is straightforward in the replica trick to get the entanglement entropy in the isospin space of the form

$$S_{EE,I}(s,x) = -\sum_{i=1}^{3} \frac{1}{\sum_{j=1}^{3} |\lambda_{M,j}|^2} |\lambda_{M,i}|^2 \ln\left(\frac{1}{\sum_{j=1}^{3} |\lambda_{M,j}|^2} |\lambda_{M,i}|^2\right). \tag{8.10}$$

with the appropriate eigenvalues as found above.

For final states at angles $\theta, \theta' \neq 0$ we can define two reduced density matrices $\rho_1$ and $\rho_2$ and ultimately define the relative entropy $D_I(\rho_1\|\rho_2)$ using the replica trick.

Table 4: Isospin relative entropy of $\pi_0 + \pi_0 \rightarrow all$ for S-matrices on the river (for $N_{max}$ = 10). Labels 1 and 2 refer to the choice of parameters $(x_1, x_2) = (0.999, 0.001)$ and $(x_1, x_2) = (0.999, 0.99)$ respectively (where $x_i$ refers to $\cos(\theta_i)$). Labels U and L refer to Upper and Lower boundary respectively.

| $S_{EE,I}$ | | | | |
|---|---|---|---|---|
| $(s_0, s_2)$ | $s = 4.01\mathbf{(1)}$ | $s = 30\mathbf{(1)}$ | $s = 4.01\mathbf{(2)}$ | $s = 30\mathbf{(2)}$ |
| (0.35,2.08) (U) | $2.74\,10^{-11}$ | 0.0162 | $8.82\,10^{-15}$ | $4.41\,10^{-6}$ |
| (1.3,1.637) (U) | $4.03\,10^{-13}$ | 0.551 | $1.29\,10^{-16}$ | $4.7\,10^{-5}$ |
| (0.9,0.775) (L) | $4.69\,10^{-11}$ | 0.485 | $1.50\,10^{-14}$ | $2.1\,10^{-4}$ |
| (2.85,0.775) (L) | $2.46\,10^{-11}$ | 0.816 | $7.91\,10^{-15}$ | $1.1\,10^{-5}$ |

## 8.2 Entanglement Power

Next, we consider another quantity called *Entanglement Power* [47–50]. It deals directly with the S-matrix of the system rather than depending on the initial states. The initial state is defined as,

$$|\psi_i\rangle := \hat{R}(\Omega_1) \otimes \hat{R}(\Omega_2) |p\hat{z}, a_1; -p\hat{z}, a_2\rangle, \tag{8.11}$$

where $\hat{R}(\Omega_i)$ is the rotation operator in the isospin space acting on the $i^{th}$ particle. Since we are only operating on the isospin indices, we can consider writing, $|k\hat{n}, a_1; -k\hat{n}, a_2\rangle \equiv |a_1\rangle \otimes |a_2\rangle$. Now, the target final state can be constructed by taking restriction of the S-matrix to 2-particle isospin space in a specific way. The details are chalked out in appendix E.2. Following that, we can obtain a target final state

$$|\psi_f\rangle = \frac{1}{w(p)} (2\pi)^4 \delta^{(4)}(0) \sum_{\substack{c_1,c_2 \\ b_1,b_2}} |p\hat{n}, c_1; -p\hat{n}, c_2\rangle \, \mathcal{S}_{b_1 b_2}^{c_1 c_2}(s, \cos\theta) \, \langle b_1 | \hat{R}(\Omega_1) | a_1 \rangle \, \langle b_2 | \hat{R}(\Omega_2) | a_2 \rangle, \tag{8.12}$$

using which we can define the final state density matrix $\rho_{\psi_f}$ with matrix elements in the basis, $\{|q\hat{n}, c_1; -q\hat{n}, c_2\rangle\}$ are given by,

$$\left(\rho_{\psi_f}\right)^{b_1 b_2}_{c_1 c_2}(s, \cos\theta) = \tag{8.13}$$

$$\frac{\sum_{x_1,x_2}\sum_{y_1,y_2} \mathcal{M}^{b_1 b_2}_{x_1 x_2}(s,\cos\theta)\left[\mathcal{M}^{y_1 y_2}_{c_1 c_2}(s,\cos\theta)\right]^* \hat{R}(1)^{b_1}_{a_1}\hat{R}(2)^{b_2}_{a_2}\,(\hat{R}(1)^{c_1}_{a_1}\hat{R}(2)^{c_2}_{a_2})^*}{\sum_{z_1 z_2}\sum_{x_1,x_2}\sum_{y_1,y_2} \mathcal{M}^{z_1 z_2}_{x_1 x_2}(s,\cos\theta)\left[\mathcal{M}^{y_1 y_2}_{z_1 z_2}(s,\cos\theta)\right]^* \hat{R}(1)^{z_1}_{a_1}\hat{R}(2)^{z_2}_{a_2}\,(\hat{R}(1)^{y_1}_{a_1}\hat{R}(2)^{y_2}_{a_2})^*},$$

$$\tag{8.14}$$

where, $\hat{R}(1)^a_b = \langle a|\hat{R}(\Omega_1)|b\rangle$. Similar to the previous subsection, we can define the reduced density matrix

$$\bar{\rho}_1 := \mathrm{tr}_2\,\rho_{\psi_f}\,. \tag{8.15}$$

Now, the entanglement power of the S-Matrix is given by,

$$\mathcal{E} = 1 - \int \frac{d\Omega_1}{4\pi}\frac{d\Omega_2}{4\pi}\mathrm{tr}_1\left[\bar{\rho}_1^2\right], \quad d\Omega_i := \sin\theta_i d\theta_i d\phi_i\,. \tag{8.16}$$

Table 5: Entanglement Power ($\mathcal{E}$) for S-matrices on the river. Labels U and L refer to Upper and Lower boundary respectively. These data points are obtained with $N_{max} = 10$.

| $\mathcal{E}$ | | | | |
|---|---|---|---|---|
| $(s_0, s_2)$ | $s = 4.01$ | $s = 10$ | $s = 30$ | $s = 100$ |
| (0.25,2.60) (U) | 0.60 | 0.30 | 0.37 | 0.295 |
| (0.35,2.08) (U) | 0.533 | 0.371 | 0.498 | 0.39 |
| (0.42,2.02) (U) | 0.527 | 0.369 | 0.463 | 0.46 |
| (1.3,1.637) (U) | 0.66 | 0.56 | 0.404 | 0.296 |
| (0.9,0.775) (L) | 0.52 | 0.33 | 0.37 | 0.27 |
| (2.85,0.775) (L) | 0.42 | 0.49 | 0.39 | 0.31 |

# 9   Future directions

In this paper, we have initiated investigations into the role of relative entropy in scattering. We have studied several standard quantum field theories and have also motivated the role that relative entropy can play in the recent revival of the S-matrix bootstrap. In the context of S-matrix bootstrap, we asked how to distinguish theories living on the boundary of allowed regions provided by numerics. We also made some preliminary studies of additional constraints which could shrink the space of allowed S-matrices. We will now conclude with a short road map for the future.

- In eq.(1.2) and related expressions, the quantity of interest that appears is $\ln\sigma'_{el}(x)$ and its second derivative. Curiously, when experimentalists model the diffraction peak in scattering, they introduce a slope parameter which involves precisely this quantity and

its first derivative [58][18]–the second derivative is presumably related to the curvature of the diffraction peak. As such this formula appears to be quite suited for experimental investigations in the future.

- We did not consider monotonicity of relative entropy in the sense used in the quantum information literature. There could a potential connection between this and renormalization group flows, and it will be interesting to examine this in the context of scattering. Other fundamental inequalities like strong subadditivity may also be useful to examine in the context of scattering and put to experimental tests using collider data. Note that monotonicity of relative entropy played a major role in deriving the averaged null energy condition (ANEC) in recent times [59, 60].

- It would have been desirable to find a situation where some measure like entanglement entropy/relative entropy would extremize near interesting physical theories. This then could become a powerful selection criterion for studying the space of allowed S-matrices. Our findings in this direction are not very conclusive, but we do hope to return to this in the future. In some vague sense, we would then have some "quantum thermodynamic" selection rule in the space of S-matrices.

- Apropos our bootstrap findings, we would like to improve our numerics and extract the spectrum of resonances in the two different regions we have identified in this paper. The main question here is: What are the extra ingredients we need to put in, such that the allowed S-matrices zooms into the standard model? More intriguingly, could there be a different allowed theory which matches with experimental results so far which would be disconnected from $\chi PT$?

- While we made some tentative observations about positivity (see Section (5.4)) as well as finding that positivity in the sense used in Section (7.6) can constrain the space of allowed S-matrices, we did not make any concrete statements connecting up our work with the positivity of amplitudes or the positivity of the underlying geometry in [51, 52, 61–66]. It will be interesting to investigate this in the future.

- We started a preliminary exploration of entanglement in isospin degrees of freedom. We explored two quantum information theoretic measures in this regard, quantum relative entropy and entanglement power. The isospin state space being finite-dimensional these measures do not suffer from the divergences coming from momentum space integrals. We presented results of numerical explorations with $N_{max}$ = 10 in this work. Our preliminary studies with higher $N_{max}$s [67] indicated a sharp drop in the entanglement power $\mathcal{E}$ near the kink. *This may hint at a quantum information theoretic selection principle at play, and it would be fascinating to understand this.*

- It will be important to extend our analysis to external particles with spin. We have given a formal setup for this purpose in Appendix (C.2). We leave the detailed study for external spinning particles in the spirit of this work for future endeavour.

- In the context of AdS/CFT correspondence relative entropy led to the derivation of linearised Einstein's equations while positivity of relative entropy constrained non-linear perturbations (e.g. [73, 74]). It may also be worthwhile to use relative entropy to study scattering in AdS space–via the connection in Mellin space, this will then constrain the dual CFTs. The high energy bounds like the Froissart bound have been studied recently in [75] and it may be interesting to study the analogous bounds for relative entropy in AdS scattering.

---

[18]See eq.(5.1.5).

# Acknowledgements

We acknowledge useful discussions with B. Ananthanarayan, Biplob Bhattacharjee, Yu-tin Huang, Ashoke Sen and Sasha Zhiboedov. We thank Andrea Guerrieri for providing us useful tips regarding the numerics and Joao Penedones for useful comments on v1 of the manuscript. We are especially thankful to Prasad Hegde and Sudhir Vemapti for enabling us to remotely use their computing clusters in CHEP, IISc and to Urbasi Sinha and her team (especially Sourav Chatterjee) for allowing us to remotely use two powerful workstations in her lab in RRI during the difficult times of Covid-19, which enabled us to complete this work. A.S. acknowledges support from a DST Swarnajayanti Fellowship Award DST/SJF/PSA-01/2013-14.

# A  Scattering fundamentals

In this appendix, we provide with a brief review of scattering theory in relativistic quantum field theory in $\tilde{D} + 1$ dimensional Minkowski space-time $\mathbb{M}^{\tilde{D},1}$.

## A.1  S-matrix basics

We consider scattering of two scalar massive particles $A, B$ with masses $m_A, m_B$ respectively. In the most general case we have,

$$A + B \rightarrow A + B + X, \tag{A.1}$$

where $A, B$ are single particles and $X$ is some multi-particle product or may be some bound state as well. We have the so called "elastic channel": $A + B \rightarrow A + B$ and the "inelastic channel": $A + B \rightarrow X$.

In scattering, one focuses attention in asymptotic past and asymptotic future where one assumes that the particles are free. Our initial and target final states will be such free states. Suppose there is some initial state $|i\rangle$ then the final state $|f\rangle$ is given by the S-matrix, $\mathcal{S}$,

$$|f\rangle = \mathcal{S} |i\rangle . \tag{A.2}$$

The S-matrix has the following structure:

$$\mathcal{S} = \mathbb{1} + i\mathcal{T} . \tag{A.3}$$

Here $\mathbb{1}$ accounts for no interaction and $\mathcal{T}$ captures the interaction. This is also called the *transfer matrix* The matrix element of the $S$–matrix between two momentum states $|i\rangle$ and $|f\rangle$ can be written as

$$\langle i | \mathcal{S} | f \rangle = \delta^{(\tilde{D}+1)} \left( \sum p_i - \sum p_f \right) \langle i | \mathbf{s} | f \rangle , \tag{A.4}$$

where $\sum p_i$ denotes the total $\tilde{D} + 1$–momentum in the state $|i\rangle$ and similarly for $\sum p_f$[19] and $\langle i | \mathbf{s} | f \rangle$ is the amplitude. Note that, the conservation of the four momentum is enforced by the presence of the delta function and in the amplitude itself this conservation constraint is already applied.

In scattering theory, however, of more importance is the matrix element of $\mathcal{T}$ because, this amplitude is precisely the one capturing the scattering procedure. In fact, the corresponding matrix element can be written as

$$\langle i | \mathcal{T} | f \rangle = \delta^{(\tilde{D}+1)} \left( \sum p_i - \sum p_f \right) \mathcal{M}(i \rightarrow f) , \tag{A.5}$$

---

[19]From now on, Minkowski momenta will be denoted without any arrow and spatial momenta will be denoted along with arrows.

where, $\mathcal{M}(i \to f)$ is the amplitude of the transition $i \to f$. It is this amplitude which is calculated using Feynman graphs in perturbative quantum field theory. Note that, if we insist on taking $|i\rangle \neq |f\rangle$ i.e., we *exclude* the possibility of *no scattering* then, evidently, one has

$$\langle i | \mathbf{s} | f \rangle = i \mathcal{M}(i \to f). \tag{A.6}$$

We will use this relation later in our analysis.

## A.2 Hilbert space for incoming and outgoing states

Now, let us specialize to elastic channel *i.e.*, we consider that the final state with the particles $A$ and $B$. For scattering problems, we usually scatter off particles with definite momenta in the asymptotic past and also looks for outgoing particles with definite momenta in the asymptotic future. In these two extreme limits, these particles are assumed to be non-interacting. In these limits, we will consider the momentum states to be residing in a Fock space. We are considering the scalar bosons. Let $|\Omega\rangle$ be the bosonic Fock vacuum which is normalized to unity *i.e.*,

$$\langle \Omega | \Omega \rangle = 1. \tag{A.7}$$

Then the single particle state with definite momentum is defined by,

$$|\vec{k}_i\rangle := \sqrt{2E_{\vec{k}_i}}\, \mathrm{a}^\dagger(\vec{k}_i)|\Omega\rangle, \tag{A.8}$$

where $i$ stands for $A, B$ in the present case. Here $\mathrm{a}_i^\dagger(\vec{k})$ is the corresponding creation operator and $E_{\vec{k}_i}$ is the usual relativistic energy of a free particle

$$E_{\vec{k}_i} = \sqrt{k_i^2 + m_i^2}, \tag{A.9}$$

with $k_i = |\vec{k}_i|$. The bosonic Fock space creation-annihilation operators satisfy the usual algebra with respect to commutators

$$[\mathrm{a}(\vec{k}_i), \mathrm{a}^\dagger(\vec{l}_j)] = \delta_{ij}\, \delta^{(\tilde{D})}(\vec{k}_i - \vec{l}_j), \tag{A.10}$$

$$[\mathrm{a}(\vec{k}_i), \mathrm{a}(\vec{l}_j)] = 0, \tag{A.11}$$

$$[\mathrm{a}^\dagger(\vec{k}_i), \mathrm{a}^\dagger(\vec{l}_j)] = 0. \tag{A.12}$$

With this algebra, we have the following inner product between the single particle states in eq.(A.8),

$$\langle \vec{k}_i | \vec{l}_j \rangle = \sqrt{2E_{\vec{k}_i} 2E_{\vec{l}_j}}\, \delta_{ij}\, \delta^{(\tilde{D})}(\vec{k}_i - \vec{l}_j). \tag{A.13}$$

Now, we move onto two-particle state. A generic two-particle state as a member of the above Fock space is given by,

$$|\vec{k}_i, \vec{l}_j\rangle = \sqrt{2E_{\vec{k}_i} 2E_{\vec{l}_j}}\, \mathrm{a}^\dagger(\vec{k}_i)\mathrm{a}^\dagger(\vec{l}_j)|\Omega\rangle. \tag{A.14}$$

Note that, when $i = j$ then we have identical bosons. The state has the obvious symmetrization property by virtue of the Fock algebra, eq.(A.10)-eq.(A.12). The inner product between two such two-particle state is given by

$$\langle \vec{p}_i, \vec{q}_j | \vec{k}_m, \vec{l}_n \rangle = 4\sqrt{E_{\vec{k}_m} E_{\vec{l}_n} E_{\vec{p}_i} E_{\vec{q}_j}} \tag{A.15}$$

$$\times \Big[ \delta_{im}\delta_{jn}\delta^{(\tilde{D})}(\vec{p}_i - \vec{k}_m)\delta^{(\tilde{D})}(\vec{q}_j - \vec{l}_n) + \delta_{in}\delta_{jm}\delta^{(\tilde{D})}(\vec{p}_i - \vec{l}_n)\delta^{(\tilde{D})}(\vec{q}_j - \vec{k}_m) \Big]. \tag{A.16}$$

Now, comes a very important point. There exists an *inner-product preserving isomorphism* between the subspace of the two-particle states in the Fock space with the symmetrized tensor product of two copies of the space of single particle Fock states. Let $\mathcal{V}^{(1)}$ be the space of single particle Fock states $\{|\vec{k}_i\rangle\}$[20] and $\mathcal{V}^{(2)}$ be the space of two-particle Fock states $\{|\vec{k}_i,\vec{l}_j\rangle\}$. Then,

$$\mathcal{V}^{(2)} \cong \left(\mathcal{V}^{(1)} \otimes \mathcal{V}^{(1)}\right)_S, \tag{A.17}$$

where, the subscript $S$ denotes symmetrization. In fact this is a *natural isomorphism*. Furthermore, this isomorphism can be made to preserve the inner product, eq.(A.15). Let $\mathcal{F}$ be the corresponding isomorphism i.e., $\mathcal{F} : \mathcal{V}^{(2)} \to \left(\mathcal{V}^{(1)} \otimes \mathcal{V}^{(1)}\right)_S$. Then,

$$\mathcal{F}(|\vec{k}_i,\vec{l}_j\rangle) = \frac{1}{\sqrt{2}}\left[|\vec{k}_i\rangle \otimes |\vec{l}_j\rangle + |\vec{l}_j\rangle \otimes |\vec{k}_i\rangle\right]. \tag{A.18}$$

It is easy to verify that the inner-product, eq.(A.15), is preserved under this isomorphism. There exist also the dual isomorphisms which exists between the corresponding dual spaces. Note that the symmetrization leads to a slightly different Hilbert Space structure than the one mentioned in previous section.

The reason for explicitly pointing out this isomorphism is that we will make use of this when we are interested in the explicit product space structure of the two-particle Hilbert space for the outgoing particles. One such instance is while taking partial traces of the density matrix of the joint system.

## B   Details of quantum entanglement measures

### B.1   Entanglement entropy

In this appendix, we provide the detailed calculations leading to the expression for the entanglement entropy, eq.(3.8). In particular, we give the detailed evaluation of the integral,

$$\mathcal{I}_E := \int_{-1}^{1} dx\, \mathcal{P}_g(x) \ln \mathcal{P}_g(x), \tag{B.1}$$

with $\mathcal{P}_g$ given by eq.(3.7) and eq.(3.5). We can divide the above integral in two pieces,

$$\mathcal{I}_E = \int_{-1}^{1} dx\, \delta_\sigma(x-y)\mathcal{M}_y(s,x) \ln\left[\delta_\sigma(x-y)\right] + \int_{-1}^{1} dx\, \delta_\sigma(x-y)\mathcal{M}_y(s,x) \ln\left[\mathcal{M}_y(s,x)\right], \tag{B.2}$$

where we have defined,

$$\mathcal{M}_y(s,x) := \frac{|\mathcal{M}(s,x)|^2}{\int_{-1}^{1} dx\, g_y(x)|\mathcal{M}(s,x)|^2} \equiv \frac{|\mathcal{M}(s,x)|^2}{\mathcal{I}_g(s,y)}. \tag{B.3}$$

Now, we are forced to do the second integral under the approximation that $\delta_\sigma(x-y) \to \delta(x-y)$ since otherwise the exact integral is of the form of a Gaussian times the log of a function following which we are pretty much stuck. Now, since we have used the limiting form of $\delta_\sigma(x-y)$ in the numerator, we should do the same with the denominator *i.e.* with $\mathcal{I}_g(s,y) \to |\mathcal{M}(s,y)|^2$ as well leaving us with

$$\int_{-1}^{1} dx\, \delta_\sigma(x-y)\mathcal{M}_y(s,x) \ln\left[\mathcal{M}_y(s,x)\right] \approx \frac{|\mathcal{M}(s,y)|^2}{\mathcal{I}_g(s,y)} \ln\left(\frac{|\mathcal{M}(s,y)|^2}{\mathcal{I}_g(s,y)}\right) \approx 0. \tag{B.4}$$

---

[20]Here $i$ includes both $A$ and $B$ in our case.

Thus we have essentially,

$$\mathcal{I}_E \approx \int_{-1}^{1} dx \, \delta_\sigma(x-y)\mathcal{M}_y(s,x)\ln\left[\delta_\sigma(x-y)\right]. \tag{B.5}$$

This integral will be of central importance in many analyses that follow. Thus we will evaluate this integral with gory details. Then,

$$\mathcal{I}_E = \frac{1}{2\sqrt{\pi\sigma}}\int_{-1}^{1} dx \, e^{-\frac{(x-y)^2}{4\sigma}}\mathcal{M}_y(s,x)\ln\left(\frac{1}{2\sqrt{\pi\sigma}}e^{-\frac{(x-y)^2}{4\sigma}}\right)$$
$$\equiv -\left[\frac{1}{2}\mathcal{I}_{E,1} + \ln(2\sqrt{\pi\sigma})\mathcal{I}_{E,2}\right], \tag{B.6}$$

where

$$\mathcal{I}_{E,1} := \int_{-1}^{1} dx \left(\frac{1}{2\sqrt{\pi\sigma}}e^{-\frac{(x-y)^2}{4\sigma}}\right)\mathcal{M}_y(s,x)\left[\frac{x-y}{\sqrt{2\sigma}}\right]^2, \tag{B.7}$$

$$\mathcal{I}_{E,2} := \int_{-1}^{1} dx \left(\frac{1}{2\sqrt{\pi\sigma}}e^{-\frac{(x-y)^2}{4\sigma}}\right)\mathcal{M}_y(s,x). \tag{B.8}$$

We see immediately that,

$$\mathcal{I}_{E,2} = \frac{1}{\mathcal{I}_g(s,y)}\int_{-1}^{1} g(x)|\mathcal{M}(s,x)|^2 = 1. \tag{B.9}$$

Now, we focus upon $\mathcal{I}_{E,1}$. To do the integral, first introduce a change of variable as

$$u = \frac{x-y}{\sqrt{2\sigma}} \implies x = \sqrt{2\sigma}\,u + y. \tag{B.10}$$

Following this change of variable, we can write

$$\mathcal{I}_{E,1} = \frac{1}{4\sqrt{\pi\sigma^3}}\int_{-\frac{1+u}{\sqrt{2\sigma}}}^{\frac{1-u}{\sqrt{2\sigma}}}\sqrt{2\sigma}\,du\,e^{-\frac{u^2}{2}}\mathcal{M}_y(s,x)2\sigma u^2 \approx \int_{-\infty}^{\infty} du\left(\frac{e^{-\frac{u^2}{2}}}{\sqrt{2\pi}}\right)\mathcal{M}_y(s,x)u^2, \tag{B.11}$$

where, we have used the approximation that, since $\sigma \ll 1$, the limits of the integration can be effectively extended to $-\infty$ and $\infty$ because, anyway the integrand is extremely suppressed there due to the Gaussian. It is to be noted that, this can ONLY be done when $-1 < y < 1$, i.e., $y \neq -1, 1$. We will justify this in further details later on. Now, using the partial wave expansion of the amplitude, we can write

$$|\mathcal{M}(s,x)|^2 = \sum_{L=0}^{\infty}\mathcal{M}_L(s)x^L, \tag{B.12}$$

where, we have expanded the Legendre polynomials and have rearranged the infinite sum [21]. Substituting this back into eq.(B.11), we get

$$\mathcal{I}_{E,1} = \sum_{L=0}^{\infty}\frac{\mathcal{M}_L(s)}{\mathcal{I}_g(s,y)}\int_{-\infty}^{\infty} Du\left(\frac{e^{-\frac{u^2}{2}}}{\sqrt{2\pi}}\right)(\sqrt{2\sigma}u+y)^L u^2$$
$$= \sum_{L=0}^{\infty}\sum_{i=0}^{L}\frac{\mathcal{M}_L(s)}{\mathcal{I}_g(s,y)}\binom{L}{i}(\sqrt{2\sigma})^i y^{L-i}\left(\int_{-\infty}^{\infty} du\left(\frac{e^{-\frac{u^2}{2}}}{\sqrt{2\pi}}\right)u^{2+i}\right). \tag{B.13}$$

---

[21]This is possible since this series is absolutely and in fact uniformly convergent inside the Lehmann Ellipse and we are certainly inside the Lehmann Ellipse since we are considering physical $s$ and $x$

Define by $< u^i >_{0,1}$ the $i^{th}$ moment of the Gaussian(0,1) distribution. This has the following simple form:

$$< u^i >_{0,1} = \begin{cases} 0 & \text{if } i \text{ odd} \\ (i-1)!! & \text{if } i \text{ even} \end{cases}, \tag{B.14}$$

where $i!! = 1.3.5...(n-2).n$ for odd $i$ and $i!! = 2.4.6...(n-2).n$ for even $i$.

Now, assuming the series in eq.(B.13) converges, we can switch the order of the sum as

$$\sum_{L=0}^{\infty} \sum_{i=0}^{L} (...) = \sum_{i=0}^{\infty} \sum_{L=i}^{\infty} (...).$$

Furthermore, we also make the observation that

$$\sum_{L=i}^{\infty} \binom{L}{i} \mathcal{M}_L(s) y^{L-i} = \sum_{L=i}^{\infty} \frac{L!}{i!(L-i)!} \mathcal{M}_L(s) y^{L-i}$$
$$= \frac{1}{i!} \frac{\partial^i}{\partial x^i} \left( \sum_{L=i}^{\infty} \mathcal{M}_L(s) x^L \right) \Bigg|_{x=y} \tag{B.15}$$
$$= \frac{1}{i!} \frac{\partial^i}{\partial x^i} (|\mathcal{M}(s,x)|^2) \Bigg|_{x=y},$$

where, we have used $\frac{\partial^i}{\partial x^i} (\sum_{L=0}^{i-1} \mathcal{M}_L(s) x^L) = 0$ in the final step. Therefore, we can simplify $\mathcal{I}_{E,1}$ to

$$\mathcal{I}_{E,1} = \frac{1}{\mathcal{I}_g(s,y)} \sum_{i=0}^{\infty} < u^{i+2} >_{0,1} (\sqrt{2\sigma})^i \left( \sum_{L=i}^{\infty} \binom{L}{i} \mathcal{M}_L(s) y^{L-i} \right)$$
$$= \sum_{i=0,2,4...}^{\infty} < u^{i+2} >_{0,1} \frac{(\sqrt{2\sigma})^i}{i!} \frac{\partial^i}{\partial x^i} (\mathcal{M}_y(s,x)) \Bigg|_y. \tag{B.16}$$

Upon further using eq.(B.14), we get

$$\mathcal{I}_{E,1} = \frac{1}{\mathcal{I}_g(s,y)} \sum_{i=0}^{\infty} (2i+1) \frac{\partial^{2i}}{\partial x^{2i}} (|\mathcal{M}(s,x)|^2) \Bigg|_y \frac{\sigma^i}{i!}. \tag{B.17}$$

Similarly, we can show that, the integral in the denominator has a similar form,

$$\mathcal{I}_g(s,y) = \sum_{i=0}^{\infty} \frac{\partial^{2i}}{\partial x^{2i}} (|\mathcal{M}(s,x)|^2) \Bigg|_y \frac{\sigma^i}{i!}, \tag{B.18}$$

which allows us to simplify eq.(B.17) as

$$\mathcal{I}_{E,1} = 1 + \frac{2\sigma}{\mathcal{I}_g(s,y)} \frac{\partial^2}{\partial x^2} (\mathcal{I}_g(s,x)) \Bigg|_y, \tag{B.19}$$

where, we have used

$$\sum_{i=0}^{\infty} i \frac{\partial^{2i}}{\partial x^{2i}} (|\mathcal{M}(s,x)|^2) \Bigg|_y \frac{\sigma^i}{i!} = \sigma \sum_{i=0}^{\infty} \frac{\partial^{2i+2}}{\partial x^{2i+2}} (|\mathcal{M}(s,x)|^2) \Bigg|_y \frac{\sigma^i}{i!} = \sigma \frac{\partial^2}{\partial x^2} (\mathcal{I}_g(s,x)) \Bigg|_y. \tag{B.20}$$

Let us now try to justify our expansion a bit more by going back to the individual terms appearing in eq.(B.16) (which we can generalize to the later sections as well). The $i^{th}$ order

term consists of the $< u^{i+2} >_{0,1}$, $i$ even. However, if we go back to eq.(B.11), we will see that the actual term, if calculated exactly, would have been something like

$$I_i = \int_{-\frac{1+y}{\sqrt{2\sigma}}}^{\frac{(1-y)}{\sqrt{2\sigma}}} du \left( \frac{1}{\sqrt{2\pi}} e^{-\frac{u^2}{2}} \right) u^{i+2}. \tag{B.21}$$

Without loss of generality, let us choose $0 < y < 1$. This would imply that

$$\int_{-\frac{1-y}{\sqrt{2\sigma}}}^{\frac{(1-y)}{\sqrt{2\sigma}}} du \left( \frac{1}{\sqrt{2\pi}} e^{-\frac{u^2}{2}} \right) u^{i+2} < I_i < \int_{-\frac{1+y}{\sqrt{2\sigma}}}^{\frac{(1+y)}{\sqrt{2\sigma}}} du \left( \frac{1}{\sqrt{2\pi}} e^{-\frac{u^2}{2}} \right) u^{i+2} \le (< u^{i+2} >_{0,1}). \tag{B.22}$$

Now, the definite integral of the Gaussian up-to a finite range is a very well documented function, the so called "Error Function", $Erf(z)$ defined s.t.

$$Erf\left( \frac{z}{\sqrt{2}} \right) = \int_{-z}^{z} du \left( \frac{1}{\sqrt{2\pi}} e^{-\frac{u^2}{2}} \right). \tag{B.23}$$

A similar expression for the exact moment integrals is as follows:

$$\int_{-z}^{z} du \left( \frac{1}{\sqrt{2\pi}} e^{-\frac{u^2}{2}} \right) u^{2i} = < u^{2i} >_{0,1} \left( Erf\left( \frac{z}{\sqrt{2}} \right) - \frac{2}{\pi} e^{-\frac{z^2}{2}} z \sum_{j=0}^{i-1} \frac{z^{2j}}{(2j+1)!!} \right). \tag{B.24}$$

This can be understood by the fact that the peak of the integrand gets shifted more and more to the right with the increasing power of $u$ (in fact, the peak corresponding to the $2i^{th}$ moment is at $\sqrt{2i}$). Hence, the integration has to be carried out in a larger range to get the same accuracy (approximately, till the peak+$N\sigma$ where $N\sigma$ will give the desired accuracy in the base integral of the Gaussian). Therefore, the higher the moment integral, the larger we have to choose $z = (1-y)/\sqrt{2\sigma} \implies$ smaller $\sigma$ for a fixed $y$ ( A Similar logic is valid for $-1 < y < 0$ as well).

However, our approximation of the sum in eq.(B.16) is saved because each higher moment integral is further suppressed by a factor of $\sigma^{i/2}$ (assuming the derivatives of the amplitude are well-behaved functions and do not vary greatly). Therefore, a reasonably small $\sigma$ keeping in mind the accuracy of the integral up-to a finite order of the moment integrals (In our case, we can safely do that by just considering the leading term) is enough to guarantee a good level of accuracy of our approximation of the total sum over all the integrals. One can reverse the logic on its head in the sense that for a given $\sigma$ small, we can find a range of $y$ for which our approximation is valid. This is done simply by finding the $z_0$ such that the leading term in the series has the desired accuracy (this has to be done numerically on a computer). Then we have the following range of valid values of $y$ for our approximation,

$$\left| \frac{(1-y)}{\sqrt{2\sigma}} \right| > \frac{z_0}{\sqrt{2}} \implies -1 + z_0\sqrt{\sigma} < y < 1 - z_0\sqrt{\sigma}. \tag{B.25}$$

Coming back to our simplified expressions, combining eq.(B.17) and eq.(B.18), we have that

$$-\mathcal{I}_E = \ln(2\sqrt{\pi\sigma}) + \frac{1}{2} + \sigma \frac{\partial^2}{\partial x^2} \left( \frac{\mathcal{I}_g(s,x)}{\mathcal{I}_g(s,y)} \right) \bigg|_y,$$

$$= \ln(2\sqrt{\pi\sigma}) + \frac{1}{2} + \frac{1}{\sum_{i=0}^{\infty} \frac{\sigma^i}{i!} \frac{\partial^{2i}}{\partial x^{2i}} (|\mathcal{M}(s,x)|^2)\big|_y} \left( \sum_{i=0}^{\infty} i \frac{\sigma^i}{i!} \frac{\partial^{2i}}{\partial x^{2i}} (|\mathcal{M}(s,x)|^2) \bigg|_y \right) \tag{B.26}$$

$$\xrightarrow{\sigma \to 0}, \ \ln(2\sqrt{\pi\sigma}) + \frac{1}{2}.$$

The first term $\ln(2\sqrt{\pi\sigma})$ diverges in the limit $\sigma \to 0$. However, it is a constant type of infinity (just like the term $\ln\left(\frac{2T\pi}{k^2 V}\right)$ term) and hence can be ignored by saying that we simply shift the absolute entropy by that "infinite" amount.

Furthermore, the second term will simply give the $O(\sigma^0)$ term in both the numerator and the denominator in the limit $\sigma \to 0$ under the assumption that none of the even derivatives of $\mathcal{M}(s, x)$ diverge. Collecting everything together we obtain the expression eq.(3.8).

## B.2 Quantum relative entropy

In this appendix, we delineate the detailed steps leading to relative entropy expression eq.(3.18). We start with eq.(3.13), using $g_i(x) = \delta_\sigma(x - x_i)$ to have

$$
\begin{aligned}
D\left(\rho_A^{(1)} \| \rho_A^{(2)}\right) &= \int_{-1}^1 dx\, \delta_\sigma(x - x_1) \frac{|\mathcal{M}(s, x)|^2}{\mathcal{I}_g(s, x_1)} \ln\left(\frac{\delta_\sigma(x - x_1)}{\delta_\sigma(x - x_2)} \frac{\mathcal{I}_g(s, x_2)}{\mathcal{I}_g(s, x_1)}\right) \\
&= \int_{-1}^1 dx\, \delta_\sigma(x - x_1) \mathcal{M}_1(s, x) \ln\left(\frac{\delta_\sigma(x - x_1)}{\delta_\sigma(x - x_2)}\right) \\
&\quad + \ln\left(\frac{\mathcal{I}_g(s, x_2)}{\mathcal{I}_g(s, x_1)}\right)\left(\int_{-1}^1 dx\, \delta_\sigma(x - x_1) \mathcal{M}_1(s, x)\right) \approx \mathcal{I}_R + \ln\left(\frac{\mathcal{I}_g(s, x_2)}{\mathcal{I}_g(s, x_1)}\right),
\end{aligned}
\tag{B.27}
$$

where, like before, the second integral is simply 1 since it is just $\int_{-1}^1 dx\, \mathcal{P}_{g_1}(x) = 1$ and

$$
\mathcal{I}_R := \frac{1}{\mathcal{I}_g(s, x_1)} \int_{-1}^1 dx\, \delta_\sigma(x - x_1) |\mathcal{M}(s, x)|^2 \ln\left(\frac{\delta_\sigma(x - x_1)}{\delta_\sigma(x - x_2)}\right),
\tag{B.28}
$$

where $\mathcal{I}_{E,2}$ is defined in eq.(B.8). The rest of our efforts will be directed towards calculating and simplifying $\mathcal{I}_R$. Using eq.(3.5), we have

$$
\begin{aligned}
\mathcal{I}_R &= \frac{1}{2\sqrt{\pi\sigma}} \int_{-1}^1 dx\, e^{-\frac{(x-x_1)^2}{4\sigma}} \mathcal{M}_1(s, x) \ln\left[\text{Exp}\left\{-\frac{1}{4\sigma}\left[(x - x_1)^2 - (x - x_2)^2\right]\right\}\right] \\
&= \frac{\Delta x}{4\sqrt{\pi\sigma^3}} \int_{-1}^1 dx\, e^{-\frac{(x-x_1)^2}{4\sigma}} \mathcal{M}_1(s, x)\left[x - \frac{(x_1 + x_2)}{2}\right]
\end{aligned}
\tag{B.29}
$$

with $\Delta x := x_1 - x_2$. Now, performing a similar partial wave expansion as in eq.(B.12) we get,

$$
\mathcal{I}_R = \sum_{L=0}^\infty \frac{\Delta x}{4\sqrt{\pi\sigma^3}} \frac{\mathcal{M}_L(s)}{\mathcal{I}_g(s, x_1)} \int_{-1}^1 dx\, e^{-\frac{(x-x_1)^2}{4\sigma}} x^L\left[x - \frac{(x_1 + x_2)}{2}\right].
\tag{B.30}
$$

Now, we perform the same change of variables as in the last section:

$$
y = \frac{x - x_1}{\sqrt{2\sigma}} \implies x = \sqrt{2\sigma}\, y + x_1.
$$

Using this in eq.(B.30), we get (please note again that $-1 \le x_1 \le 1$)

$$
\begin{aligned}
\mathcal{I}_R &= \sum_{L=0}^\infty \frac{\Delta x}{4\sqrt{\pi\sigma^3}} \frac{\mathcal{M}_L(s)}{\mathcal{I}_g(s, x_1)} \int_{-\frac{(1+x_1)}{\sqrt{2\sigma}}}^{\frac{(1-x_1)}{\sqrt{2\sigma}}} \sqrt{2\sigma}\, dy\, e^{-\frac{y^2}{2}} (\sqrt{2\sigma}\, y + x_1)^L (\sqrt{2\sigma} y + \frac{\Delta x}{2}) \\
&\approx \sum_{L=0}^\infty \sum_{i=0}^L \frac{\mathcal{M}_L(s)}{\mathcal{I}_g(s, x_1)} \binom{L}{i} x_1^{L-i} (\sqrt{2\sigma})^i \left[\frac{\Delta x}{\sqrt{2\sigma}} < y^{i+1} >_{0,1} + \frac{(\Delta x)^2}{4\sigma} < y^i >_{0,1}\right].
\end{aligned}
\tag{B.31}
$$

Here we have approximated the limits like the previous section and $< y^i >_{0,1}$ is as in eq.(B.14). Following a similar procedure as in the previous section *i.e.*, changing the order of the infinite

sum in eq.(B.31) and using eq.(B.15), we obtain

$$
\begin{aligned}
\mathcal{I}_R &= \frac{\Delta x}{\mathcal{I}_g(s,x_1)} \sum_{i=0}^{\infty} \frac{(\sqrt{2\sigma})^{i-1}}{i!} \frac{\partial^i}{\partial x^i} (|\mathcal{M}(s,x)|^2) \Big|_{x_1} < y^{i+1} >_{0,1} \\
&+ \frac{(\Delta x)^2}{2\mathcal{I}_g(s,x_1)} \sum_{i=0}^{\infty} \frac{(\sqrt{2\sigma})^{i-2}}{i!} \frac{\partial^i}{\partial x^i} (\mathcal{M}(s,x)) \Big|_{x_1} < y^i >_{0,1} \\
&\equiv \mathcal{I}_{R,1} + \mathcal{I}_{R,2} .
\end{aligned}
\tag{B.32}
$$

Next we use eq.(B.14). Since only odd spins contribute, we substitute $2i + 1 \rightarrow i$ and hence get

$$
\begin{aligned}
\mathcal{I}_{R,1} &= \frac{\Delta x}{\mathcal{I}_g(s,x_1)} \sum_{i=0}^{\infty} \sigma^i \frac{\partial^{2i+1}}{\partial x^{2i+1}} (|\mathcal{M}(s,x)|^2) \Big|_{x_1} \frac{(1.3.5...(2i+1))2^i}{(1.2.3.4...(2i).(2i+1))} , \\
&= \frac{\Delta x}{\mathcal{I}_g(s,x_1)} \sum_{i=0}^{\infty} \frac{\partial^{2i+1}}{\partial x^{2i+1}} (|\mathcal{M}(s,x)|^2) \Big|_{x_1} \frac{\sigma^i}{i!} .
\end{aligned}
\tag{B.33}
$$

Similarly,

$$
\mathcal{I}_{R,2} = \frac{(\Delta x)^2}{4\sigma \, \mathcal{I}_g(s,x_1)} \sum_{i=0}^{\infty} \frac{\partial^{2i}}{\partial x^{2i}} (|\mathcal{M}(s,x)|^2) \Big|_{x_1} \frac{\sigma^i}{i!} = \frac{(\Delta x)^2}{4\sigma} ,
\tag{B.34}
$$

where, we have used eq.(B.18) in the last step. Therefore, we return to our original goal and finally get the relative entropy as the following:

$$
\begin{aligned}
D\left(\rho_A^{(1)} \| \rho_A^{(2)}\right) &\approx \ln\left( \sum_{i=0}^{\infty} \frac{\partial^{2i}}{\partial x^{2i}} (|\mathcal{M}(s,x)|^2) \Big|_{x_2} \frac{\sigma^i}{i!} \right) - \ln\left( \sum_{i=0}^{\infty} \frac{\partial^{2i}}{\partial x^{2i}} (|\mathcal{M}(s,x)|^2) \Big|_{x_1} \frac{\sigma^i}{i!} \right) \\
&+ \frac{\Delta x}{\left( \sum_{i=0}^{\infty} \frac{\partial^{2i}}{\partial x^{2i}} (|\mathcal{M}(s,x)|^2) \Big|_{x_1} \frac{\sigma^i}{i!} \right)} \left( \sum_{i=0}^{\infty} \frac{\partial^{2i+1}}{\partial x^{2i+1}} (|\mathcal{M}(s,x)|^2) \Big|_{x_1} \frac{\sigma^i}{i!} \right) + \frac{(\Delta x)^2}{4\sigma} \\
&\xrightarrow[\text{leading order}]{\sigma \rightarrow 0} \frac{(\Delta x)^2}{4\sigma} + \ln\left( \frac{|\mathcal{M}(s,x_2)|^2}{|\mathcal{M}(s,x_1)|^2} \right) + \frac{\Delta x}{|\mathcal{M}(s,x_1)|^2} \left( \frac{\partial}{\partial x} (|\mathcal{M}(s,x)|^2) \Big|_{x_1} \right) ,
\end{aligned}
\tag{B.35}
$$

where, in the first line it is actually the log of ratio of two series. Therefore, the terms inside the log are dimensionless as they should be. It's just for ease of writing that we have separated the two. We also note that the term $\frac{(\Delta x)^2}{4\sigma}$ is responsible for divergence in the limit $\sigma \rightarrow 0$ and hence we cannot take $F_i(x)^2$ to be the delta functions exactly. Furthermore, since this is the relative entropy, we cannot just simply shift the infinity away as we would have done in the absolute case. Also , we have that $D(\rho \| \rho) = 0$ as it should be since ($\Delta x = 0, x_1 = x_2$ for $\rho_A^{(1)} = \rho_A^{(2)}$).

Now, for small $\Delta x$, we can further simplify the relative entropy by expanding the log term up-to second order in $\Delta x$ since the other two terms are of first and second order in $(\Delta x)$ respectively and hence will dominate the other sub-leading terms higher than second order.

Therefore, We have (remember $x_2 = x_1 - \Delta x$),

$$
\begin{aligned}
\ln\left(\frac{\mathcal{I}_g(s, x_1 - \Delta x)}{\mathcal{I}_g(s, x_1)}\right) &= \ln\left(1 - \frac{\Delta x}{\mathcal{I}_g(s, x_1)}\frac{\partial}{\partial x}(\mathcal{I}_g(s, x))\Big|_{x_1} + \frac{(\Delta x)^2}{2\,\mathcal{I}_g(s, x_1)}\frac{\partial^2}{\partial x^2}(\mathcal{I}_g(s, x))\Big|_{x_1} + O((\Delta x)^3)\right) \\
&= -\frac{\Delta x}{\mathcal{I}_g(s, x_1)}\frac{\partial}{\partial x}(\mathcal{I}_g(s, x))\Big|_{x_1} \\
&\quad + \frac{(\Delta x)^2}{2}\left(\frac{1}{\mathcal{I}_g(s, x_1)}\frac{\partial^2}{\partial x^2}(\mathcal{I}_g(s, x))\Big|_{x_1} - \left(\frac{1}{\mathcal{I}_g(s, x_1)}\frac{\partial}{\partial x}(\mathcal{I}_g(s, x))\Big|_{x_1}\right)^2\right) \\
&\quad + O((\Delta x)^3),
\end{aligned}
$$
(B.36)

which can be substituted back into eq.(B.35) in the exact form. It would give the expression (for small $\Delta x$) as

$$
D\left(\rho_A^{(1)}\|\rho_A^{(2)}\right) \approx \frac{(\Delta x)^2}{4\sigma} + \frac{(\Delta x)^2}{2}\frac{\partial^2}{\partial x^2}\left(\ln\left(\frac{\mathcal{I}_g(s, x)}{\mathcal{I}_g(s, x_1)}\right)\right)\Big|_{x_1} + O((\Delta x)^3).
$$
(B.37)

However, if substituted while only keeping the terms leading in $\sigma$, we get a really simple expression of the form (alternatively, taking the limit $\sigma \to 0$ in eq.(B.37)),

$$
\begin{aligned}
D\left(\rho_A^{(1)}\|\rho_A^{(2)}\right) &\approx \frac{(\Delta x)^2}{4\sigma} + \frac{(\Delta x)^2}{2}\left(\frac{\partial^2}{\partial x^2}\left(\frac{|\mathcal{M}(s, x)|^2}{|\mathcal{M}(s, x_1)|^2}\right)\Big|_{x_1} - \left(\frac{\partial}{\partial x}\left(\frac{|\mathcal{M}(s, x)|^2}{|\mathcal{M}(s, x_1)|^2}\right)\Big|_{x_1}\right)^2\right) \\
&\quad + O((\Delta x)^2\sigma).
\end{aligned}
$$
(B.38)

It is important to note that the order of the limits does *not* matter in this case, both the orders give the same answer (taking the leading terms in $(\Delta x)$ of the limiting form in eq.(B.35) would have given the same result as eq.(B.38)).

Furthermore, the $\Delta x$ term in the limiting form of eq.(B.35) can never be leading because in the physically sensible case of $\sigma \ll \Delta x$ *i.e.*, the angular separation of the states being considered is much higher than the resolution of the detector, the $\frac{(\Delta x)^2}{4\sigma}$ term will always dominate the former.

Lastly, we see that in the limit $\sigma \to 0$, if we are able to justify physically the neglecting of the diverging term $(\Delta x)^2/4\sigma$, then the leading term in $\Delta x$ is the term quadratic in $\Delta x$ for any order of $\sigma$.

## B.3 Réyni divergence

In this appendix we try to find the form of $T_n(\rho_A^{(1)}\|\rho_A^{(2)})$. Starting with eq.(3.24), one obtains

$$
\begin{aligned}
T_n\left(\rho_A^{(1)}\|\rho_A^{(2)}\right) &= \int_{-1}^{1} dx\,(\mathcal{P}_{g_1}(x))^n(\mathcal{P}_{g_2}(x))^{1-n}, \\
&= (\mathcal{I}_g(s, x_1))^{-n}(\mathcal{I}_g(s, x_2))^{n-1}\int_{-1}^{1} dx\, g_n^{12}(x)\,|\mathcal{M}(s, x)|^2,
\end{aligned}
$$
(B.39)

where,

$$
g_n^{12}(x) := \left(\frac{1}{2\sqrt{\pi\sigma}}e^{-\frac{(x-x_1)^2}{4\sigma}}\right)^n\left(\frac{1}{2\sqrt{\pi\sigma}}e^{-\frac{(x-x_2)^2}{4\sigma}}\right)^{1-n} = e^{-\frac{(\Delta x)^2}{4\sigma}n(1-n)}\left(\frac{1}{2\sqrt{\pi\sigma}}e^{-\frac{(x-x_n^{12})^2}{4\sigma}}\right),
$$
(B.40)

with $x_n^{12} := nx_1 + (1-n)x_2$. Therefore, we have

$$T_n\left(\rho_A^{(1)}\|\rho_A^{(2)}\right) = e^{-\frac{(\Delta x)^2}{4\sigma}n(1-n)}\left(\frac{\mathcal{I}_g(s,x_2)}{\mathcal{I}_g(s,x_1)}\right)^{n-1}\left(\frac{\mathcal{I}_g(s,x_n^{12})}{\mathcal{I}_g(s,x_1)}\right),\tag{B.41}$$

which, in the limit $\sigma \to 0$, gives

$$T_n\left(\rho_A^{(1)}\|\rho_A^{(2)}\right) \to e^{-\frac{(\Delta x)^2}{4\sigma}n(1-n)}\left(\frac{|\mathcal{M}(s,x_2)|^2}{|\mathcal{M}(s,x_1)|^2}\right)^{n-1}\left(\frac{|\mathcal{M}(s,x_n^{12})|^2}{|\mathcal{M}(s,x_1)|^2}\right),\tag{B.42}$$

with $x_n^{12} = nx_1 + (1-n)x_2$, $\Delta x = x_1 - x_2$.

Let us verify our previously derived expressions in eq.(B.35) in both the exact forms and the limiting version. Firstly, we note that,

$$\frac{\partial}{\partial n}(\mathcal{I}_g(s,x_n^{12})) = (\Delta x)\sum_{i=0}^{\infty}\frac{\partial^{2i+1}}{\partial x^{2i+1}}(|\mathcal{M}(s,x)|^2)\Bigg|_{x_n^{12}}.\tag{B.43}$$

Using this, along with the fact that $x_n^{12} \to x_1$ as $n \to 1$ we find that,

$$\lim_{n\to 1}\frac{\partial}{\partial n}\left(T_n\left(\rho_A^{(1)}\|\rho_A^{(2)}\right)\right) = \frac{(\Delta x)^2}{4\sigma} + \ln\left(\frac{\mathcal{I}_g(s,x_2)}{\mathcal{I}_g(s,x_1)}\right) + \frac{\Delta x}{\mathcal{I}_g(s,x_1)}\frac{\partial}{\partial x}(\mathcal{I}_g(s,x))\Bigg|_{x_1},\tag{B.44}$$

which matches exactly with the expression from eq.(B.35). Furthermore, if we had repeated this exercise with the limiting form given in eq.(B.42), we would have found the limiting form as in eq.(B.35). Therefore, taking the limit $\sigma \to 0$ commutes with taking the derivative followed by the limit $n \to 1$.

Lastly, using eq.(B.42) and the definitions, we can easily see the form of the Réyni Divergence coming out to be

$$D_n\left(\rho_A^{(1)}\|\rho_A^{(2)}\right) = n\frac{(\Delta x)^2}{4\sigma} + \ln\left(\frac{\mathcal{I}_g(s,x_2)}{\mathcal{I}_g(s,x_1)}\right) + \frac{1}{n-1}\ln\left(\frac{\mathcal{I}_g(s,x_n^{12})}{\mathcal{I}_g(s,x_1)}\right),$$
$$\xrightarrow{\sigma\to 0} n\frac{(\Delta x)^2}{4\sigma} + \ln\left(\frac{|\mathcal{M}(s,x_2)|^2}{|\mathcal{M}(s,x_1)|^2}\right) + \frac{1}{n-1}\ln\left(\frac{|\mathcal{M}(s,x_n^{12})|^2}{|\mathcal{M}(s,x_1)|^2}\right).\tag{B.45}$$

Taking the limit of this is straightforward as follows:

$$\lim_{n\to 1}D_n\left(\rho_A^{(1)}\|\rho_A^{(2)}\right) = \frac{(\Delta x)^2}{4\sigma} + \ln\left(\frac{\mathcal{I}_g(s,x_2)}{\mathcal{I}_g(s,x_1)}\right) + \lim_{n\to 1}\left(\frac{1}{n-1}\ln\left(\frac{\mathcal{I}_g(s,x_n^{12})}{\mathcal{I}_g(s,x_1)}\right)\right).\tag{B.46}$$

Now, we have that,

$$\frac{1}{n-1}\ln\left(\frac{\mathcal{I}_g(s,x_n^{12})}{\mathcal{I}_g(s,x_1)}\right) = \frac{1}{n-1}\ln\left(\frac{\mathcal{I}_g(s,x_1+(n-1)(\Delta x))}{\mathcal{I}_g(s,x_1)}\right),$$
$$= \frac{\Delta x}{\mathcal{I}_g(s,x_1)}\frac{\partial}{\partial x}(\mathcal{I}_g(s,x))\Bigg|_{x_1} + O(n-1),\tag{B.47}$$

where we have considered first, the expansion of $\mathcal{I}_g(s,x_1+(n-1)(\Delta x))$ in $\Delta x$, and then used the expansion of the resulting logarithm. So all the higher order terms *w.r.t* $(n-1)$ will go to 0 in the limit $n \to 1$. Therefore, what we are left with is exactly the expression in eq.(B.35). The same exercise could have been repeated with the $\sigma \to 0$ form of the Réyni divergence to get the corresponding limiting form of the Relative Entropy!

### B.4 Quantum information variance

We start with eq.(3.29) where, $T_n(\rho_A^{(1)}\|\rho_A^{(2)})$ is as in eq.(B.42). To take the derivatives and simplify we will use the notation

$$T_n\left(\rho_A^{(1)}\|\rho_A^{(2)}\right) = T_1(n)T_2(n)T_3(n)\,, \tag{B.48}$$

where

$$T_1(n) := e^{\frac{(\Delta x)^2}{4\sigma}n(n-1)}, \quad T_2(n) := \left(\frac{\mathcal{I}_g(s,x_2)}{\mathcal{I}_g(s,x_1)}\right)^{n-1}, \quad T_3(n) := \left(\frac{\mathcal{I}_g(s,x_n^{12})}{\mathcal{I}_g(s,x_1)}\right), \tag{B.49}$$

such that all of the above tend to 1 as $n \to 1$. Their respective derivatives are as follows:

$$\frac{\partial}{\partial n}(T_1(n)) = e^{\frac{(\Delta x)^2}{4\sigma}n(n-1)}\frac{(\Delta x)^2}{4\sigma}(2n-1)\,, \tag{B.50}$$

$$\frac{\partial^2}{\partial n^2}(T_1(n)) = e^{\frac{(\Delta x)^2}{4\sigma}n(n-1)}\left(\frac{(\Delta x)^2}{4\sigma}\right)^2(2n-1)^2 + e^{\frac{(\Delta x)^2}{4\sigma}n(n-1)}2\frac{(\Delta x)^2}{4\sigma}\,. \tag{B.51}$$

Similarly,

$$\frac{\partial}{\partial n}(T_2(n)) = \left(\frac{\mathcal{I}_g(s,x_2)}{\mathcal{I}_g(s,x_1)}\right)^{n-1}\ln\left(\frac{\mathcal{I}_g(s,x_2)}{\mathcal{I}_g(s,x_1)}\right) \tag{B.52}$$

$$\frac{\partial^2}{\partial n^2}(T_2(n)) = \left(\frac{\mathcal{I}_g(s,x_2)}{\mathcal{I}_g(s,x_1)}\right)^{n-1}\left(\ln\left(\frac{\mathcal{I}_g(s,x_2)}{\mathcal{I}_g(s,x_1)}\right)\right)^2\,. \tag{B.53}$$

Lastly,

$$\frac{\partial}{\partial n}(T_3(n)) = \frac{\Delta x}{\mathcal{I}_g(s,x_1)}\frac{\partial}{\partial x}(\mathcal{I}_g(s,x))\Big|_{x_n^{12}}\,, \tag{B.54}$$

$$\frac{\partial^2}{\partial n^2}(T_3(n)) = \frac{(\Delta x)^2}{\mathcal{I}_g(s,x_1)}\frac{\partial^2}{\partial x^2}(\mathcal{I}_g(s,x))\Big|_{x_n^{12}}\,. \tag{B.55}$$

After some straightforward algebra making use of these various derivatives, we obtain

$$V\left(\rho_A^{(1)}\|\rho_A^{(2)}\right) = \frac{(\Delta x)^2}{2\sigma} + (\Delta x)^2\frac{\partial^2}{\partial x^2}\left(\ln\left(\frac{\mathcal{I}_g(s,x)}{\mathcal{I}_g(s,x_1)}\right)\right)\Big|_{x_1}\,, \tag{B.56}$$

which, in leading order in $\sigma \to 0$, gives

$$V\left(\rho_A^{(1)}\|\rho_A^{(2)}\right) \approx \frac{(\Delta x)^2}{2\sigma} + (\Delta x)^2\frac{\partial^2}{\partial x^2}\left(\ln\left(\frac{|\mathcal{M}(s,x)|^2}{|\mathcal{M}(s,x_1)|^2}\right)\right)\Big|_{x_1}\,. \tag{B.57}$$

## C  Generalized Scattering Configurations

In the main text, we delineated the entanglement analysis of the scattering $A + B \to A + B$, $A$ and $B$ being non-identical particles, in details. In this appendix, we spell out a detailed analysis of the most general $2 \to 2$ scattering $A + B \to C + D$. Here, $C, D$ and $A, B$ can be identical. Also $C, D$ can be different from $A, B$. We assume that, in terms of mass, either we have $m_A = m_C$, $m_B = m_D$ or $m_A = m_D$, $, m_B = m_C$. In this setup, some algebraic steps are least cumbersome. It can be generalized quite straightforwardly to all unequal masses. We will

comment on that in a while. Furthermore, we are assuming that we are scattering off bosonic particles without spin. These particles may or may not have internal quantum numbers like isospin. We will work with a generic two-particle state

$$|\vec{p}, \mu; \vec{q}, \nu\rangle := a_\mu^\dagger(\vec{p}) a_\nu^\dagger(\vec{q}) |\Omega\rangle, \tag{C.1}$$

where, $|\Omega\rangle$ is the bosonic Fock vacuum. These states are normalized according to eq.(A.15) *i.e.*,

$$\begin{aligned}
\langle \vec{p}, \mu; \vec{q}, \nu \,|\, \vec{k}, \alpha; \vec{l}, \beta \rangle = \Big[ &2 E_{\vec{p}}^\mu \, 2 E_{\vec{q}}^\nu \, \delta^{(\tilde{D})}(\vec{p} - \vec{k}) \, \delta^{(\tilde{D})}(\vec{q} - \vec{l}) \, \delta_{\mu,\alpha} \, \delta_{\nu,\beta} \\
&+ 2 E_{\vec{p}}^\mu \, 2 E_{\vec{q}}^\nu \, \delta^{(\tilde{D})}(\vec{p} - \vec{l}) \, \delta^{(\tilde{D})}(\vec{q} - \vec{k}) \, \delta_{\mu,\beta} \, \delta_{\nu,\alpha} \Big],
\end{aligned} \tag{C.2}$$

with $E_{\vec{p}}^i = \sqrt{\vec{p}^2 + m_i^2}$ and $\tilde{D}$ is the dimension.

Now, let us consider the initial state before scattering to be

$$|\vec{k}, a; -\vec{k}, b\rangle, \tag{C.3}$$

where, we are in the centre of mass frame. This state corresponds to the *A* particle to be in state $|\vec{k}, a\rangle$ and the *B* particle to be in state $|-\vec{k}, b\rangle$ where, these are single particle Fock states as in eq.(A.8), $|\vec{p}, \alpha\rangle := \sqrt{2E_{\alpha\vec{p}}} \, a_\alpha^\dagger(\vec{p})$. We will introduce the short-hand notation $|\vec{k}; a, b\rangle\!\rangle := |\vec{k}, a; -\vec{k}, b\rangle$ for our convenience.

The state after scattering is given by,

$$\mathcal{S} |\vec{k}; a, b\rangle\!\rangle. \tag{C.4}$$

Next, we need to project this state onto the two-particle state $|\vec{q}_1, c : \vec{q}_2, d\rangle$ in the background of detector geometry as explained in Section (2). To do so, we introduce the projector $\mathcal{Q}_{CD}^{(F)}$ given by

$$\mathcal{Q}_{CD}^{(F)} := \int d\Pi_{\vec{q}_1}^c \, d\Pi_{\vec{q}_2}^d \, F(\theta_{\vec{q}_1 d}) \, |\vec{q}_1, c, \vec{q}_2, d\rangle \langle \vec{q}_1, c, \vec{q}_2, d|, \qquad d\Pi_{\vec{k}}^i := \frac{d^{\tilde{D}} \vec{p}_i}{2 E_{\vec{k}}^i}, \tag{C.5}$$

where $F$ is the same as in Section (2.1). Then, we have the target final state as

$$|f_{CD}\rangle = \int d\Pi_{\vec{q}_1}^c \, d\Pi_{\vec{q}_2}^d \, F(\theta_{1\vec{q}_1 d}) \, |\vec{q}_1, c, \vec{q}_2, d\rangle \langle \vec{q}_1, c, \vec{q}_2, d| \mathcal{S} |\vec{k}; a, b\rangle\!\rangle. \tag{C.6}$$

Now, we consider a density matrix for the joint system in this state $|f_{CD}\rangle$.

$$\begin{aligned}
|f_{CD}\rangle \langle f_{CD}| = \int d\Pi_{\vec{q}_1}^c \, d\Pi_{\vec{q}_2}^d \, d\Pi_{\vec{r}_1}^c \, d\Pi_{\vec{r}_2}^d F(\theta_{1 d\vec{q}_1}) F(\theta_{1 d\vec{r}_1}) & |\vec{q}_1, c, \vec{q}_2, d\rangle \\
& \langle \vec{r}_1, c, \vec{r}_2, d| \langle \vec{q}_1, c, \vec{q}_2, d| \mathcal{S} |\vec{k}; a, b\rangle\!\rangle \\
& \langle\!\langle \vec{k}; a, b| \mathcal{S}^\dagger |\vec{r}_1, c, \vec{r}_2, d\rangle.
\end{aligned} \tag{C.7}$$

However, this is not quite the density matrix because, $|f_{CD}\rangle$ is not correctly normalized. Thus, the correct density matrix is

$$\rho_{CD}^{(F)} := \frac{|f_{CD}\rangle \langle f_{CD}|}{\langle f_{CD} | f_{CD}\rangle}. \tag{C.8}$$

It is quite straightforward to find that

$$\langle f_{CD} | f_{CD}\rangle = \frac{\delta^{(\tilde{D}+1)}(0)}{4k(4E_{\vec{k}})} \tag{C.9}$$

$$\int d^{\tilde{D}} \vec{p} \, \delta(p - k) \left[ F(\theta_{1p})^2 \delta_{cd} + 2\delta_{cd} F(\theta_{1p}) F(\pi - \theta_{1p}) + F(\pi - \theta_{1p})^2 \right] \tag{C.10}$$

$$\left| \langle\!\langle \vec{p}; c, d | \mathbf{s} | \vec{k}; a, b\rangle\!\rangle \right|^2. \tag{C.11}$$

Based on this density matrix we can construct various reduced density matrices. Especially, we can trace out the $D$ particle states to obtain the reduced density matrix $\rho_c$:

$$\rho_C = \frac{\int d\Pi_{\tilde{p}} \left[\delta_{cd}(F(\theta_{1p})^2 + 2F(\theta_{1p})F(\pi - \theta_{1p})) + F(\pi - \theta_{1p})^2\right] |\vec{p}, d\rangle \langle \vec{p}, d| \, \delta(p - k) \left|\langle\!\langle \vec{p}; c, d|\mathbf{s}|\vec{k}; a, b\rangle\!\rangle\right|^2}{\delta^{(\tilde{D})}(0) \int d^{\tilde{D}}\vec{p} \, \delta(p - k) \left[\delta_{cd}(F(\theta_{1p})^2 + 2F(\theta_{1p})F(\pi - \theta_{1p})) + F(\pi - \theta_{1p})^2\right] \left|\langle\!\langle \vec{p}; c, d|\mathbf{s}|\vec{k}; a, b\rangle\!\rangle\right|^2} \, . \tag{C.12}$$

As in the main text, we will consider $F(\theta_1)$ as a Gaussian approximation of the delta function. We have three separate cases to consider.

1. Particles are identical ($\delta_{cd} = 1$) and the mean of the Gaussian is 0. This implies that,

$$F(\theta_1) = F(\pi - \theta_1) \, . \tag{C.13}$$

The $\rho_C$ now becomes

$$\rho_C = \frac{1}{\delta^{(\tilde{D})}(0)} \frac{\int d\Pi_{\tilde{p}} \, (F(\pi - \theta_{1p})^2) \, |\vec{p}, c\rangle \langle \vec{p}, c| \, \delta(p - k) \left|\langle\!\langle \vec{p}; c, c|\mathbf{s}|\vec{k}; a, b\rangle\!\rangle\right|^2}{\int d^{\tilde{D}}\vec{p} \, \delta(p - k) \, (F(\pi - \theta_{1p})^2) \left|\langle\!\langle \vec{p}; c, c|\mathbf{s}|\vec{k}; a, b\rangle\!\rangle\right|^2} \, , \tag{C.14}$$

which, further gives

$$\text{tr}_C(\rho_C)^n = \left[\frac{\delta(0)}{2^{\tilde{D}-2}\pi k^2 \delta^{(\tilde{D})}(0)}\right]^{n-1} \int_{-1}^{1} dx [\mathcal{P}_g(x)]^n \, , \tag{C.15}$$

with

$$\mathcal{P}_g(x) := \frac{g(x)|\langle\!\langle \vec{p}; c, c|\mathbf{s}|\vec{k}; a, b\rangle\!\rangle|^2}{\int_{-1}^{1} dx \, g(x)|\langle\!\langle \vec{p}; c, c|\mathbf{s}|\vec{k}; a, b\rangle\!\rangle|^2} \, , \tag{C.16}$$

where, we have again defined $F(-x)^2 := g(x)$.

2. Particles are identical and mean of the Gaussian distribution $F$ is not 0. This causes the cross terms to vanish since their contribution is negligible compared to the square terms because of different support. We have

$$\rho_C = \frac{1}{\delta^{(\tilde{D})}(0)} \frac{\int d\Pi_{\tilde{p}} \, (F(\theta_{1p})^2 \delta_{cd} + F(\pi - \theta_{1p})^2) \, |\vec{p}, c\rangle \langle \vec{p}, c| \, \delta(p - k) \left|\langle\!\langle \vec{p}; c, c|\mathbf{s}|\vec{k}; a, b\rangle\!\rangle\right|^2}{\int d^{\tilde{D}}\vec{p} \, \delta(p - k) \, (F(\theta_{1p})^2 + F(\pi - \theta_{1p})^2) \left|\langle\!\langle \vec{p}; c, c|\mathbf{s}|\vec{k}; a, b\rangle\!\rangle\right|^2} \, . \tag{C.17}$$

Using this, one gets

$$\rho_C^n = \left[\frac{\delta(0)}{\delta^{(\tilde{D})}(0)}\right]^{n-1} \frac{\int d^{\tilde{D}}\vec{p} \, \delta(p - k) \left[(F(\theta_{1p})^2 + F(\pi - \theta_{1p})^2) \left|\langle\!\langle \vec{p}; c, c|\mathbf{s}|\vec{k}; a, b\rangle\!\rangle\right|^2\right]^n}{\left[\int d^{\tilde{D}}\vec{p} \, \delta(p - k) \, (F(\theta_{1p})^2 + F(\pi - \theta_{1p})^2) \left|\langle\!\langle \vec{p}; c, c|\mathbf{s}|\vec{k}; a, b\rangle\!\rangle\right|^2\right]^n} \, . \tag{C.18}$$

Now let us evaluate integral in the denominator first. Upon using polar co-ordinates and carrying out integrals over $(\varphi_2, \ldots, \varphi_{\tilde{D}-2})$ and the radial integral using the delta function, we are only left with the $x$ integral. Then, if we make the substitution $x \to -x$, the integral of the first part of the integrand becomes the same as the second integral, since the amplitude must be symmetric in $t$ and $u$ in the identical case. Hence we have:

$$I_{den} = 2^{\tilde{D}-1}\pi k^2 \int_{-1}^{1} (F(-x)^2) \left|\langle\!\langle \vec{p}; c, c|\mathbf{s}|\vec{k}; a, b\rangle\!\rangle\right|^2 \, . \tag{C.19}$$

Now, when evaluating the numerator, we follow similar procedure. Then, we carry out a binomial expansion of $(F(\theta_{1p})^2 + F(\pi - \theta_{1p})^2)^2$ which will have cross terms but their contribution will be negligible since the approximate delta functions will not have the same

support. Hence, we can approximate the integrand to have only $(F(-x)^{2n} + F(x)^{2n})(...)$. Consequently, we can easily see again that changing variables $x \to -x$ in the first part of the total integral makes it the same as the second part. Therefore, we have

$$I_{num} = 2^{\tilde{D}-1} \pi k^2 \int_{-1}^{1} dx \, F(-x)^{2n} \left[ \left| \langle\!\langle \vec{p}; c, c | \mathbf{s} | \vec{k}; a, b \rangle\!\rangle \right|^2 \right]^n. \tag{C.20}$$

Now combining both results we have the following expression:

$$tr_C(\rho_C^n) = \left[ \frac{\delta(0)}{2^{\tilde{D}-1} \pi k^2 \delta^{(\tilde{D})}(0)} \right]^{n-1} \int_{-1}^{1} [\mathcal{P}_F(x)]^n, \tag{C.21}$$

where,

$$\mathcal{P}_F(x) := \frac{F(-x)^2 \, | \langle\!\langle \vec{p}; c, c | \mathbf{s} | \vec{k}; a, b \rangle\!\rangle |^2}{\int_{-1}^{1} dx \, F(-x)^2 \, | \langle\!\langle \vec{p}; c, c | \mathbf{s} | \vec{k}; a, b \rangle\!\rangle |^2}. \tag{C.22}$$

3. When the particles are non-identical $\delta_{cd} = 0$. Here we simply have

$$\rho_C = \frac{1}{\delta^{(\tilde{D})}(0)} \frac{\int d\Pi_{\vec{p}} \, (F(\pi - \theta_{1p})^2) \, |\vec{p}, c\rangle \langle \vec{p}, c| \, \delta(p-k) \, \left| \langle\!\langle \vec{p}; c, d | \mathbf{s} | \vec{k}; a, b \rangle\!\rangle \right|^2}{\int d^{\tilde{D}} p \, \delta(p-k) \, (F(\pi - \theta_{1p})^2) \, \left| \langle\!\langle \vec{p}; c, d | \mathbf{s} | \vec{k}; a, b \rangle\!\rangle \right|^2}. \tag{C.23}$$

This is essentially same as the $A + B \to A + B$ analysis and therefore, we have

$$tr_C(\rho_C)^n = \left[ \frac{\delta(0)}{2^{\tilde{D}-2} \pi k^2 \delta^{(\tilde{D})}(0)} \right]^{n-1} \int_{-1}^{1} dx [\mathcal{P}_F(x)]^n, \tag{C.24}$$

with

$$\mathcal{P}_F(x) := \frac{F(-x)^2 \, | \langle\!\langle \vec{p}; c, d | \mathbf{s} | \vec{k}; a, b \rangle\!\rangle |^2}{\int_{-1}^{1} dx \, F(-x)^2 \, | \langle\!\langle \vec{p}; c, d | \mathbf{s} | \vec{k}; a, b \rangle\!\rangle |^2}. \tag{C.25}$$

So far, we have considered scattering configurations with a specific choice of masses. Now, we are going to relax that and consider scattering event for $A + B \to C + D$ with all unequal masses. One can throw in all kinds of other quantum numbers other than spin of course. This case is a straightforward generalization of case 3 above. In fact, in this case we can obtain a quite straightforward generalization of eq.(C.24) above

$$tr_C(\rho_C)^n = \left[ \frac{\delta(0)}{2^{\tilde{D}-2} \pi \, h(k) \delta^{(\tilde{D})}(0)} \right]^{n-1} \int_{-1}^{1} dx [\mathcal{P}_F(x)]^n, \tag{C.26}$$

with $\mathcal{P}_F(x)$ being same as in eq.(C.25) and

$$h(k) = h(k; m_A, m_B, m_C, m_D) := k^2 \left[ \frac{2k^2 - 2\sqrt{(m_A^2 + k^2)(m_B^2 + k^2)} + \Delta m^2 + m_A^2 + m_B^2}{2k^2 - 2\sqrt{(m_A^2 + k^2)(m_B^2 + k^2)} + \Delta m^2 + m_C^2 + m_D^2} \right]$$
$$+ \Delta m^2 \left[ \frac{\Delta m^2}{4} + \sqrt{(m_A^2 + k^2)(m_B^2 + k^2)} \right] + (m_A^2 m_B^2 - m_C^2 m_D^2), \tag{C.27}$$

where, we have defined, $\Delta m^2 := m_A^2 + m_B^2 - m_C^2 - m_D^2$. Observe that, there is only change in the overall multiplicative factor. Thereby, the expression for $D_Q$ remains same as eq.(3.19). Also, in the two special cases mentioned previously, we can see the simplification as

$$h(k; m_A, m_B, m_A, m_B) = h(m_A, m_B, m_B, m_A) = k^2. \tag{C.28}$$

## C.1 Generalized relative entropy

In this section, we repeat the steps of Appendix (B.2) for eq.(3.36). However, we shall only do that for the identical case when both $x_1$ and $x_2$ are not equal to 0 or $x_2 = 0$, while the non-identical case just gives the previously obtained answer as in eq.(3.16). The remaining case of $x_1 = 0$ will result in meaningless complication and hence will be avoided. We still have two cases depending upon whether $x_1$ and $x_2$ have the same or opposite signs:

1. $x_1$ and $x_2$ have the same signs:

$$D(\rho_C(x_1)\|\rho_C(x_2)) \approx \ln\left(\frac{\mathcal{I}_g(s, x_2)}{\mathcal{I}_g(s, x_1)}\right) + (\Delta x)\frac{\partial}{\partial x}\left(\ln\left(\frac{\mathcal{I}_g(s, x)}{\mathcal{I}_g(s, x_1)}\right)\right)\Bigg|_{x_1} + \frac{(\Delta x)^2}{4\sigma}, \tag{C.29}$$

with $\Delta x = x_1 - x_2$.

2. $x_1$ and $x_2$ have the opposite signs:

$$D(\rho_C(x_1)\|\rho_C(x_2)) \approx \ln\left(\frac{\mathcal{I}_g(s, -x_2)}{\mathcal{I}_g(s, x_1)}\right) + (\Delta x)\frac{\partial}{\partial x}\left(\ln\left(\frac{\mathcal{I}_g(s, x)}{\mathcal{I}_g(s, x_1)}\right)\right)\Bigg|_{x_1} + \frac{(\Delta x)^2}{4\sigma}, \tag{C.30}$$

with $\Delta x = x_1 + x_2$.

since when they are of opposite signs, the terms surviving in eq.(3.34) are different than when they are of the same signs. All the cases can now be combined into the following with our previous definitions modified ever so slightly

$$D(\rho_C(x_1)\|\rho_C(x_2)) \approx \ln\left(\frac{\mathcal{I}_g(s, |x_2|)}{\mathcal{I}_g(s, |x_1|)}\right) + (\Delta x)\frac{\partial}{\partial x}\left(\ln\left(\frac{\mathcal{I}_g(s, x)}{\mathcal{I}_g(s, |x_1|)}\right)\right)\Bigg|_{|x_1|} + \frac{(\Delta x)^2}{4\sigma}, \quad \text{(C.31)}$$

with $\Delta x := |x_1| - |x_2|$. This is so as

$$\Delta x := |x_1| - |x_2| = \begin{cases} x_1 - x_2 & \text{if } x_1 > 0, x_2 > 0 \\ x_1 + x_2 & \text{if } x_1 > 0, x_2 < 0 \\ -(x_1 + x_2) & \text{if } x_1 < 0, x_2 > 0 \\ -(x_1 - x_2) & \text{if } x_1 < 0, x_2 < 0 \end{cases}. \tag{C.32}$$

Now, the amplitude squared $\mathcal{M}_{a,b}^{c,d}(s, x)$ is an even function of $x$ due to crossing symmetry. Hence we have that $\mathcal{I}_g(s, x_1) = \mathcal{I}_g(s, |x_1|)$ since even derivatives of even functions are still even while odd derivatives of even functions are odd. This is now enough to see why eq.(C.31) is valid. We start with the case $x_1 > 0, x_2 > 0$ where it is obviously correct since taking the absolute value doesn't change anything. Next, if we consider $x_1 > 0, x_2 < 0$, in eq.(3.36) will be exactly the same as eq.(B.35) with $x_2 \to -x_2$ and hence in the original expression, $\Delta x = x_1 - x_2 \to x_1 + x_2$ which is exactly what the modified definition of $\Delta x$ gives. Furthermore, when we then consider $x_1 < 0, x_2 < 0$, we see that changing back to the first case via $x_1 \to -x_1 > 0$ and $x_2 \to -x_2 > 0$ does two things. Firstly, the even derivatives (including the amplitude itself) do not change because of the aforementioned logic. However, the odd derivatives do change signs. Nonetheless, this sign is cancelled due to the change in sign of our modified $\Delta x$ as can be seen in eq.(C.32). Therefore, overall the expression does not change at all. Similarly, the last case of $x_1 < 0, x_2 > 0$ can also be argued to be exactly the same as that in eq.(C.31). Therefore, as long as we are avoiding $x_1 = 0$, our calculations/expressions derived in the non-identical particles final state are valid even in the identical particle scenario.

## C.2 Generalization to external spin

We can generalize the analysis above to external spinning particles–see eg [44]. We will consider massive spinning particle for the present purpose. First, we need to specify the state of the external particles. As before, such single particle states are momentum eigenstates. However, now we need further specifications of the states. We are interested to remain in $3 + 1$ dimensions. These states transform irreducibly under the universal cover of the Little group $SO(3)$ *i.e.*, in the $irrep^n$ of $SU(2)$. Then any single-particle state will have at least three labels

$$|\vec{p}, J, \lambda\rangle .\tag{C.33}$$

The label $J$ is non-negative half-integer called *spin* ($J = 0, 1/2, 1, 3/2, 2, \dots$). The spin $(2n + 1)/2$, $n \in \mathbb{Z}^{\geq}$ representations are the fermionic states. The label $\lambda$ denotes the components of the spin $J$ $irrep^n$. One can choose $\lambda$ to be the helicity, i.e., the projection of spin $J$ on the direction of the momentum $\vec{p}$. $\lambda$ can take $2J + 1$[22] values from $\lambda = -J$ to $\lambda = +J$ in steps of unity. The normalization of the states in eq.(C.33) is chosen as to be

$$\langle \vec{p}, J, \lambda \,|\, \vec{p}', J', \lambda' \rangle = 2E_{\vec{p}}\, \delta_{JJ'}\, \delta_{\lambda\lambda'}\, \delta^{(\breve{D})}(\vec{p} - \vec{p}') .\tag{C.34}$$

Multi-particle states can be constructed out of them following the general prescription of constructing Fock states as reviewed in Appendix A.2. Now, for specific analysis we consider the elastic scattering event of two-particles $A_1, A_2$ with spins, respectively, $J_1, J_2$:

$$A_1 + A_2 \to A_1 + A_2 .\tag{C.35}$$

One example will be $pp$-scattering. However, we are considering here $A_1$ and $A_2$ to be non-identical to set aside the algebraic complication that arises due to identity of particles[23]. Now, we consider an event where we send in polarized particles i.e., the particles are of definite helicities. Furthermore, we also consider that we collect the outgoing particles to be in definite polarized states. This may be a very restrictive situation but this generalizes the analysis for scalar particles quite directly. A more interesting scenario will be presented later in this section. Let $\lambda_1, \lambda_2$ be the initial helicities of the particles $A_1, A_2$ respectively and $\lambda_1', \lambda_2'$ be the respective final helicities. Thus we consider that, initial state of the joint system is $|\vec{k}, J_1, \lambda_1 ; -\vec{k}, J_2, \lambda_2\rangle$[24]. To shorten notation, from now on, we will drop the spin labels and keep only the momentum labels and helicity labels. Thus we will denote the above state by $|\vec{k}; \lambda_1, \lambda_2\rangle\rangle$ and so on. Moreover, we are keeping detector configuration of the scalar scattering as it is. Thus, the projector defined in eq.(2.4) is trivially generalized in this case to

$$^{\lambda_1', \lambda_2'}\mathfrak{Q}_{A_1 A_2}^{(F)} := \int d\Pi_{A_1 \vec{p}_1} d\Pi_{A_2 \vec{p}_2}\, F(\theta_{A_2})\, |\vec{p}_1, \lambda_1'; \vec{p}_2, \lambda_2'\rangle \langle \vec{p}_1, \lambda_1'; \vec{p}_2, \lambda_2'|, \qquad d\Pi_{i\vec{k}} := \frac{d^3\vec{p}}{2E_{i\vec{k}}} .\tag{C.36}$$

Then, we have the target final state

$$|\mathfrak{f}\rangle = {}^{\lambda_1', \lambda_2'}\mathfrak{Q}_{A_1 A_2}^{(F)}\, \mathcal{S}\, |\vec{k}; \lambda_1, \lambda_2\rangle\rangle .\tag{C.37}$$

Again, as before, this state is not automatically normalized. Defining $\mathfrak{N} := \langle \mathfrak{f} | \mathfrak{f} \rangle$, it is straightforward to obtain

$$\mathfrak{N} = \frac{\delta^{(3+1)}(0)}{4p(E_{A_1\vec{p}} + E_{A_2\vec{p}})} \int d^3\vec{p}_1\, \delta(p_1 - p)\, F(\pi - \theta_{\vec{p}_1})^2 \left| \langle\langle \vec{p}_1; \lambda_1' \lambda_2' | \mathbf{s} | \vec{k}; \lambda_1 \lambda_2 \rangle\rangle \right|^2 .\tag{C.38}$$

---

[22]Note that, we are considering massive particles. For massless particles, there are always two helicity states irrespective of spin.

[23]Especially, the treatment of identical fermions will be different from identical bosons due to anticommuting property of fermions. We leave the detailed exploration of entanglement in $pp$-scattering for future work.

[24]We are in CoM frame.

Now, we can consider the density matrix for the joint system to be

$$\tilde{\rho}^{(F)} := \frac{1}{\mathfrak{N}} |\mathfrak{f}\rangle \langle \mathfrak{f}| .  \tag{C.39}$$

Starting from this density matrix, now, one can obtain reduced density matrices by tracing out subsystems systems. Thus, we can trace out $A_2$ to obtain the reduced density matrix

$$\tilde{\rho}_{A_1}^{(F)} = \frac{1}{\delta^{(3)}(0)} \frac{\int d\Pi_{A_1 \vec{p}_1} \, \delta(p_1 - k) \, F(\pi - \theta_{A_1}) \left| \langle\!\langle \vec{p}_1; \lambda_1', \lambda_2' | \mathbf{s} | \vec{k}; \lambda_1, \lambda_2 \rangle\!\rangle \right|^2 |\vec{p}_1, \lambda_1'\rangle \langle \vec{p}_1, \lambda_1'|}{\int d^3 \vec{p}_1 \, \delta(p_1 - k) \, F(\pi - \theta_{A_1}) \left| \langle\!\langle \vec{p}_1; \lambda_1', \lambda_2' | \mathbf{s} | \vec{k}; \lambda_1, \lambda_2 \rangle\!\rangle \right|^2} .  \tag{C.40}$$

Using these density matrix one can now easily follows in footsteps of the analysis done for the scalar scattering case to reach various expressions for entanglement measures. Specifically, we can reach the following generalization of the relative entropy eq.(3.19),

$$D_Q^{\lambda_1', \lambda_2'; \lambda_1, \lambda_2}(\tilde{\rho}_{A_1}^{(1)} \| \tilde{\rho}_{A_1}^{(2)}) = \frac{(\Delta x)^2}{2} \frac{\partial^2}{\partial x^2} \left[ \ln \left( \frac{\left| \mathcal{M}^{\lambda_1', \lambda_2'; \lambda_1, \lambda_2}(s, x) \right|^2}{\left| \mathcal{M}^{\lambda_1', \lambda_2'; \lambda_1, \lambda_2}(s, x_1) \right|^2} \right) \right]_{x = x_1} + O((\Delta x)^3 \sigma) ,  \tag{C.41}$$

with

$$\mathcal{M}^{\lambda_1', \lambda_2'; \lambda_1, \lambda_2}(s, x) \equiv \langle\!\langle \vec{p}_1; \lambda_1', \lambda_2' | \mathbf{s} | \vec{k}; \lambda_1, \lambda_2 \rangle\!\rangle ,  \tag{C.42}$$

where, as before we avoid the forward direction by the detector configuration. While these have been so far quite straightforward, we will now define a new quantity called *unpolarized relative entropy* quite analogous to unpolarized scattering cross-section. By *unpolarized relative entropy* we will mean the quantity

$$\overline{D}_Q(\tilde{\rho}_{A_1}^{(1)} \| \tilde{\rho}_{A_1}^{(2)}) := \sum_{\substack{\lambda_1', \lambda_2' \\ \lambda_1, \lambda_2}} \frac{1}{(2J_1 + 1)(2J_2 + 1)} D_Q^{\lambda_1', \lambda_2'; \lambda_1, \lambda_2}(\tilde{\rho}_{A_1}^{(1)} \| \tilde{\rho}_{A_1}^{(2)}) .  \tag{C.43}$$

One can attempt to study various bootstrap analyses with this quantity for scattering involving spinning particle like, say, $pp$-collision.

While, the above configuration is a perfectly valid one, it misses one very interesting possibility. The above setup misses out entanglement among helicity degrees of freedom for the outgoing particles as a result of the scattering. Let us briefly sketch how one can investigate this. The crux of this investigation is to modify the projector in eq.(C.36) to incorporate the possibility of entanglement between helicity degrees of freedom of the outgoing particles. This is done by defining the new projector

$$\mathfrak{Q}_{A_1 A_2}^{(F)} := \sum_{\lambda_1', \lambda_2'} \int d\Pi_{A_1 \vec{p}_1} d\Pi_{A_2 \vec{p}_2} \, F(\theta_{A_2}) |\vec{p}_1, \lambda_1'; \vec{p}_2, \lambda_2'\rangle \langle \vec{p}_1, \lambda_1'; \vec{p}_2, \lambda_2'| .  \tag{C.44}$$

Now, as before, we consider that the incoming particles are polarized i.e., they are in definite helicity states. Thus, the modified target final state is now given by

$$|\mathfrak{f}'\rangle = \mathfrak{Q}_{A_1 A_2}^{(F)} \mathcal{S} |\vec{k}; \lambda_1, \lambda_2 \rangle\!\rangle ,  \tag{C.45}$$

where again we are in CoM frame. Now, one can consider the density matrix of the joint system to be given by,

$$\rho'^{(F)} := \frac{1}{\mathfrak{N}'} |\mathfrak{f}'\rangle \langle \mathfrak{f}'| ,  \tag{C.46}$$

with $\mathfrak{N}' := \langle \mathfrak{f}' | \mathfrak{f}' \rangle$. Starting from this density matrix one can consider various reduced density matrices by tracing out one of the particle states. Note that this time one can also trace out helicity degree of freedom of one of the outgoing particle. Thus, in this setup we can explore the entanglement between helicities that is generated as a result of scattering. It is worthwhile to mention here that there is a contentious issue regarding Lorentz invariance of quantum entanglement while dealing with spinning particles, see for example [68–72]. We wish to address these issues and explore entanglement assisted bootstrap analysis of spinning particles in future.

## D   Pion-pion scattering

We want to calculate both entanglement entropy and relative entropy for specific pion pion interactions. Both of these necessitate the knowledge of the contribution of various in channels in the amplitude $\mathcal{M}(s,t,u)$. In the following calculations we will drop the momentum label of the states and focus on indices. $\pi^0, \pi^+, \pi^-$ can be defined to be:-

$$|\pi^0\rangle = |0\rangle, \quad |\pi^+\rangle = \frac{1}{\sqrt{2}}\left(|1\rangle + i\,|2\rangle\right), \quad |\pi^-\rangle = \frac{1}{\sqrt{2}}\left(|1\rangle - i\,|2\rangle\right). \tag{D.1}$$

We aim to carry out the above entropy computations for scattering that involve $\pi^0, \pi^+, \pi^-$. Since the basis is orthogonal, the formalism for normalisation and the procedure of finding the entropy remains the same as before. In our calculations above we have taken the trace over $|\vec{p}, d\rangle$ and hence we will calculate entropy with respect to the $C$ particle. So in a generic reaction $A1 + A2 \to A3 + A4$, we can construct two types of final states i.e $|A3, c, A4, d\rangle$ and $|A3, d, A4, c\rangle$. If we use the first final state entropy will be calculated with respect to $A3$ and if we use the second final state the entropy will be calculated with respect to $A4$. This is something we must keep in mind while evaluating the reactions. We first calculate projectors in the (0,1,2) basis.

**Indices**

We first specialize for $a, b, c, d$ taking values in $(0, 1, 2)$. Applying crossing symmetry and O(N) symmetry, the amplitude $\mathcal{M}_{a,b}^{c,d}(s,t,u)$ takes the form:

$$\mathcal{M}_{a,b}^{c,d}(s,t,u) = A(s|t,u)\,\delta_{ab}\,\delta^{cd} + A(t|s,u)\,\delta_a^c\,\delta_b^d + A(u|s,t)\,\delta_a^d\,\delta_b^c. \tag{D.2}$$

The projectors of $O(3)$ take the following form:-

$$\mathbb{P}^0 = \frac{1}{3}(\delta_{ab}\delta^{cd}), \quad \mathbb{P}^1 = \frac{1}{2}(\delta_a^c\delta_b^d - \delta_a^d\delta_b^c), \quad \mathbb{P}^2 = \frac{1}{2}(\delta_a^c\delta_b^d + \delta_a^d\delta_b^c - \frac{2}{3}\delta_{ab}\delta^{cd}). \tag{D.3}$$

Now writing $\mathcal{M}(s,t,u)$ in terms of singlet,symmetric and anti-symmetric Projectors, we have

$$\begin{aligned}
\mathcal{M}_{a,b}^{c,d}(s,t,u) &= (3A(s|t,u) + A(t|s,u) + A(u|s,t))\,\mathbb{P}_{a,b}^{0\,c,d} + (A(t|s,u) - A(u|s,t))\,\mathbb{P}_{a,b}^{1\,c,d} \\
&\quad + (A(t|s,u) + A(u|s,t))\,\mathbb{P}_{a,b}^{2\,c,d}.
\end{aligned} \tag{D.4}$$

Making the following redefinition:-

$$\begin{aligned}
A^0(s,t,u) &= 3A(s|t,u) + A(t|s,u) + A(u|s,t) \\
A^1(s,t,u) &= A(t|s,u) - A(u|s,t) \\
A^2(s,t,u) &= A(t|s,u) + A(u|s,t),
\end{aligned} \tag{D.5}$$

such that $\mathcal{M}_{a,b}^{c,d}(s,t,u)$ now reads:-

$$\mathcal{M}_{a,b}^{c,d}(s,t,u) \;=\; A^0(s,t,u)\,\mathbb{P}_{a,b}^{0\,c,d} \;+\; A^1(s,t,u)\,\mathbb{P}_{a,b}^{1\,c,d} \;+\; A^2(s,t,u)\,\mathbb{P}_{a,b}^{2\,c,d}\,. \tag{D.6}$$

All terms of the form $\langle a,b \,|\, \mathcal{T} \,|\, c,d \rangle$ with three indices unequal disappear. This can be easily seen from eq.(D.3). Furthermore the terms $\langle a,b \,|\, \mathcal{T} \,|\, c,d \rangle$ with three indices equal and one distinct also disappears. The non-vanishing cases are listed below and will be useful when we calculate pion pion reactions.

1. For $a = b = c = d$,

$$\mathbb{P}^0 \;=\; \frac{1}{3},\;\; \mathbb{P}^1 \;=\; 0,\;\; \mathbb{P}^2 \;=\; 1 - \frac{1}{3}\,. \tag{D.7}$$

2. For $a = c \neq b = d$,

$$\mathbb{P}^0 \;=\; 0,\;\; \mathbb{P}^1 \;=\; \frac{1}{2},\;\; \mathbb{P}^2 \;=\; \frac{1}{2}\,. \tag{D.8}$$

3. For $a = b \neq c = d$

$$\mathbb{P}^0 \;=\; \frac{1}{3},\;\; \mathbb{P}^1 \;=\; 0,\;\; \mathbb{P}^2 \;=\; -\frac{1}{3}\,. \tag{D.9}$$

4. For $a = d \neq c = b$

$$\mathbb{P}^0 \;=\; 0,\;\; \mathbb{P}^1 \;=\; -\frac{1}{2},\;\; \mathbb{P}^2 \;=\; \frac{1}{2}\,. \tag{D.10}$$

The cases calculated here exhausts the possibilities of the $\mathcal{T}-$matrix elements that might occur upon the basis change. Now, we will do the cases, where indices take values for $\pi^+, \pi^-$ and $\pi^0$. We can then use these results directly into our expressions for density matrix and entropy derived above.

Table 6: Amplitudes for various pion reactions

| Amplitudes | | |
|:---:|:---:|:---:|
| **Reaction** | **Entropy for** | $\mathcal{M}(s,t,u)$ |
| $\pi^{+(-)} + \pi^{+(-)} \rightarrow \pi^{+(-)} + \pi^{+(-)}$ | $\pi^{+(-)}$ | $A^2(s,t,u)$ |
| $\pi^0 + \pi^0 \rightarrow \pi^0 + \pi^0$ | $\pi^0$ | $\frac{1}{3}A^0(s,t,u) + \frac{2}{3}A^2(s,t,u)$ |
| $\pi^{+(-)} + \pi^0 \rightarrow \pi^{+(-)} + \pi^0$ | $\pi^{+(-)}$ | $\frac{1}{2}\left(A^1(s,t,u) + A^2(s,t,u)\right)$ |
| $\pi^{+(-)} + \pi^0 \rightarrow \pi^{+(-)} + \pi^0$ | $\pi^0$ | $\frac{1}{2}\left(-A^1(s,t,u) + A^2(s,t,u)\right)$ |
| $\pi^+ + \pi^- \rightarrow \pi^0 + \pi^0$ | $\pi^0$ | $\frac{1}{3}\left(A^0(s,t,u) - A^2(s,t,u)\right)$ |
| $\pi^+ + \pi^- \rightarrow \pi^+ + \pi^-$ | $\pi^+$ | $\frac{1}{3}A^0(s,t,u) + \frac{1}{2}A^1(s,t,u) + \frac{1}{6}A^2(s,t,u)$ |
| $\pi^+ + \pi^- \rightarrow \pi^+ + \pi^-$ | $\pi^-$ | $\frac{1}{3}A^0(s,t,u) + \frac{1}{2}A^1(s,t,u) + \frac{1}{6}A^2(s,t,u)$ |

# E   Isospin Entanglement in Pion Scattering

In this appendix, we provide with the technical details for the analysis presented in section 8. We want to quantify entanglement in isospin resulted by pion scattering. Recall that, pions $\pi^0$, $\pi^+$, $\pi^-$ are considered as three states of the same particle in the isospin space. The

states are defined as in eq.(D.1). They transform in the spin-1 representation of $O(3)$[25]. We will calculate quantum relative entropy and another quantity called *entanglement power* for investigating entanglement in isospin.

## E.1   Quantum Relative Entropy

For the analysis that follows we will always remain in the CoM frame. We assume that the incoming particles are moving along $\hat{z}$ so that the in state is given by

$$|in\rangle := |p\hat{z}, a_1; -p\hat{z}, a_2\rangle \,, \tag{E.1}$$

where $a_1, a_2$ are isospin labels for the incoming particles. Each of $a_1, a_2$ can take 3 values. The evolved state under S-Matrix evolution is given by $S|in\rangle$. Now we will project this state onto a two-particle state. However, since we are interested in isospin entanglement only, we will be considering projection onto a definite momentum configuration. Let's consider that the outgoing momentum configuration is (in CoM frame)

$$|q\hat{n}, b_1; -q\hat{n}, b_2\rangle \,, \tag{E.2}$$

where, without any loss of generality, we can consider $\hat{n}$ to be an unit vector lying in $x-z$ plane making an angle $\theta$ with the +ve $\hat{z}$ direction. Thus, in the usual spherical polar coordinates we have

$$\hat{n} \equiv (\sin\theta, 0, \cos\theta) \,. \tag{E.3}$$

By 4−momentum conservation, one has $p = q$. Further, let us take $\theta \neq 0$ strictly. Then, the desired target state is given by

$$\begin{aligned}
|f\rangle &= \sum_{b_1, b_2} |q\hat{n}, b_1; -q\hat{n}, b_2\rangle \langle q\hat{n}, b_1; -q\hat{n}, b_2 |S| p\hat{z}, a_1; -p\hat{z}, a_2\rangle \\
&= (2\pi)^4 \delta^{(4)}(0) \sum_{b_1, b_2} |q\hat{n}, b_1; -q\hat{n}, b_2\rangle \, \mathcal{S}^{b_1 b_2}_{a_1 a_2}(s, \cos\theta) \,,
\end{aligned} \tag{E.4}$$

where we have defined

$$\mathcal{S}^{b_1 b_2}_{a_1 a_2}(s, \cos\theta) := \langle q\hat{n}, b_1; -q\hat{n}, b_2 |S| p\hat{z}, a_1; -p\hat{z}, a_2\rangle \,. \tag{E.5}$$

Since we will be strictly considering non-forward scattering, we can use with impunity

$$\mathcal{S}^{b_1 b_2}_{a_1 a_2}(s, \cos\theta) \equiv \mathcal{M}^{b_1 b_2}_{a_1 a_2}(s, \cos\theta) \,, \tag{E.6}$$

where $\mathcal{M}^{b_1, b_2}_{a_1, a_2}$ is the standard scattering amplitude for $a_1 a_2 \to b_1 b_2$. Then, we can construct the density matrix from the state $|f\rangle$ given by

$$\begin{aligned}
\rho_f = &\mathcal{N} (2\pi)^8 \left[\delta^{(4)}(0)\right]^2 \sum_{b_1, b_2} \sum_{c_1, c_2} \left(\mathcal{M}^{b_1 b_2}_{a_1 a_2}(s, \cos\theta) \mathcal{M}^{a_1 a_2}_{c_1 c_2}(s, \cos\theta)^*\right) \\
&|q\hat{n}, b_1; -q\hat{n}, b_2\rangle \langle q\hat{n}, c_1; -q\hat{n}, c_2| \,,
\end{aligned} \tag{E.7}$$

where, the constant $\mathcal{N}$ is fixed by the normalization requirement $\mathrm{tr}_1 \mathrm{tr}_2 \rho_f = 1$ where the traces are taken *only over the isospin states with respect to the basis* $\{|q\hat{n}, x_1; -q\hat{n}, x_2\rangle\}$. Fixing the normalization constant then, we are left with the final density matrix $\rho_f$ with matrix elements given by

$$\left(\rho_f\right)^{b_1 b_2}_{c_1 c_2}(s, \cos\theta) = \frac{\mathcal{M}^{b_1 b_2}_{a_1 a_2}(s, \cos\theta) \left[\mathcal{M}^{a_1 a_2}_{c_1 c_2}(s, \cos\theta)\right]^*}{\sum_{x,y} \mathcal{M}^{xy}_{a_1 a_2}(s, \cos\theta) \left[\mathcal{M}^{a_1 a_2}_{xy}(s, \cos\theta)\right]^*} \,. \tag{E.8}$$

---

[25]For our purpose we can be content with the representation theory of $SU(2)$!

The reduced density matrix $\rho_1 = \mathrm{tr}_2\,\rho_f$, then, has matrix elements

$$\left(\rho_1\right)^{b_1}_{c_1}(s,\cos\theta) = \frac{\sum_z \mathcal{M}^{b_1 z}_{a_1 a_2}(s,\cos\theta)\left[\mathcal{M}^{a_1 a_2}_{c_1 z}(s,\cos\theta)\right]^*}{\sum_{x,y}\mathcal{M}^{xy}_{a_1 a_2}(s,\cos\theta)\left[\mathcal{M}^{a_1 a_2}_{xy}(s,\cos\theta)\right]^*}. \tag{E.9}$$

To remind ourselves that we are essentially considering finite-dimensional state spaces, the density matrix $\rho_f$ is a $9 \times 9$ matrix and $\rho_1$ is a $3 \times 3$ matrix.

For relative entropy, we will consider another density matrix in obtaining which, one chooses a different external momentum configuration to get the state $|f\rangle$. Let's consider one where $\theta$ is different. Thus, we have now another reduced density matrix $\sigma_1$ with matrix elements given by

$$\left(\sigma_1\right)^{b_1}_{c_1}(s,\cos\bar{\theta}) = \frac{\sum_z \mathcal{M}^{b_1 z}_{a_1 a_2}(s,\cos\bar{\theta})\left[\mathcal{M}^{a_1 a_2}_{c_1 z}(s,\cos\bar{\theta})\right]^*}{\sum_{x,y}\mathcal{M}^{xy}_{a_1 a_2}(s,\cos\bar{\theta})\left[\mathcal{M}^{a_1 a_2}_{xy}(s,\cos\bar{\theta})\right]^*}, \tag{E.10}$$

where, $\bar{\theta} \neq \theta \neq 0$. Now, we can consider the quantum relative entropy $D(\rho_1\|\sigma_1)$.

## E.2  Entanglement Power

The concept of *entanglement power* or *entangling power* of an unitary evolution operator was introduced in [47,48] . Let us briefly recapitulate the basic concept behind this quantity. Unitary transformations $U$ i.e., quantum evolutions acting upon state-space of multi-particle systems describe non-trivial interactions between the degrees of freedom among different subsystems. One can ask how *efficient* is $U$ as *entangler* according to some criterion. [47,48] addressed this issue by considering how much entanglement is produced by $U$ on the average acting on a given distribution of separable i.e., unentangled quantum states. For a bi-partite system with state space $\mathcal{H} = \mathcal{H}_1 \otimes \mathcal{H}_2$, if $E$ is an entanglement measure over the same then *entanglement power* of $U$ with respect to $E$ is defined by

$$e_p(U) := \overline{E(U\,|\beta_1\rangle \otimes |\beta_2\rangle)}^{\,\beta_1,\beta_2}, \quad \beta_i \in \mathcal{H}_i, \tag{E.11}$$

where, the bar denotes the average over all product states $|\beta_1\rangle\otimes|\beta_2\rangle$ with respect to some probability distribution $p(|\beta_1\rangle,|\beta_2\rangle)$. As an entanglement measure of a normalized state $|\psi\rangle \in \mathcal{H}$ was used the *linear* entropy

$$E(|\psi\rangle) = 1 - \mathrm{tr}_1\rho^2, \quad \rho := \mathrm{tr}_2\,|\psi\rangle\langle\psi|. \tag{E.12}$$

Physically, $e_p(U)$ measures how much entanglement $U$ can impart upon an otherwise unentangled state. The important point to note is that, for a given $p(|\beta_1\rangle,|\beta_2\rangle)$, entanglement power is essentially a property of $U$. Thus, one can try to make an *entanglement assisted comparison* between different unitary evolutions in terms of $e_p(U)$.

Recently, [49,50] explored the entanglement power in the context of S-Matrix evolution. In particular, they used a particular form of $p(|\beta_1\rangle,|\beta_2\rangle)$. We will apply their construct to our consideration of isospin entanglement.

First note that, for a fixed momentum configuration we can consider a generic state of the form

$$|k\hat{\alpha}, a_1; -k\hat{\alpha}, a_2\rangle, \tag{E.13}$$

$\hat{\alpha}$ being a generic unit 3–vector, as tensor product state of two single particle isospin states i.e., *for the purpose of isospin considerations only* we can consider

$$|k\hat{n}, a_1; -k\hat{n}, a_2\rangle \equiv |a_1\rangle \otimes |a_2\rangle. \tag{E.14}$$

Now, we define in state to be

$$|\psi_i\rangle := \hat{R}(\Omega_1) \otimes \hat{R}(\Omega_2) |p\hat{z}, a_1; -p\hat{z}, a_2\rangle , \qquad (E.15)$$

where, $\hat{R}(\Omega_i)$ is rotation in the isospin space of the $i^{th}$ particle. In particular, $\hat{R}(\Omega_i)$ acts on the isospin state of $i^{th}$ particle. More specifically, $\hat{R}$ is rotation operator in spin-1 representation of $SU(2)$ because isospin states transform in the same representation. In terms of usual spherical polar coordinates $(\theta_i, \phi_i)$, the rotation operator $\hat{R}(\Omega_i)$ reads

$$\hat{R}(\Omega_i) = e^{-iI_3\phi_i} e^{-iI_2\theta_i} , \qquad (E.16)$$

with

$$I_3 = \begin{pmatrix} 1 & 0 & 0 \\ 0 & 0 & 0 \\ 0 & 0 & -1 \end{pmatrix} , \qquad I_2 = \frac{1}{2}\begin{pmatrix} 0 & -\sqrt{2}i & 0 \\ \sqrt{2}i & 0 & -\sqrt{2}i \\ 0 & \sqrt{2}i & 0 \end{pmatrix} . \qquad (E.17)$$

It is worth mention that the above matrices are expressed in the basis $\{|I_3 = +1\rangle, |I_3 = 0\rangle, |I_3 = -1\rangle\}$ consisting of eigenstates of the $I_3$ operator with indicated eigenvalues. The rotation operator $\hat{R}(\Omega_i)$ then turns out to be, in this basis,

$$\hat{R}(\Omega_i) = \begin{pmatrix} e^{i\phi_i}\cos^2\left(\frac{\theta_i}{2}\right) & -\frac{e^{i\phi_i}\sin(\theta_i)}{\sqrt{2}} & e^{i\phi_i}\sin^2\left(\frac{\theta_i}{2}\right) \\ \frac{\sin(\theta_i)}{\sqrt{2}} & \cos(\theta_i) & -\frac{\sin(\theta_i)}{\sqrt{2}} \\ e^{-i\phi_i}\sin^2\left(\frac{\theta_i}{2}\right) & \frac{e^{-i\phi_i}\sin(\theta_i)}{\sqrt{2}} & e^{-i\phi_i}\cos^2\left(\frac{\theta_i}{2}\right) \end{pmatrix} . \qquad (E.18)$$

Under S-Matrix evolution we have, then, the state $|\psi_i\rangle$ goes to $S|\psi_i\rangle$. Now, for isospin entanglement we need to consider restriction of $S$ to isospin space. However, S-Matrix evolution also causes change in momenta of the scattering particles. Thus we consider projecting the S-Matrix to a definite scattering configuration with respect to momenta and then consider the restriction of the same to the isospin space i.e., the S-Matrix elements of interests are

$$\langle p\hat{n}, c_1; -p\hat{n}, c_2 | S | p\hat{z}, b_1; -p\hat{z}, b_2\rangle = (2\pi)^4\delta^{(4)}(0)\,\mathcal{S}^{c_1 c_2}_{b_1 b_2}(s, \cos\theta) , \qquad (E.19)$$

$\hat{n}$ being given by eq.(E.3). In writing this we have made use of the usual 4-momentum conservation in CoM frame. To emphasize, we are interested in the matrix $\widehat{\mathcal{S}}(s, \cos\theta)$ which acts on the two-particle isospin space and has elements given by $\mathcal{S}^{c_1 c_2}_{b_1 b_2}(s, \cos\theta)$. Thus, the target state is given by:

$$\begin{aligned}
|\psi_f\rangle &:= \frac{1}{w(p)^2}\sum_{\substack{c_1, c_2 \\ b_1, b_2}} |p\hat{n}, c_1; -p\hat{n}, c_2\rangle \langle p\hat{n}, c_1; -p\hat{n}, c_2 | S | p\hat{z}, b_1; -p\hat{z}, b_2\rangle \langle p\hat{z}, b_1; -p\hat{z}, b_2 | \psi_i\rangle \\
&= \frac{1}{w(p)}(2\pi)^4\delta^{(4)}(0)\sum_{\substack{c_1, c_2 \\ b_1, b_2}} |p\hat{n}, c_1; -p\hat{n}, c_2\rangle \mathcal{S}^{c_1 c_2}_{b_1 b_2}(s, \cos\theta)\langle b_1 | \hat{R}(\Omega_1) | a_1\rangle \langle b_2 | \hat{R}(\Omega_2) | a_2\rangle ,
\end{aligned}$$

$$(E.20)$$

where we have used the inner product

$$\langle k\hat{n}, b_1; -k\hat{n}, b_2 | \psi_i\rangle = w(k)\langle b_1 | \hat{R}(\Omega_1) | a_1\rangle \langle b_2 | \hat{R}(\Omega_2) | a_2\rangle , \qquad (E.21)$$

$w(k)$ being some momentum dependent normalization factor dependent on the magnitude of the momentum, $k$. Next, we proceed through the same steps as before to construct the density matrix

$$\rho_{\psi_f} = \bar{\mathcal{N}} |\psi_f\rangle \langle \psi_f| , \qquad (E.22)$$

where, as before $\bar{\mathcal{N}}$ is fixed by the normalization requirement $\mathrm{tr}_1 \mathrm{tr}_2 \rho_{\psi_f} = 1$. Finally, in the basis $\{|q\hat{n}, c_1; -q\hat{n}, c_2\rangle\}$ the matrix elements of $\rho_{\psi_f}$ are given by

$$\left(\rho_{\psi_f}\right)^{b_1 b_2}_{c_1 c_2}(s, \cos\theta) = \tag{E.23}$$

$$\frac{\sum_{x_1, x_2} \sum_{y_1, y_2} \mathcal{M}^{b_1 b_2}_{x_1 x_2}(s, \cos\theta) \left[\mathcal{M}^{y_1 y_2}_{c_1 c_2}(s, \cos\theta)\right]^* \hat{R}(1)^{b_1}_{a_1} \hat{R}(2)^{b_2}_{a_2} (\hat{R}(1)^{c_1}_{a_1} \hat{R}(2)^{c_2}_{a_2})^*}{\sum_{z_1 z_2} \sum_{x_1, x_2} \sum_{y_1, y_2} \mathcal{M}^{z_1 z_2}_{x_1 x_2}(s, \cos\theta) \left[\mathcal{M}^{y_1 y_2}_{z_1 z_2}(s, \cos\theta)\right]^* \hat{R}(1)^{z_1}_{a_1} \hat{R}(2)^{z_2}_{a_2} (\hat{R}(1)^{y_1}_{a_1} \hat{R}(2)^{y_2}_{a_2})^*},$$
$$\tag{E.24}$$

where, $\hat{R}(1)^a_b = \langle a | \hat{R}(\Omega_1) | b \rangle$ and in writing this we are again considering scattering in strictly non-forward direction. From this we can again construct the reduced density matrix by tracing out particle 2 to get

$$\bar{\rho}_1 = \mathrm{tr}_2 \rho_{\psi_f}, \tag{E.25}$$

whose matrix elements are given by

$$\left(\bar{\rho}_1\right)^{b_1}_{c_1} = \sum_{\ell} \left(\rho_{\psi_f}\right)^{b_1 \ell}_{c_1 \ell}(s, \cos\theta). \tag{E.26}$$

Then, the *entanglement power* of the restricted S-Matrix $\hat{\mathcal{S}}(s, \cos\theta)$ is given by [49, 50]

$$\mathcal{E}\left[\hat{\mathcal{S}}(s, \cos\theta)\right] = 1 - \int \frac{d\Omega_1}{4\pi} \frac{d\Omega_2}{4\pi} \mathrm{tr}_1\left[\bar{\rho}_1^2\right], \tag{E.27}$$

where, $d\Omega_i = \sin\theta_i \, d\theta_i \, d\phi_i$ is the usual volume element on $S^2$ in spherical polar coordinates. Let us compare $\mathcal{E}$ above with $e_p$ defined in eq.(E.11). The product state of $|\beta_1\rangle \otimes |\beta_2\rangle$ of eq.(E.11) is now labelled by the spherical polar coordinates $(\Omega_1(\theta_1, \phi_1), \Omega_2(\theta_2, \phi_2))$. Thus we need to consider the probability distribution function $p(\{\theta_1, \phi_1\}, \{\theta_2, \phi_2\})$. It is clear from eq.(E.27) that,

$$p(\{\theta_1, \phi_1\}, \{\theta_2, \phi_2\}) = \left(\frac{1}{4\pi}\right)^2 \sin\theta_1 \sin\theta_2. \tag{E.28}$$

It is straightforward to check that $p(\{\theta_1, \phi_1\}, \{\theta_2, \phi_2\})$ is properly normalized i.e.,

$$\int_0^\pi d\theta_1 \int_0^{2\pi} d\phi_1 \int_0^\pi d\theta_2 \int_0^{2\pi} d\phi_2 \; p(\{\theta_1, \phi_1\}, \{\theta_2, \phi_2\}) = 1. \tag{E.29}$$

Here, it is worth mentioning that the entanglement power $\mathcal{E}$ is defined with respect to the particular probability distribution of eq.(E.28) above. Clearly, this is not a unique choice. It may be worthwhile to study entanglement power defined with respect to other probability distributions. We leave these questions for future exploration.

# F   Numerics

We shall briefly describe the numerical techniques used to obtain the results in Section (7). The ansatz eq.(7.1) is formally an infinite sum. We need to employ a cut off of $N_{max}$, $L_{max}$ to perform computations. Unitarity is imposed on a grid of points uniformly distributed on the upper half plane of $\eta_s$ defined below eq.(7). The unitarity condition $\left|S^{(I)}_\ell(s)\right|^2 \leq 1$ can be cast into a semi-definite condition as mentioned in [16]. Starting with the expansion,

$$S^{(I)}_\ell(s) = 1 + i \, \vec{y} \cdot \vec{f}^{(I)}_\ell(s), \tag{F.1}$$

where $\vec{y}$ is the parameter set and $\vec{f}$ is obtained after integrating over the Legendre polynomial. Using this expansion we can write

$$0 \leq \left(1 - \vec{y}.\vec{I}_\ell^{(I)}\right)^2 + \left(\vec{y}.\vec{R}_\ell^{(I)}\right)^2 \leq 1$$
$$\implies U_\ell \equiv 2\vec{y}.\vec{I}_\ell^{(I)} - (\vec{y}.\vec{I}_\ell^{(I)})^2 - (\vec{y}.\vec{R}_\ell^{(I)})^2 \geq 0,$$
$$\text{and also} \quad U_\ell^{(I)} \leq 1. \tag{F.2}$$

Here, $R_\ell^{(I)} = \mathrm{Re}(f_\ell^{(I)})$ and $I_\ell^{(I)} = \mathrm{Im}(f_\ell^{(I)})$. This can further be written as a matrix of the form,

$$M_\ell^{(I)} \equiv \begin{pmatrix} 1 + \vec{y}.\vec{R}_\ell^{(I)} & 1 - \vec{y}.\vec{I}_\ell^{(I)} \\ 1 - \vec{y}.\vec{I}_\ell^{(I)} & 1 - \vec{y}.\vec{R}_\ell^{(I)} \end{pmatrix}, \tag{F.3}$$

whose positive semi-definiteness will imply unitarity. Numerically, this condition is imposed using S.D.P.B ( [13]). Our calculations have a precision of 200 decimal digits. Unitarity was imposed on 200 points.

To find the lake and the river of fig.(11) and fig.(15) first we impose an Adler zero at a position $s_0$ in $T_0^{(0)}(s)$. Next step is to extremize $T_0^{(2)}(s)$ at various values of $s$. We can impose an Adler zero if the maximum is positive and the minimum is negative. Repeating the procedure for other $s_0$ values we find the figures. This procedure is simple and the boundary can be ideally found by brute force. However we did not have the computational power and hence had to improvise.

We used a makeshift algorithm to obtain the boundary points of fig.(11) and fig.(15) by using information from nearby points. We fix some $s_0$ and carry out a quadratic fit of the $\mathrm{Max}[T_0^{(2)}(s_2)]$ or $\mathrm{Min}[T_0^{(2)}(s_2)]$ close to the boundary. Zeroes of the fitting function should ideally give us the point $s_2^*$ where Max or $\mathrm{Min}[T_0^{(2)}(s_2^*)] = 0$ if the variation is quadratic. This is true in case of the Lake and the approximation works very well even if the approximating points are sufficiently away ($\mathrm{Max}[T_0^{(2)}(s)] \approx 10^{-3}$) and can give results of order upto $10^{-8}$. However the variation of Max or $\mathrm{Min}[T_0^{(2)}(s)]$ for the river is more complicated. In some regions of both the lower bank and the upper bank the Max or $\mathrm{Min}[T_0^{(2)}(s_2)]$ behaves linearly with $s_2$ even when we move closer to the bank. They become quadratic very close to the edge and hence we are required to choose points much closer to the boundary. It is observed that better results are obtained if we choose points from either side of the boundary. Since we knew the approximate location of the lake boundary from [17], re-plotting the lake was easy. However to find the river we had to first generate a grid of allowed region. We discovered that the upper boundary arose because the maximum became negative and the lower boundary because the minimum became positive. After this grid was refined enough, we started employing our algorithm to generate boundary points. All boundary points obtained have an order of $\leq 10^{-6}$. Examples of fitting function for both fig.(11) and fig.(15) are given in fig.(21).

Our computational resources included one 10-core and one 8-core workstations, one 20-core cluster and a 16 core cluster. Given these limited numerical resources we could only work extensively with a cut off of $N_{max} = 10, L_{max} = 11$. However, recently, we have checked that for $N_{max} = 14$ and $L_{max} = 17$ there is virtually no discernible change to our river plots. We hope to work with an increased cut off in the near future and demonstrate convergence explicitly. However we are certain about the validity of our result since fig.(11) matches very well with the one presented in [17]

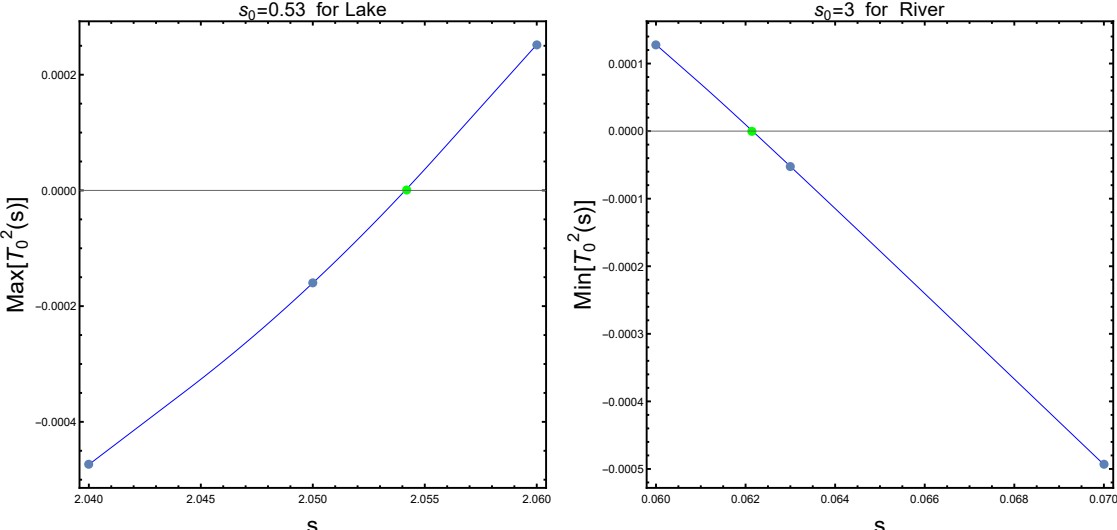

Figure 21: The Blue points are the approximate values and the green point is the quadratic zero prediction. The values at the green point for both are $\approx 10^{-8}$.

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
