# Peer review of "Relative entropy in scattering and the S-matrix bootstrap"

_SciPost Physics, doi:SciPost Phys. 9, 081 (2020)_

## Round 1 · Referee Report · Anonymous (Referee 5) · 2020-9-11

Strengths
1. Original idea and interesting results
2. Detailed calculations
Weaknesses
1. Structure of the paper is difficult to follow
2. Missing more physical discussions of the results
Report
This paper studies how entanglement can be used to constrain the space of valid S-matrices. The authors study the relative entropy between two density matrices which can be interpreted as two different theories or the same theory being measured by a Gaussian detector. The authors go one step further by connecting the relative entanglement entropy with the usual positivity bounds for effective field theories. Moreover, assuming the positivity bounds, they show that the positivity of the relative entanglement can further constrain the space of S-matrices. In particular, it can exclude a big part of the parameter space without the need of experimental inputs.
This is a new and interesting direction and indeed deserves further exploration. Before recommending publication, I would like to ask a few clarifications from the authors.
Requested changes
1. Below Eq. (2.2) the authors write that the analysis differs in crucial points from the ones in Refs. [3]. However, these crucial points are not explained with much detail and I believe it is worth a better comparison with the previous literature. Their main point is that instead of integrating over the full phase space and having to deal with divergences, they integrate just over the part in which the detector is sensible. Then, calculating the relative entropy, they are able to get rid of the detector dependence at least at leading order. However, what is not obvious is how problematic is the regularization issue, that the authors claim to avoid, as the regularization scheme just modifies an overall factor (PRD 100 (2019) 7, 076012 ) and in the end it is not important when looking to the entanglement variations.
2. One point raised in the previous literature is that the density matrix is not Lorentz invariant if one looks to only the spin or momentum entanglement (see e.g PRD 97, 016011 (2018)). As the authors are always looking just to the momentum entanglement, why this is not an issue for the spinning generalization in sec. 2.3 ?
3. The S-matrix is defined as S=1+iT and, of course, the interesting part is in the T-matrix. However, the 1 part should be taken into account in the normalization of the density matrix. It is not clear if this is the case.
4. Regarding the processes in Eq. (4.11), the ones in the first and last lines have the same initial/final Hilbert space, which seems to not be the case for the process in the second line. I wonder if makes sense to compute the entanglement entropy in this case. The authors could clarify this point and also add an explanation in the text.
5. The fact that positivity bounds from the relative entanglement is correlated with the usual positivity bounds, as in Eq. (4.25), seems related with the fact that the relevant quantity in both cases is the second derivative of the amplitude. However, a more physical explanation is missing in the text.
6. Eq. (5.25) implies that the maximum of the relative entanglement is in the forward region. Is there a physical reason? Moreover, this seems to not be the case for the black/red curves of Fig. 5. Why?
7. In general, the relative entropy is not only non-negative but also bounded from below (Rev.Mod.Phys. 90 (2018) 3, 035007). Could this have any impact in the results, giving stronger constrains than just imposing positivity?
Moreover, a few improvements in the paper structure would be desirable. For example, in pag. 7 it is confusing that the angles $\theta_{Dt}$, $\theta_D$ and $\alpha$ are the same, but it would be more convenient to unify the notation for $\theta_{Dt}$ when referring to the detector angle (a similar confusion also appears in other parts). Also, $x$ is occasionally used as $\theta$ (e.g. fig. 3) but it is actually the $\cos\theta$. Section 3.2 seems to consider the case A+B$\rightarrow$A+B but fig.3 refers to A+B$\rightarrow$C+D, which is then only discussed in sec. 3.5. Another confusing point is the $F$ and $g$ functions, as they are basically the same and the only important point is that the support function is chosen as a density probability function in order to ensure the positivity properties of $\mathcal{P}_g$. One could emphasize this point in the definition of Eq. (2.4). In Eq. (3.1) the replica trick is shown and not the definition of the entanglement entropy. For someone who is not familiar with the geographic plots - river, lake, peninsula - it would be helpful to add in the captions that the blue is the allowed region.
Author: Parthiv Haldar on 2020-10-13 [id 1005]
(in reply to Report 1 on 2020-09-11)
1. While the variation of entanglement entropy might have gotten rid of divergence, quantum relative entropy is a bona fide quantum information quantity which has multiple interesting applications like hypothesis testing which have been explored in the paper.
2. We will explore issues related to spinning generalizations in our future exploration.
3. In our normalization we have indeed considered the whole S-Matrix. Equations (2.8)-(2.10) tells this explicitly. The expressions for the density matrix contain matrix elements of S only, (write the mathematical expression). However, the presence of the “angle selector” function makes the choice when to consider $T$ and when not. To be specific, when we considered the angle of the detected particle to be strictly not equal to zero, we have put $S=T$ with impunity.
4. In considering the processes given in (4.11), we have assumed that there is a single two-particle Hilbert space given by the Fock construction. This we can always do. Similar scenarios arise whenever one considers of reactions of the type $A+B\to A+B$, where there are two different particles residing in the same Hilbert space.
5. The implication of eq.(5.25) can't apply to black/red curves of figure 5 because the latter corresponds to chiral perturbation theory which is an effective field theory.
6. The lower bound on the relative entropy involves an extra operator (Eqn. 48, Rev.Mod.Phys. 90 (2018) 3, 035007). We don’t know currently what kind of operator we can use in order to constrain S-Matrix bootstrap. However, we would like to come to this exploration in future.
Other comments regarding notations and figures have been addressed in the new manuscript.
Author: Parthiv Haldar on 2020-10-13 [id 1002]
(in reply to Report 3 by Joao Penedones on 2020-09-21)We list down comments in reply to the points raised by Prof. Penedones in the Referee report.
We have fixed the definition of $D_Q$. Earlier $D_Q$ was defined to be entire sub-leading expansion the relative entropy $D$ about $\sigma=0$ which is not right. The $D_Q$ of importance is only the $\sigma^0$ term which is clearly universal and independent of the human chosen parameter $\sigma$.
We have added a preliminary exploration of entanglement in isospin for pions. We wish to explore these more in future. While we have formally set up the entanglement analysis for spinning particles, we hope to address them in full glory in future endeavour.
We have addressed this issue in a comment below eq 4.10. This serves as a precursor to more logical high energy expansions in the later sections.
We had fixed the parameters $\sigma$ (Gaussian parameter), $x_1$ and $(s-4)$. Changes in these parameters regulate the order of the numbers appearing in Table 1. The numbers become smaller as σ is decreased or $x_1$ is set closer to $0$ or$ (s-4)$ is reduced further. However, the hypothesis testing conclusions are insensitive to small changes in these parameters. Furthermore, note that the integrand of $D(\rho_{boot}||\rho_{\chi PT})$ is not small (order of $10^-2$). The small integrals due to cancellation of the integration in different regions of $x$.
We calculated violation of elastic unitarity for $N_{max} = 16$ and saw that the reduction in elastic unitarity violation is not uniform for all S-matrices. For S-matrices of the green region on the lower boundary, the violation does decrease such that they become admissible into the brown allowed region at $N_{max}=16$. However other We S-matrices still show violations, and in a few cases the violation increases. Hence this is a delicate issue, and we intend to report on this in future.
Our unitarity constraints are for a finite grid of $s$-values and are imposed up to a finite spin. This leads to violations. It is also interesting to note that the violations are exclusively in the extended region outside the Mandelstam triangle.

---

## Round 1 · Referee Report · Anonymous (Referee 4) · 2020-9-18

Strengths
1- The idea is original and has many potential applications.
2- Many examples in the form of known theories are given.
3- Most impressively, the presented methods provide new constraints for the S-matrix bootstrap.
Weaknesses
1- The appearance of the square of the delta function and how it is handled is not really convincing. However, it seems that this cannot be avoided and the obtained results are convincing.
2- It seems that while conditions (5.10) on scattering lengths are rigorously derived, the conditions (5.11) are merely assumed based on two examples (experiments and chiral perturbation theory). Following the bootstrap philosophy of constraining the space of theories on theoretical grounds, it would make sense to also discuss the improvements to the S-matrix bootstrap that follow from (5.10) alone, without imposing (5.11).
3- The name "river" for figure 1 is a bit unfortunate and might be confusing as it turns around the established association of land with allowed regions (lake, peninsula, island, archipelago, ...). "Land bridge" might have been more fitting.
Report
The paper is well written, provides many details and exposes the idea of studying scattering using relative entropy from many angles. It meets the expectations of the journal as it opens a new pathway into the S-matrix bootstrap and provides a novel link between scattering theory and quantum information theory. The general acceptance criteria of the journal are all met. I recommend publication in SciPost Physics after a very minor revision.
Requested changes
I would leave it up to the authors if and to what extend the points under weaknesses are addressed. The following small typos should be corrected:
1- Two of the \rho_1 above (3.8) should be \rho_2.
2- Fix arguments of delta functions in (C.2).
3- Below (C.30) there should be an equality for I_g.
Author: Parthiv Haldar on 2020-10-13 [id 1004]
(in reply to Report 2 on 2020-09-18)
1. The issue regarding square of the delta functions is genuine. However, rigorously speaking the very concept of any power of delta function is ill-defined and confusing. That is why we could not give any more rigorous construction. We have used standard textbook treatment of powers of $\delta(0)$. For the other powers of $\delta(x)$ that appear, we have used a Gaussian regularization. This seemed to be physically reasonable.
2. If we impose just the D-wave inequalities, we would get a larger version of the lake instead of a river. The effect of S-wave inequalities is far larger in comparison to D-wave inequalities. However, our main goal was to fix the sign of $D_Q$ for various pion reactions and for that shall need both S and D wave inequalities to be satisfied together.

---

## Round 1 · Referee Report · Joao Penedones (Referee 1) · 2020-9-21

Report
This paper opens up an interesting connection between quantum information and the S-matrix bootstrap. This is an important contribution that may lead to many fruitful developments in the future.
The authors consider a scattering setup that allows them to define a measure of relative entropy associated with a 2 to 2 scattering amplitude (sections 2 and 3).Then, they compute this relative entropy for several known theories (section 4) and in specific kinematic limits like low and high energy (section 5). Finally, they make some reasonable positivity assumptions and explore their consequences for the S-matrix bootstrap studies of pion scattering (section 7).
I recommend this paper for publication after the authors address the following comments:
1. In the center of mass frame, the two particles in the final state have opposite spatial momentum. This means that measuring the scattering angle of one particle, immediately tells us the scattering angle of the other. We can say that the 2 final particles are maximally entangled in the momentum basis, independently of the theory. This makes it difficult to define a measure of entanglement that is theory dependent. The authors propose equation (1.2). This is an interesting proposal but it has two drawbacks that are worth emphasizing:
- it is entirely classical because it only depends on the cross section. Formula (3.11) is the classical relative entropy between two probability distributions. From this, we conclude that $D_Q$ basically measures how the prior distribution (given by the differential cross section) affects the Gaussians of the detectors.
- it depends on human chosen parameters like the detector width $\sigma$ and separation $\Delta x$. Moreover, it diverges when $\sigma \to 0$.
2. The intuition of the previous paragraph suggests that one should focus on extra degrees of freedom like for example the helicity. The authors comment on this generalization (appendix C.2). I encourage them to pursue this further. It would be natural to consider the final density matrix only over the spin degrees of freedom and define the entanglement entropy of the two final particle in this space. This would be finite and independent of the detector details. Moreover, it would be sensitive to quantum effects. It would also be very interesting to use the isospin quantum numbers and compute this quantity for pion scattering.
3. Equation (4.9) is the high energy limit of an amplitude obtained from a low energy EFT. This does not make sense (e.g. the next term in the EFT expansion would dominate over this one).
4. Why are the numbers so small in table 1? What sets this scale?
5. In section 7.5, the authors comment on the violations of elastic unitarity of the numerical results. In my experience, these violations (for given fixed spin and fixed energy) decrease when one increases the number of parameters in the ansatz ($N_{max}$). Therefore, I do not understand why one should eliminate part of the parameter space using this criterium.
6. In section 7.6, the authors discuss the impact of imposing some extra positivity constraints (7.8 - 7.10). I am surprised that this changed the allowed region as shown in figure 20. If I understand correctly these positivity constraints follow from analyticity, crossing and unitarity that were already imposed. Therefore, they should be automatic. Perhaps they are not automatic because there are small unitarity violations between the energy grid points chosen for SDPB?
Typos or minor comments:
1. Page 9: What is the reason for the notation change from $F$ to $g$?
2. Above equation (3.8) $\rho_1$ is repeated.
3. LHS of 4.2 is not correct
4. There is a square missing on the LHS of (6.2)
5. After equation (7.5) "... plotting the the lake boundary ..."
6. Page 53: "This integral will be of central in many analyses that follow."
7. There some typos that can be found with a spellchecker. For instance "non-idenitical" on page 66.

---

## Round 2 · Referee Report · Joao Penedones (Referee 3) · 2020-10-26

Strengths

1-

Report

The new version of the paper successfully addresses most of the comments raised by the referees. In particular, the new discussion about entanglement in isospin space is a very significant improvement of the paper. Therefore, I recommend the paper for publication.

I have only one minor question: is it possible to simplify the expression for entanglement power and write it solely in terms of the phase shifts? A similar result was obtained in equation (3) of reference [42] for the case of nucleon scattering.

Equation (E.11) has two typos (the second $\beta_1$ in the state and $b_2$ in the exponent should both be $\beta_2$).
  • validity: -
  • significance: -
  • originality: -
  • clarity: -
  • formatting: -
  • grammar: -

Author:  Parthiv Haldar  on 2020-11-06  [id 1036]

(in reply to Report 1 by Joao Penedones on 2020-10-26)

We had looked into an analytic formula for the entanglement power, $\mathcal{E}$, akin to equation (3) of reference[42]. But, the problem is that there is a complicated angle dependence in the denominator that makes analytic evaluation of the final two angular integrals not possible (at least we have not managed it).

---

## Round 2 · Referee Report · Anonymous (Referee 2) · 2020-10-30

Report

All the changes requested by me were implemented, as well as several larger changes requested by the other referees. I recommend the paper in this form for publication.

---

## Round 2 · Referee Report · Anonymous (Referee 1) · 2020-11-2

Report

I recommend the publication of this paper.

---

## Round 2 · List of Changes

List of Changes to the Draft To address various points raised in invited referee reports, we did some significant changes to the draft. We list out these in the following. 1. We added an analysis of entanglement in isospin for pions in section 8 supplemented by Appendix E. We explore quantum relative entropy and a quantity called entanglement power. In Appendix E.2, we give a concise review of the basic concept of entanglement power for the convenience of the reader as well as to make the presentation self-contained. In the main text, i.e. in section 8, we provide results of some elementary numerical analysis. We report some preliminary findings from the numerical exploration in regards to entanglement power in the “Future Directions” section, which we wish to present in future work.

  1. We shifted the formal setup for analysis involving spinning particle entirely to the appendix, removing the corresponding section from the main text. Since, in this work we consider only entanglement in momentum space and isospin space for spinless particle, we thought it to be judicious to keep the explorations for the spinning particle to the appendix. The appendix serves as the formal point of invitation for future exploration. As rightly pointed out by the honourable, the issue of Lorentz invariance of entanglement involving spinning one is a significant one. We wish to address this fully in future endeavour.

  2. Corrected an issue in the definition of $D_Q$ in eq.(3.19). The $D_Q$ does not depend upon the human chosen parameter $\sigma$ and is universal.

  3. Added a footnote on page 10 (footnote 3) to address the confusion surrounding the notation of $F$ and $g$ around equation (2.14) . We would like to point out that $F$ is the function entering the projector $Q_{AB}^{(F)}$ defined in eq.(2.4). $F$ is assumed to be a function of $\theta$ without any special form assumed. In the final density matrix, it is $F^2$, and not $F$, that enters. We have used $g$ to denote $F^2$. Further, at this point we have assumed that $F$ is a function of $\theta_A$ via $\cos\theta_A$, i.e. $F(\theta_A)\equiv F(\cos\theta_A)$. Denoting $\cos\theta_A$ by $x$, we have defined $g(x):=F(-x)^2$. This notation follows to all subsequent references to $F$ and $g$. Thus, mathematically, $F$ and $g$ are not same.

  4. We made changes to both figure 2 and figure 3 to clear out confusions.

  5. We fixed various typos. Further, we thoroughly checked the spelling and grammatical constructs using Grammarly.

---

## Editorial Decision

published